# Quantifying input uncertainty in the calibration of water quality models: reordering errors via the secant method

Xia Wu [1,2,3,4], Lucy Marshall [2*], and Ashish Sharma [2]

[1] State Key Laboratory of Hydrology-Water Resources and Hydraulic Engineering, Hohai University, Nanjing, 210098, China

[2] UNSW Water Research Center, School of Civil and Environmental Engineering, University of New South Wales, Sydney, NSW 2052, Australia

[3] College of Hydrology and Water Resources, Hohai University, Nanjing, 210098, China

[4] CMA-HHU Joint Laboratory for Hydrometeorological Studies, Hohai University, Nanjing, 210098, China

*Correspondence to*: Lucy Marshall (lucy.marshall@unsw.edu.au)

**Abstract.** Uncertainty in inputs can significantly impair parameter estimation in water quality modeling, necessitating accurate quantification of input errors. However, decomposing input error from model residual error is still challenging. This study develops a new algorithm, referred to as Bayesian error analysis with reordering (BEAR), to address this problem. The basic approach requires sampling errors from a pre-estimated error distribution and then reordering them with their inferred ranks via the secant method. This approach is demonstrated in the case of total suspended solids (TSS) simulation via a conceptual water quality model. Based on case studies using synthetic data, the BEAR method successfully improves the input error identification and parameter estimation by introducing the error rank estimation and the error position reordering. The results of a real case study demonstrate that even with the presence of model structural error and output data error, the BEAR method can approximate the true input and bring a better model fit through an effective input modification. However, its effectiveness depends on the accuracy and selection of the input error model. The application of the BEAR method in TSS simulation can be extended to other water quality models.

## 1 Introduction

For robust water management, uncertainty analysis is of growing importance in water quality modeling (Refsgaard et al., 2007). It can provide knowledge of error propagation and the magnitude of uncertainty impacts in model simulations to guide improved predictive performance (Radwan et al., 2004). However, the implementation of uncertainty analysis in water quality models (WQMs) is still challenging due to complex interactions among sources of multiple errors, generally caused by a simplified model structure (structural uncertainty), imperfect observed data (input uncertainty and observation uncertainty in calibration data) and limited parameter identifiability (parametric uncertainty) (Refsgaard et al., 2007).

Among them, input uncertainty is expected to be particularly significant in a WQM, interpreted here as the observation uncertainty of any input data. Observation uncertainty is different from other sources of uncertainty in modeling since these uncertainties arise independently of the WQM itself, thus, their properties (e.g. probability distribution family and distribution parameters) can, at least in principle, be estimated prior to the model calibration and simulation by analysis of the data acquisition instruments and procedures (McMillan et al., 2012). Rode and Suhr (2007) and Harmel et al. (2006) reviewed the uncertainty associated with selected water quality variables based on the empirical quality of observations. The general methodology developed in their studies can be extended to the analysis of other water quality variables. Besides the error coming from the measurement process, the error from surrogated data is another major source of input uncertainty (McMillan et al., 2012). Measurements of water quality variables often lack desirable temporal and spatial resolutions, thus, the use of surrogate or proxy data is necessary for improved inference of water quality parameters (Evans et al., 1997, Stubblefield et al., 2007). The probability distribution of the surrogate error is easy to estimate from the residuals between the measurements and proxy values. In this process, the measurement errors are ignored given the errors introduced from the surrogate process are commonly much larger than the measurement errors (McMillan et al., 2012). These estimated error distributions are "prior knowledge" of input uncertainty before any model calibration and can serve as the a-priori uncertainty estimation in the modeling process.

Input uncertainty can lead to bias in parameter estimation in water quality modeling (Chaudhary and Hantush, 2017, Kleidorfer et al., 2009, Willems, 2008). Improved model calibration requires isolating the input uncertainty from the total uncertainty. However, the precise quantification of time-varying input errors is still challenging when other types of uncertainties are propagated through to the model results. In hydrological modeling, several approaches have been developed to characterize time-varying input errors, and these may hold promise for application in WQMs. The Bayesian total error analysis (BATEA) method provides a framework that has been widely used (Kavetski et al., 2006). Time-varying input errors are defined as multipliers on the input time series and inferred along with the model parameters in a Bayesian calibration scheme. This leads to a high-dimensionality problem, which cannot be avoided (Renard et al., 2009) and restricts the application of this approach to the assumption of event-based multipliers (the same multiplier applied to all time steps of one storm event). In the Integrated Bayesian Uncertainty Estimator (IBUNE) (Ajami et al., 2007) approach, multipliers are not jointly inferred with the model parameters, but sampled from the assumed distribution and then filtered by the constraints of simulation fitting. This approach reduces the dimensionality significantly and can be applied in the assumption of data-based multiplier (one multiplier for one input data point) (Ajami et al., 2007). However, this approach is less effective because the probability of co-occurrence of all optimal error values is very low, resulting in an underestimation of the multiplier variance and misidentification of the uncertainty sources (Renard et al., 2009).

To complete this goal, this study attempts to add a reordering strategy into the IBUNE framework and names this developed algorithm as Bayesian Error Analysis with Reordering (BEAR). The derivation and details of the BEAR algorithm in quantifying input errors are described in Sect. 2. Section 3 introduces the build-up/wash-off model (BwMod) to illustrate this

approach. Its model input, streamflow, often suffers from observational errors from a rating curve. By comparing the results with other calibration frameworks, the ability of the BEAR method is explored in two synthetic and one real case study. In this way, the new algorithm is tested in a controlled situation (with the knowledge of the true error and data value) and in a realistic

situation (with the interference of multiple error sources) respectively. Section 3 describes the setup of these case studies and Sect. 4 demonstrates their results. Section 5 evaluates the BEAR method and its implementation. Finally, Section 6 outlines the main conclusions and recommendations for this work.

## 2 Methodology

### 2.1 Basic theory of identifying the input error in model calibration

A WQM in the ideal situation without any error can be described as

$$Y^* = M(X^* \mid \theta^*) \tag{1}$$

where the asterisk $^*$ implies the true value without error, and the true output $Y^*$ is simulated by the perfect model $M$ with the true input $X^*$ and the true model parameter $\theta^*$. Here and in the following contents, a capital bold letter (e.g. $X, Y$) represents a vector and a lower case (e.g. $x, y$) represents a scalar.

In reality, the model input $X^o$ (typically the rainfall or streamflow in a WQM) inevitably suffers from input error $\varepsilon_X$. This will result in a calibrated model parameter $\theta^c$ biased from the true value $\theta^*$ (Kleidorfer et al., 2009). Thus, under the assumption that the output data and model structure are generally without errors and the input errors are additive to the true input data $X^*$, the model residual $\varepsilon$ in a traditional calibration can be described by

$$\varepsilon = Y^o - Y^s = Y^o - M(X^o \mid \theta^c) = Y^* - M(X^* + \varepsilon_X \mid \theta^c), \varepsilon_X \sim N(\mu_X, \sigma_X^2) \tag{2}$$

where $Y^s$ is the output simulated from the model $M$ corresponding to the observed input $X^o$ and model parameter $\theta^c$, and the observed output $Y^o$ is assumed without observational errors in the derivation, thus can be denoted as $Y^*$. Input error $\varepsilon_X$ is assumed to follow a Gaussian distribution with mean $\mu_X$ and variance $\sigma_X^2$.

It should be noted that this study focuses on identifying the input errors in the process of parameter estimation and the derivation of the BEAR method is based on the assumption that the model only suffers from input error and parameter error,

but other sources of error (i.e. model structural error and output observational error) are inevitable in the WQM and can impair the effectiveness of input error identification and parameter estimation. Considering this realistic situation, the ability of the

BEAR method will be tested later in one synthetic and one real case study where the interference of other sources of error has been considered.

To counter the influence of input errors in a traditional calibration, a common approach is to subtract estimated errors $\boldsymbol{\varepsilon}_X^p$ from the observed input $X^o$. This is illustrated as the "proposed" approach and the superscript $p$ represents the values in this "proposed" approach. The residual $\boldsymbol{\varepsilon}^p$ will change to

$$\boldsymbol{\varepsilon}^P = Y^o - Y^p = Y^* - M(X^p \mid \boldsymbol{\theta}^P) = Y^* - M(X^o - \boldsymbol{\varepsilon}_X^p \mid \boldsymbol{\theta}^P) = Y^* - M(X^* + \boldsymbol{\varepsilon}_X - \boldsymbol{\varepsilon}_X^p \mid \boldsymbol{\theta}^P) \qquad (3)$$

If the equivalence between $\boldsymbol{\varepsilon}_X$ and $\boldsymbol{\varepsilon}_X^p$ can be ensured for each data point, the modified input $X^p$ then becomes the same as the true value $X^*$. The proposed calibration (Eq. (3)) will turn into an ideal calibration where the optimal parameters $\boldsymbol{\theta}^p$ will lead to the same simulation corresponding to the true values $\boldsymbol{\theta}^*$ and the model residual $\boldsymbol{\varepsilon}^p$ will decrease to zero. If the inverse problem (from the zero residual to find the optimal parameter) is not unique, the calibrated parameter $\boldsymbol{\theta}^p$ may not converge to the true parameter $\boldsymbol{\theta}^*$, but lead to the same simulation as what the true parameter corresponds to. In this study, these parameters are also denoted as $\boldsymbol{\theta}^*$ and called ideal model parameters. Besides, if the identified input error and the model parameter can compensate each other, multiple combinations of model parameter and input error may yield zero residual and their estimates will be biased from the ideal values. A possible way to weaken this compensation effect will be explored in Sect. 5.3. Although the aforementioned problems cannot be avoided, selecting the optimal input error series according to the model residual error is the basic theory of not only this study but also current methods identifying the input errors (i.e. BATEA (Kavetski et al., 2006). and IBUNE (Ajami et al., 2007)).

The above approach does not improve the input error model itself but improves the WQM specification to have parameters closer to what would be achieved under no error conditions. Then the model can be more effectively used for scenario analysis (where we may know the hydrologic regime of a catchment in a hypothetical future), for forecasting under the assumption of perfect inputs (where the driving hydrologic forecast is independently obtained via a numerical weather prediction and a hydrologic model) or for regionalization of the WQM (where the model is transferred to a catchment without data). In all of these cases, an ideal model should have unbiased parameter estimates. This is our goal in identifying the optimal input errors, not to use the model for predictions with input data suffering the same errors.

## 2.2 The introduction of error rank estimation via the secant method

Unlike directly estimating the input error value via existing methods, this study attempts to transform the input error quantification into the rank domain. Here, the rank is defined as the order of any individual value relative to the other sampled values, and determines the relative magnitude of each error in all data errors. For example, in the 1st iteration in Table A 1, the error at the 15th time step, -0.29, is the smallest value among all the sampled errors, therefore, its rank is 1. In current methods,

an assumption of input error model is necessary to set, which provides an overall distribution for the estimated input errors. If the knowledge of the error distribution (i.e. cumulative distribution function (CDF) of input errors) has been got, the error value only depends on its rank in this distribution. Therefore, under the condition of a certain input error model, the rank estimation will bring similar results as the direct value estimation. Besides, the rank estimation has a few advantages over the direct value estimation. The discussion on this is stated in Sect. 5.1.

In the rank domain, the challenge turns to find a way to effectively adjust the input error rank to push the residual error equal to zero. The secant method can be applied to address this problem. It is an iterative process to produce better approximations to the roots of a real-valued equation (Ralston and Jennrich, 1978). Here, the root is the optimal rank of each input error and the equation is the corresponding model residual equal to zero. The secant method (Ralston and Jennrich, 1978) can be repeated as

$$k_{i,q} = k_{i,q-1} - \varepsilon_{i,q-1}^{p} \frac{k_{i,q-1} - k_{i,q-2}}{\varepsilon_{i,q-1}^{p} - \varepsilon_{i,q-2}^{p}} \tag{4}$$

until a sufficiently accurate target value is reached. In this study, the target value is a residual of zero ( $\varepsilon_{i,q}^{p} = 0$ ), indicating a perfect model fit with input errors estimated exactly. Here, $k_{i,q}$ and $\varepsilon_{i,q}^{p}$ represents the estimated rank of input error and the model residual at $i$th time step and $q$th iteration respectively. The error rank of each data point is updated respectively via Eq.(4), where $i = 1,…n$. $n$ is the data length and also the number of the estimated errors as these errors are data-based.

After calculating Eq.(4) , it is possible that the rank $k_{i,q}$ is out of the rank range (for example, less than 1 or more than n), or not an integer. Sorting $k_{i,q}$ in all the ranks $k_{i,q} (i = 1,…,n)$ can address this problem by effectively assigning to each of them a new integer rank based on its position in the sorted list. Thus, in Eq.(4), $k_{i,q}$ should be changed to $K_{i,q}$ , representing the pre-rank. After sorting $K_{i,q}$ for all the errors, the post-rank $k_{i,q}$ will then belong to reasonable values. The specific calculation of the error rank is demonstrated in the 7th and 8th row in Table A 1.

From the above, estimating the rank of input errors via the secant method can be described as the following two equations:

Update the rank of each input error $K_{i,q}$ via the secant method respectively for $i = 1,…,n$ :

$$K_{i,q} = k_{i,q-1} - \varepsilon_{i,q-1}^{p} \frac{k_{i,q-1} - k_{i,q-2}}{\varepsilon_{i,q-1}^{p} - \varepsilon_{i,q-2}^{p}} \tag{5}$$

Sorting $K_{i,q}$ $(i = 1,…,n)$ in all the error pre-ranks $\boldsymbol{K}_q$ to obtain a reasonable rank:

$$k_{i,q} = k(K_{i,q}) \tag{6}$$

where $k(\ )$ means calculating its rank.

Thus, the procedure of input error quantification has been developed via the following key steps: 1) Sample the errors (the number is equal to the number of input data) from the assumed error distribution to maintain the overall statistical characteristics of the input errors and allocate them randomly to all the time steps; 2) Update the input error ranks to force the model residual close to zero via the secant method (Eq. (5) and (6)); 3) Reorder these sampled errors according to the updated error ranks; 4) Repeat 2) and 3) for a few iterations until a defined target is achieved. This new algorithm is referred to as Bayesian error analysis with reordering (BEAR). An example to illustrate how the BEAR method works is presented in Appendix A.

## 2.3 Bayesian inference of input uncertainty and the BEAR method

When the BEAR method is applied in a realistic situation, the model structural error and output observational error will be lumped into residual error, which is often assumed to follow a Gaussian distribution with mean 0 and variance $\sigma^2$, $\boldsymbol{\varepsilon}^p \sim (0, \sigma^2)$. According to the study of Renard et al. (2010), the posterior distribution of all inferred quantities in this study is given by Bayes' theorem, as follows:

$$
\begin{aligned}
&p(\boldsymbol{\theta}^p, \boldsymbol{\varepsilon}_X^p, \mu_X, \sigma_X, \sigma \,|\, \boldsymbol{Y}^o, \boldsymbol{X}^o) \propto \\
&p(\boldsymbol{Y}^o \,|\, \boldsymbol{\theta}^p, \boldsymbol{\varepsilon}_X^p, \sigma, \boldsymbol{X}^o)\, p(\boldsymbol{\varepsilon}_X^p \,|\, \mu_X, \sigma_X)\, p(\boldsymbol{\theta}^p, \mu_X, \sigma_X, \sigma)
\end{aligned}
\tag{7}
$$

The full posterior distribution comprises the following three parts: the likelihood of the observed output $p(\boldsymbol{Y}^o \,|\, \boldsymbol{\theta}^p, \boldsymbol{\varepsilon}_X^p, \sigma, \boldsymbol{X}^o)$, the hierarchical parts of the input multiplier $p(\boldsymbol{\varepsilon}_X^p \,|\, \mu_X, \sigma_X)$ and the prior distribution of deterministic parameters and hyperparameters $p(\boldsymbol{\theta}^p, \mu_X, \sigma_X, \sigma)$.

Unlike the formal Bayesian inference, the BEAR method does not update the posterior distribution of the input errors. Considering $\boldsymbol{\varepsilon}_X^p$ are sampled from $N(\mu_X, \sigma_X^2)$, $p(\boldsymbol{\varepsilon}_X \,|\, \mu_X, \sigma_X)$ is fixed when $\mu_X, \sigma_X$ are determined and do not need to be considered in Eq.(7). The secant method in the BEAR algorithm is applied to find the optimal ranks of input errors to minimise the model residual errors towards zero, as characterised by the minimized Residual Sum of Squares (RSS). Minimizing the RSS imposes the same effect as maximizing the likelihood function. The effectiveness of this step in quantifying the input errors is based on the assumption that the input error is dominant in the residual error and then minimizing RSS is the same as allocating the total error into the input errors. Otherwise, other dominant sources of errors will affect the estimation of the optimal input errors leading to poor input error identification.

According to Eq.(3), the residual error is related to the deterministic variables $X^o$ and $Y^o$, and the updated variables $\boldsymbol{\varepsilon}_X^p$ and $\boldsymbol{\theta}^p$. In other words, corresponding to the minimized RSS, the optimal ranks of input errors only changes with the model parameters $\boldsymbol{\theta}^p$. The distribution of input errors (i.e. their cumulative distribution function (CDF)) can be characterised by hyperparameters $\mu_X, \sigma_X$. Given the value of each input error is determined by the CDF of the whole input errors (depends on $\mu_X, \sigma_X$) and its relative rank among them (depends on $\boldsymbol{\theta}^p$), $\boldsymbol{\varepsilon}_X^p$ can be represented as a deterministic function of $\mu_X, \sigma_X$ and $\boldsymbol{\theta}^p$ ($\boldsymbol{\varepsilon}_X^p = f(\boldsymbol{\theta}^p, \mu_X, \sigma_X)$), and the likelihood of the observed output $p(Y^o | \boldsymbol{\theta}^p, \boldsymbol{\varepsilon}_X^p, \sigma, X^o)$ can be represented as $p(Y^o | \boldsymbol{\theta}^p, \mu_X, \sigma_X, \sigma, X^o)$. Therefore, the posterior distribution of all inferred parameters (Eq.(7)) in the BEAR method will turn into:

$$
\begin{aligned}
& p(\boldsymbol{\theta}^p, \boldsymbol{\varepsilon}_X^p, \mu_X, \sigma_X, \sigma | Y^o, X^o) \propto \\
& p(Y^o | \boldsymbol{\theta}^p, \mu_X, \sigma_X, \sigma, X^o) p(\boldsymbol{\theta}^p, \mu_X, \sigma_X, \sigma)
\end{aligned}
\tag{8}
$$

The above derivation states if the relationship between the input errors and model parameters can be determined, the problem of parameter estimation and input error identification (Eq.(7)) can then be interpreted as updating $\boldsymbol{\theta}^p, \mu_X, \sigma_X, \sigma$ in the Bayesian inference (Eq.(8)).

## 2.4 Integrating the BEAR method into the Sequential Monte Carlo approach

The core strategy of the BEAR method is to identify the input errors by estimating their ranks, which can be easily integrated into formal Bayesian inference schemes (for example, Markov chain Monte Carlo (MCMC, (Marshall et al., 2004)) and Sequential Monte Carlo (SMC, (Jeremiah et al., 2011, Del Moral et al., 2006))) and other calibration schemes (for example, the generalized likelihood uncertainty estimation (GLUE, (Beven and Binley, 1992))). Based on the traditional calibration approach, the BEAR method works by replacing the observed input with a modified input that is obtained through the estimated input error rank via the secant method. This study applies the SMC sampler for the model parameter updating. In the SMC approach, the model parameter is first sampled from a prior distribution and then propagated through a sequence of intermediate populations by repeatedly implementing the reweighting, mutation and resampling processes, until the desired posterior distribution is achieved (Del Moral et al., 2006). The details of the SMC algorithm can be found in the study of Jeremiah et al. (2011).

Figure 1 demonstrates the integration of the BEAR method into the SMC sampler. In the SMC scheme, $s$ refers to the number of sequential populations. A population means a group of parameter vectors (particles) that is updated in each iteration. The maximum number of the population $S$ is set as 200 in this study. In each sequential population, $N$ particles of model parameters are calibrated. $N$ is set as 100 in this study. For each particle of the model parameters, the corresponding input error ranks are

updated over $q$ iterations, where $q$ increases until the acceptance probability is larger than a number randomly sampled from 0 to 1. It should be noted that if the model parameters are far away from the true values, especially in the initial population, iterative updating of the error ranks will have little effect in reducing the model residual. Therefore, the maximum number of iterations should be set, referred to as $Q$. $Q$ is set as 20 in this study. If $q$ exceeds $Q$, the algorithm returns to the mutation step in Fig. 1.

## 2.5 Comparison with other methods

In this study, three methods, including the "Traditional" method, "IBUNE" method and "BEAR" method, are compared to evaluate the ability of the BEAR method in estimating the model parameters and quantifying input errors. The "Traditional" method regards the observed input as error-free without identifying input errors (i.e. Eq. (2)), while the other two methods employ a latent variable to counteract the impact of input error and derive a modified input (i.e. Eq.(3)). In the "IBUNE" method, potential input errors are randomly sampled from the assumed error distribution and filtered by the maximization of the likelihood function (Ajami et al., 2007). Although the comprehensive IBUNE framework additionally deals with model structural uncertainty via Bayesian Model Averaging (BMA), this study only compares the capacity of its input error identification. The "BEAR" method adds a reordering process into the "IBUNE" method to improve the accuracy of input error quantification.

## 3 Case studies

### 3.1 Water quality model: the build-up/wash-off model (BwMod)

This study tests the BEAR algorithm in the context of the build-up/wash-off model (BwMod), which is a group of models to simulate two processes in sediment dynamics, including the build-up of sediments during dry periods and the wash-off process during wet periods. The two formulations were developed in a small-scale experiment (Sartor and Boyd, 1972), while in applications at the catchment scale, the conceptualized parameters largely abandon their physical meanings and the formulations can be considered a "black-box" (Bonhomme and Petrucci, 2017). This study chooses Eq. (9) to describe the build-up process and Eq. (10) to express the wash-off of sediments, representing the non-linear relationship between the wash-off load (output) and the runoff-rate (input). These two equations were applied in the research of Sikorska et al. (2015) and in this study are integrated with the BEAR method. This study will test the BEAR algorithm in a case of simulating the daily sediment dynamics of one catchment, thus, we use daily time steps and consider the catchment a single, homogeneous spatial unit. This version of BwMod has four parameters (Table 1). The model input is streamflow, which typically comes from the observation of a rating curve. As discussed in the introduction, the error distribution can be estimated prior to the model calibration via a rating curve analysis. The output of the BwMod is the concentration of total suspended solids (TSS), whose transport can be efficiently simulated by the conceptualization of the build-up/wash-off process (Bonhomme and Petrucci,

2017, Sikorska et al., 2015). Although BwMod is relatively simple compared with process-based WQMs, its nonlinearity and the use of surrogates for the input data can make it a typical WQM scenario to test the BEAR algorithm.

The overall BwMod equations are:

$$\frac{dS_{a,t}}{dt} = \kappa \cdot \left( S_{max} - S_{a,t} \right) - s\left( S_{a,t} \right) \tag{9}$$

where the descriptions of $\kappa$ and $Smax$ are shown in Table 1, $S_{a,t}$ (kg) is the sediment amount available on the catchment surface to be washed-off at time $t$; $s\left( S_{a,t} \right)$ (kg/s) is the amount of sediment in the stream at time $t$, described by the function

$$s(S_{a,t}) = a \cdot (Q_t)^b \cdot S_{a,t} \tag{10}$$

where the descriptions of $a$ and $b$ are shown in Table 1, and $Q_t$ is the streamflow at the catchment outlet at time $t$.

The output TSS concentration $C_{TSS,t}$ (kg/m$^3$) is derived via:

$$C_{TSS,t} = \frac{s\left( S_{a,t} \right)}{Q_t} \tag{11}$$

**3.2 Setup of synthetic case study**

To test the capability of the secant method in identifying the input error ranks in the process of the model parameter estimation,
the BEAR method is first implemented in a controlled situation with synthetic data. The true input $X^*$ is set as the daily streamflow data of the catchment in the real case (USGS ID: 04087030), covering 1095 days from 2009/10/01 to 2012/09/29. The true output $Y^*$ is the simulated TSS concentration via BwMod corresponding to the true input $X^*$ and model parameters set as the reference values in Table 1. In Case study 1 where the model is affected only by input errors and parameter error, the observed output $Y^o$ is assumed to be the same as the true simulation $Y^*$, i.e. without error. The observed input $X^o$ is
generated based on two types of input error models: an additive formulation and a multiplicative formulation, and the errors are assumed to follow a normal distribution with mean $\mu$ as 0.2 and standard deviation (SD) $\sigma$ as 0.5. If the input errors are estimated based on a rating curve, like the procedure in the following real case, the mean of input error should be 0. But in order to test the ability of the BEAR method in wider applications, a systematic bias 0.2 has been considered in the synthetic case study even though this is unlikely to manifest in real situations. An additive formulation (denoted as '*add*' in Table 2) is
suitable to illustrate the error generation in measurements, while the multiplicative formulation (denoted as '*mul*' in Table 2) is specifically applied for errors induced from a log-log regression procedure, which is common for water quality proxy

processes (Rode and Suhr, 2007). In the additive formulation, the generated input may be negative. If so, the negative input should be truncated to a positive value. In the multiplicative formulation, the generated input will stay positive. Given the description in the introduction, the input error model can be pre-estimated independent of calibration by analysing the input data in some studies. While in other cases, the input error model cannot be estimated or its accuracy is in question. Therefore, two scenarios about the prior information of $\sigma$ have been considered: one is fixed as the reference values (denoted as '*fixed*' in Table 2), the other one is estimated as the hyperparameters with the model parameters (denoted as '*inferred*' in Table 2). Therefore, Case study 1 considers four scenarios, including two sets of input data generating from two input error models and two types of prior information about the error parameter $\sigma$ (the details are shown in Table 2).

Case study 1 is an ideal situation that the model calibration only suffers from the input errors and parameter errors. However, in real-life cases, other sources of errors (i.e. model structural error and output data error) will impact this effectiveness. To explore the ability of the BEAR method with the interference of other sources of errors, the output observational errors with the increasing standard deviations are considered to build the synthetic data based on the scenario 3 and 4 in Case study 1 (the details has been shown in Table 2).

To sum up, two synthetic case studies have been analysed: Case study 1 generates synthetic data only suffering from input errors to evaluate the effectiveness of the BEAR method in isolating the input error and the model parameter error; Case study 2 additionally considers output observation errors via synthetic data generation to evaluate the impacts of other sources of error on the BEAR method. Each scenario in the synthetic case studies is calibrated via the Traditional method, the IBUNE method and the BEAR method respectively. Their algorithms are described in Sect. 2.5. Considering the unknown initial sediment loads in real applications, the calibration sets 90 days as a warm-up period to remove the influence of antecedent conditions.

### 3.3 Setup of real case study

To explore the ability of the BEAR method in real-life applications, a real case of one catchment located in southeast Wisconsin, USA is demonstrated. Table 3 is a description of the test catchment and data (Baldwin et al., 2013). The daily TSS concentration and streamflow data are collected from the USGS database on National Real-Time Water Quality (https://nrtwq.usgs.gov/). The daily streamflow data in the USGS database comes from a stage-streamflow rating curve, where the stage and streamflow form a log-log linear relationship and the streamflow proxy errors follow a normal distribution with $\mu$ as 0 and $\sigma$ as 0.103. This prior information is used in the real calibration, denoted as *O-fixed* scenario in Table 2, where "*O*" represents the input data that comes from the observations of the rating curve. According to the results of Figure 3 and the assumption of the methodology derivation, the BEAR method works better when the input uncertainty is more significant, so another input data source with more significant data uncertainty, a streamflow simulation from a hydrological model, has been considered. This study selects GR4J (Perrin et al., 2003) as the hydrological model and calibrates its parameters with the USGS streamflow data as calibration data. If the USGS streamflow data is regarded as the true input data, the residual error after the model calibration can approximate the data error of GR4J simulation, which follows a normal distribution in log space with

$\mu$ as 0 and $\sigma$ as 0.764. The BwMod calibration using this input data source and the prior information on data error is denoted
as *S-fixed* scenario in Table 2, where "*S*" represents the input data that comes from the simulations of GR4J model. To explore
the ability of the BEAR method in other situations where the prior information about the input error is not sufficient, two
scenarios with a wider range of the error parameters has also been considered, denoted as *O-inferred* and *S-inferred* in Table
2. The real case is also calibrated via three methods (i.e. the Traditional method, the IBUNE method and the BEAR method)
and adopts the same setting of the calibration algorithm as the synthetic case study.

## 4 Results

### 4.1 Case study 1: Synthetic data suffering from input errors

To evaluate the ability of different calibration methods (i.e. Traditional method, the IBUNE method and the BEAR method)
in identifying the input error and model parameter, the following statistical characteristics are selected to compare the results
of case study 1, which only suffers from input error and parameter errors. The SD of the estimated input errors represents the
accuracy of the input error distribution (0.5 is the reference value). The correlation between the estimated input error and the
true input error evaluates the capability of the method in catching the temporal dynamics of input error. The Nash-Sutcliffe
efficiency (NSE) of the modified input vs true input measures the precision of the input data after removing the estimated input
errors. In the calibration part, the simulated output corresponds to the modified input and estimated model parameters, and its
NSE compared to the true output measures the goodness-of-fit. In the validation part, the simulated output corresponds to the
true input and estimated model parameters, and its NSE compared to the true output can assess the accuracy of the model
parameter estimation. These statistical characteristics are calculated as the weighted-average values considering the weights
of each estimation in the posterior distribution and compared in Fig. 2. Figure B 1 in Appendix B demonstrates the temporal
dynamics of input estimations and model simulations of synthetic case 1. In Fig. B1, "Reliability" is the ratio of observations
caught by the confidence interval of 2.5%-97.5%, and the average width of this interval band is referred to as "Sharpness"
(Yadav et al., 2007, Smith et al., 2010).

Evaluating the model simulation, the BEAR method always produces the best output fit in all scenarios, supported by the
highest green bars in Fig. 2(4). Although its correlations with the true error series are much higher than the IBUNE method
(red bars) in all scenarios (in Fig. 2(2)), the BEAR method cannot ensure a better input estimation (in Fig. 2(3)) and its ability
depends on the prior information of the input error parameter. When the error parameters are fixed at the reference values (in
the scenarios *add-fixed* and *mul-fixed*), the BEAR method always outperforms the other two methods in the input modification
and model parameter estimation, as its NSE is the highest (green bars in Fig. 2(3) and (5)). Without the reordering strategy,
the IBUNE method even gives worse input modification, model simulation and parameter estimation than the Traditional
method, demonstrated by the lower red bars than blue bars in Fig. 2(3), (4) and (5). When the error parameters are inferred (in
the scenarios of *add-inferred* and *mul-inferred*), the IBUNE method can improve the input data and the model parameter

estimation compared with the Traditional method (in Fig. 2(3) and (5)) although the estimations of $\sigma$ via the IBUNE method are always smaller than the reference value (in Fig. 2(1)). This result has also been reported in the study of Renard et al. (2009), which indicates that the randomness of the likelihood function leads to an underestimation of $\sigma$ of input errors. Unlike the IBUNE method, the performance of the BEAR method depends on the setting of the input error model. In the *add-inferred* scenario, the BEAR method is still better than other methods, having a bigger NSE (in Fig. 2(3), (4) and (5)) and the closer $\sigma$

estimation to reference value (in Fig. 2(1)). While in the *mul-inferred* scenario, the modified inputs and estimated parameters via the BEAR method are worse than the IBUNE method (in Fig. 2(3) and (5)).

### 4.2 Case study 2: Synthetic data suffering from input errors and output observation errors

Nash-Sutcliffe efficiency(NSE) is selected to measure the difference between the modified input in Case study 2 and the true input. Figure 3 demonstrates in the *mul-fixed* scenario where the prior information of standard deviation of input errors is

320 accurate, the BEAR method always brings a better input modification than other methods, although its ability is impaired by the impact of the output observational errors as the NSEs reduces with the increasing SD of the output observational error. The IBUNE method leads to an even worse modified input than the input data without modification in the Traditional method. In the *mul-inferred* scenario where the standard deviation of input errors cannot be pre-estimated accurately and given in a wide range, the BEAR method brings worse input data while the IBUNE method can modify the input data.

### 4.3 Case study 3: Real data

Figure 4 compares the SDs of estimated input errors, the variances of model residual errors, and reliability and sharpness of model simulations among the four calibration scenarios and three calibration methods in the real case study. Figure 4(2) demonstrates the BEAR method always produces a better fit to the output data than the IBUNE method, consistent with the synthetic case shown in Fig. 2(4). In Fig. 4(3), except for the *O-fixed* scenario, the results of the BEAR method (in green) show

much smaller sharpness than the Traditional method (in blue) and the IBUNE method (in red) with almost the same reliability. According to the results of the Traditional method in Fig. B2, the simulations from the "*O*" streamflow (in (a1)) catch the dynamics of observed TSS concentration better than the simulations from the "*S*" streamflow (in (a3)). Thus, compared with the simulated streamflow via GR4J ("*S*" streamflow), the observed streamflow from the rating curve ("*O*" streamflow) should be closer to the true input data. In Fig. B2, the modified inputs via the BEAR method are closer to the "*O*" streamflow (blue

dots) than the "*S*" streamflow (pink dots), even in (c3) and (c4) where the original input data comes from the "*S*" streamflow. However, the modified input via the IBUNE method is always centred on the original input data it uses. Given being always closer to the "*O*" streamflow, the modified inputs via the BEAR method are more reasonable than the IBUNE method.

## 5 Discussion

### 5.1 The effectiveness of rank estimation

The novelty of the BEAR method lies in transforming a direct error value estimation to an error rank estimation. In a continuous sequence of data, the potential error values have an infinite number of combinations, while the error rank has limited combinations, dependent on the data length. For example, in Table A1, the estimated error at the 1[st] time step could be any value. Even under a constrain of the range from the minimized to the maximized sampled errors (i.e. [-0.29,0.16] in the 1[st] iteration), its value estimation still has infinite possibilities due to the continuous nature of the error. In contrast, the rank is

discrete, having only 20 possibilities (i.e. the integrity in [1, 20]). From this point of view, it is more efficient to estimate the error rank than estimate the error value.

However, the rank estimation will suffer from the sampling bias problem. For the same error distribution and the same cumulative probability distribution (corresponding to the same error rank), the errors in different samplings could be largely different, especially for a small sample size (depending on the data length) or a large $\sigma$ of the assumed error distribution. This

problem can be addressed by selecting the optimal solution from multiple samples according to the maximum likelihood function. In three case studies, the sample size is larger than 1000, where the sampling bias problem can be neglected and one error sampling is enough. But in some cases where the sample size is small (i.e. around 10), multiple samplings should be undertaken.

In addition, the rank estimation can make better use of the knowledge of the input error distribution. In a direct value estimation,

it is difficult to keep the overall error distribution constant when the errors are updated in the calibration. The estimated errors are more likely to compensate for other sources of errors to maximize the likelihood function and subsequently be overfitted. By contrast, in rank estimation, the errors at all the time steps are sampled from the pre-estimated error distribution first and then reordered. Whatever the error rank estimates are, they always follow the pre-estimated error distribution, and the compensation effect will be limited. In the IBUNE framework (Ajami et al., 2007), the errors are also sampled from the error

distribution, but not reordered. In the BEAR method, adjusting the sampled errors according to the inferred error rank reduces the randomness of the error allocation in the IBUNE framework (Ajami et al., 2007), which significantly improves the accuracy of the error estimation (as demonstrated by much higher correlations than the IBUNE method in Fig. 2(2)).

Unlike formal Bayesian inference, the rank estimation does not update the posterior distribution of the input errors, but optimises their time-varying values through the relationship between the input error rank and corresponding model residual

error. The rank estimation is implemented after the model parameters have been updated and the model residual error depends on the input error estimation. Thus, the reordering strategy identifies the optimal input error rank conditional to the model parameters, effectively considering the interaction between the input error and the parameter error. This is akin to calibrating the input errors along with the model parameters in the BATEA framework (Kavetski et al., 2006).

## 5.2 The effect of reordering on the error realization

Figure 5 demonstrates the mechanics of input error reordering in the BEAR method and input error filtering in the IBUNE method to understand their effects on the input error realizations and model parameter estimation. The 1$^{st}$ sequence represents the situation where the raw input errors are randomly sampled from the pre-estimated error distribution, therefore, their marginal means and standard deviations are the same as the parameters of overall error distribution (demonstrated as the cyan lines in column (c)). In the later sequence. these errors are optimised via different methods. In the IBUNE method, these

sampled input error series are selected by the maximized likelihood function and the interval of input errors become a little converged (in (b1)) and their marginal standard deviations reduce slightly (in (c1)). However, in the BEAR method, these input errors are reordered according to the inferred ranks via the secant method, and the reordered errors gradually converge to the true values (represented by the blue interval are near the red line in (b2)). Therefore, their marginal means are similar to the true values and their marginal standard deviations reduce to zero (in (c2)). In the BEAR method, the promotion of the input

error identification in the sequential updating will improve the model parameter estimation, represented by the posterior distribution of model parameter $b$ converging to the true value in (a2). While in the IBUNE method, the identification of input errors is not precise and the bias of the model parameter still exists in (a1).

The data length can affect the efficacy of the BEAR method but impose little effect on the IBUNE method. The IBUNE method takes advantage of the stochastic errors and keeps the marginal error distribution almost constant. The input error realization

at each time step seems independent, only filtered by the overall likelihood function. Therefore, the number of sampled errors does not matter in the IBUNE method. However, in the BEAR method, the input errors at all the time steps are not sampled independently, they are from one sample set. Therefore, before or after reordering, all errors will keep the same statistical features of the input error distribution, and only their marginal distribution changes due to the convergence to the unknown true values. Figure 5 (b2) demonstrates that when the data length (the same as the error number) is small, the input error

estimation might be biased from the true values. This likely arises from the above-mentioned sampling bias or the impacts of the model parameter error because the sampling bias reduces with the larger number of error samples and the impacts of parameter error are more likely to be offset when the data length is longer.

## 5.3 The impacts of prior information of input error model

The IBUNE method takes advantage of stochastic error samples to modify the input observations (Ajami et al., 2007). In the

395 real case study, *S-fixed* and *S-inferred* scenarios use simulated streamflow as input data, whose input error is more significant than the observed streamflow used in *O-fixed* and *O-inferred* scenarios. Figure. B5(b) demonstrates that the resultant simulations (black line) via the IBUNE method in *S-fixed* and *S-inferred* scenarios are further away from the observed outputs (red dots) than the simulations in *O-fixed* and *O-inferred* scenarios. What's more, in the synthetic case, Fig. 2(1) shows standard deviations of input errors in *fixed* scenarios are larger than those in *inferred* scenarios, which means the *fixed* scenarios have

400 more significant input errors. Fig. 2(3) demonstrates that the modified inputs in *fixed* scenarios are worse than those in *inferred*

scenarios although the standard deviations of input errors in *fixed* scenarios are set as the true value. From the above, the availability of prior information is insignificant for the IBUNE method, and the modifications of the input data and model simulation via the IBUNE method only happens when the $\sigma$ of the estimated input error is small. It is most likely to make use of the stochastic errors to approach the true input data, but not effectively identify the input error.

However, the findings in the BEAR method are quite different. Accurate prior information about the input error model is important in the BEAR method. Figure 3 demonstrates *fixed* scenarios calibrated via the BEAR method always produce a higher NSE of the modified input than *inferred* scenarios. This is likely because the prior information can constrain the input error distribution and reduce the impacts of other sources of errors. The availability of prior information of the input error relies on studies about benchmarking observational errors of water quality and hydrologic data, and the selection of a proper

input error model is important. Comparing the results in Fig. 2, when the input error model is an additive formulation, the BEAR method consistently brings the best performance regardless of the prior information of the error $\sigma$. When the input error model is a multiplicative formulation, the BEAR method cannot improve the input data if the prior information of the error $\sigma$ is not accurate. This illustrates that the compensating effect between the input error and parameter error is weaker in the additive form of the input error. This is probably related to the specific model structure, as the exponent parameter $b$ in

BwMod has a stronger interaction with the multiplicative errors than the additive errors. Thus, more comprehensive comparisons should be undertaken to explore the capacity of different input error models in different model applications.

## 6 Conclusion and recommendation

Observation uncertainty in input data is independent of the model process and the input error model can be estimated prior to the model calibration and simulation by analysis of the data itself. Taking advantage of the prior information of an input error

model, a new method, Bayesian error analysis with reordering (BEAR), is proposed to approach the time-varying input errors in WQM inference. It contains two main processes: sampling the errors from the assumed input error distribution and reordering them with the inferred ranks via the secant method. This approach is demonstrated in the case of TSS simulation via a conceptual water quality model, BwMod. Through the investigation of synthetic data and real data, the main findings are as follows:

(1) The estimation of the BEAR method focuses on the error rank rather than the error value in the existing methods, which can take advantage of the constraints of the known overall error distribution and then improve the precision of the input error estimation by optimising the error allocation in a time series.

(2) The introduction of the secant method addresses the nonlinearity in the WQM transformation and can effectively update the error rank of each input data according to minimizing its corresponding model residual.

(3) The ability of the BEAR method in decomposing the input error from model residual error is limited by the accuracy and selection of the input error model and is impacted by model structural uncertainty and output observation uncertainty.

Therefore, the study identifies several areas which need further analysis. Firstly, the availability of prior knowledge of the input error model is important. When this information is not reliable or even cannot be estimated, a significant issue is the selection of a suitable error distribution. Thus, a general measure should be found to judge whether an error model is

435 appropriate, especially in real cases where the "true" information is limited. Secondly, this study focuses on identifying the input errors in model calibration and the derivation of the BEAR method is based on the assumption that the input error is dominant in the residual error. If the reordering strategy is developed within a more comprehensive framework to quantify multiple sources of error, this assumption will be relaxed, the interactions amongst these error sources might be well-identified and the quantification of individual errors might be improved. This study provides a starting point for developing the rank

estimation via the secant method to identify input error. Further study is necessary to modify the algorithm and improve confidence in extended case studies or model scenarios.

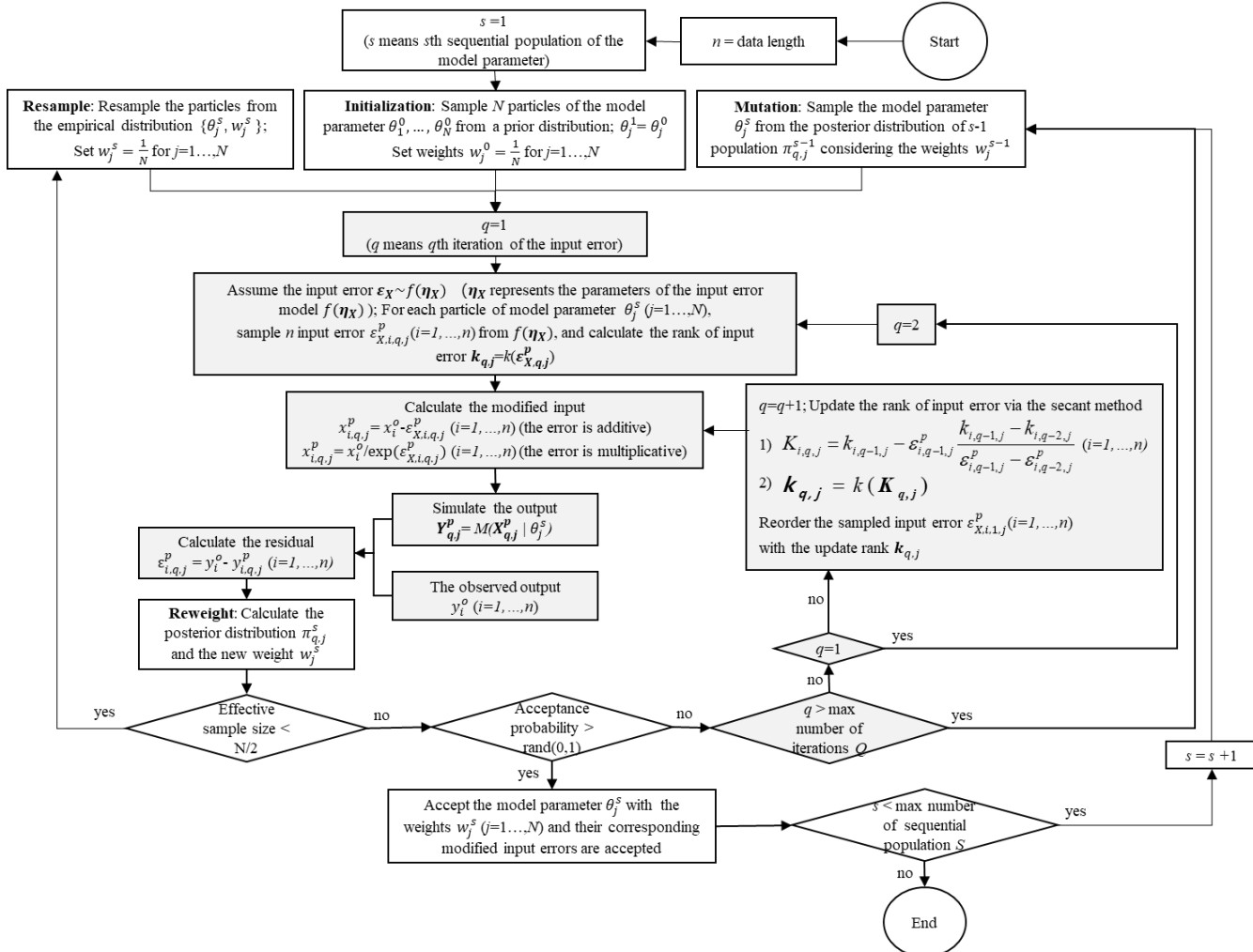

**Figure 1 Flowchart of the algorithm to quantify the input errors via Bayesian error analysis with reordering (BEAR) method in the SMC calibration scheme (The grey charts demonstrate the BEAR method while the white charts demonstrate the SMC algorithm. The details of the BEAR method can refer to Appendix A. The details of the SMC algorithm can refer to the study of Jeremiah et al. (2011), including the Mutation step, the Reweight step and calculating the acceptance probability. rand(0,1) means a number randomly sampled from 0 to 1.)**

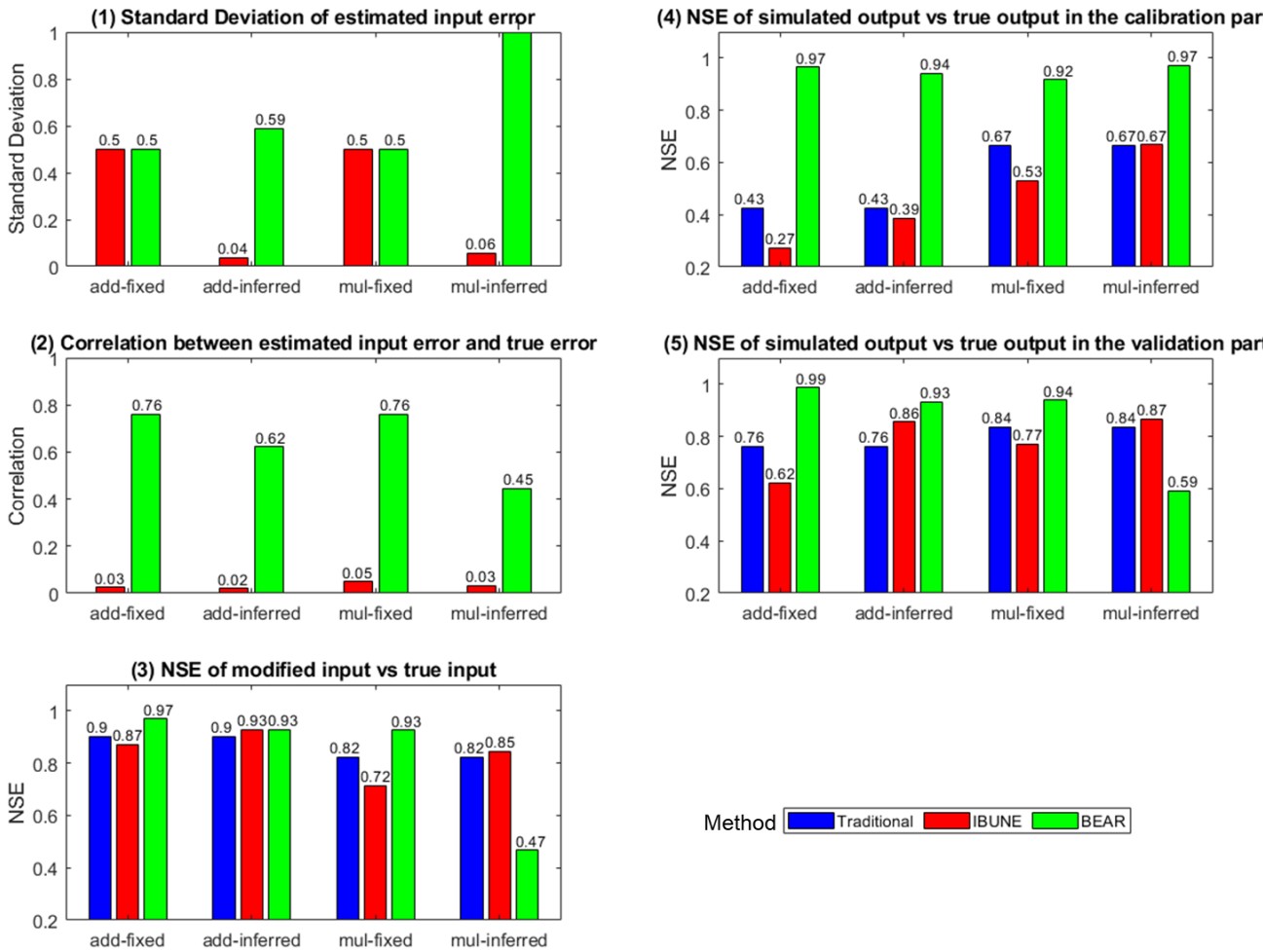

**Figure 2: Comparison of statistical characteristics of four calibration scenarios in the synthetic case 1 (including *add-fixed*, *add-inferred*, *mul-fixed* and *mul-inferred*; notations are given in Table 2) via three calibration methods (including the Traditional method, the IBUNE method and the BEAR method, their algorithms are explained in Sect. 2.4)**

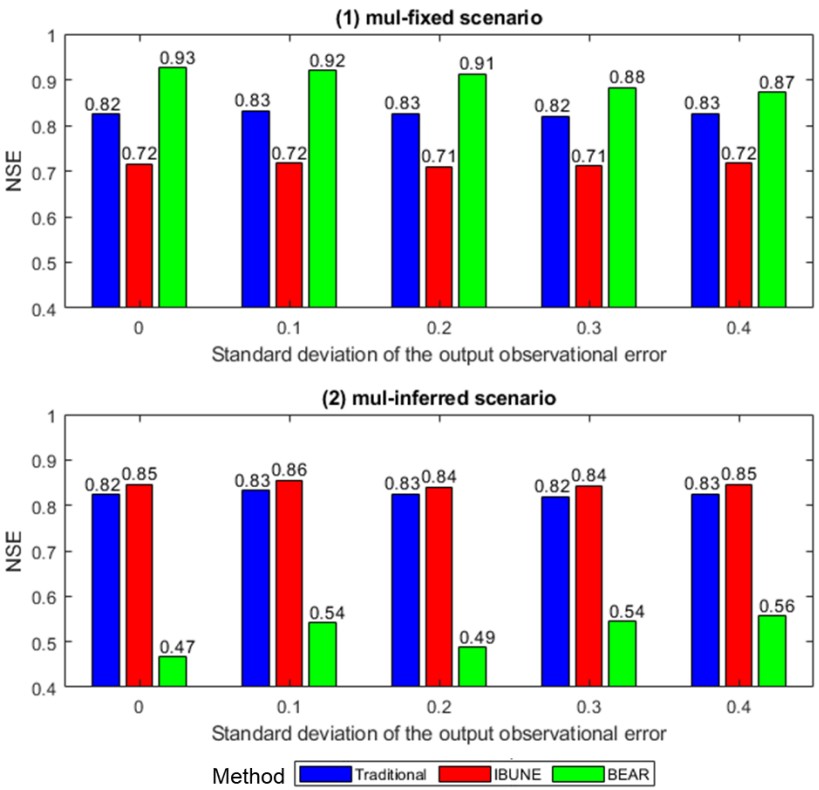

**Figure 3 Comparison of Nash-Sutcliffe efficiency (NSE) of the modified input v.s true input under the interference of the output observational errors with the increasing standard deviations in two calibration scenarios in the synthetic case 2 (including *mul-fixed* and *mul-inferred*; notations are given in Table 2) via three calibration methods (including the Traditional method, the IBUNE method and the BEAR method, their algorithms are explained in Sect. 2.4)**

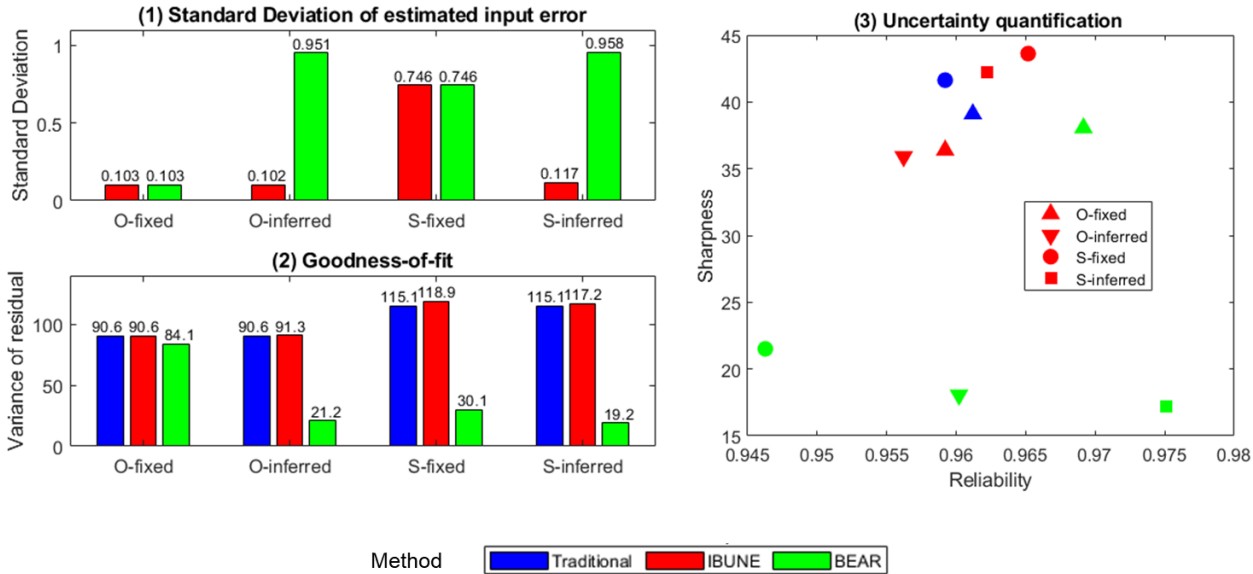

**Figure 4: Comparison of statistical characteristics of four calibration scenarios in the real case (including *O-fixed*, *O-inferred*, *S-fixed* and *S-inferred*, their notations are given in Table 2) via three calibration methods (including the Traditional method, the IBUNE method and the BEAR method, their algorithms are explained in Sect. 2.4)**

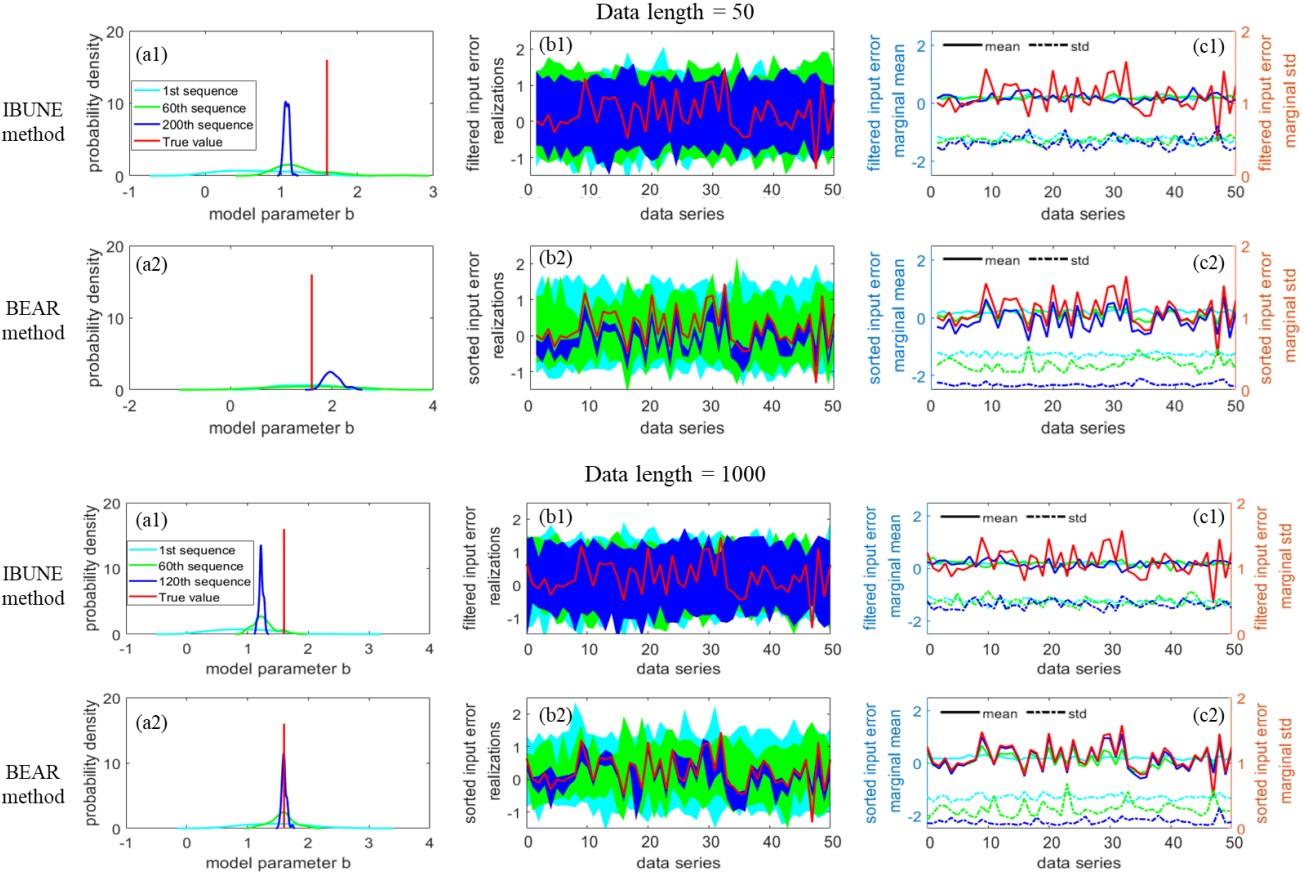

**Figure 5 Comparision of the results for scenario 1 in the synthetic case 1, the ensemble particle size of *N*=100 at different sequences of calibration (represented in different colours), via different methods (row 1: IBUNE method; row 2: BEAR method) and under different data lengths in calibration (the upper group: data length is 50; the lower group: data length is 1000, selects 50 data the same as the upper group to show). Column a shows the probability density of model parameter *b* at different sequences of calibration. The other model parameters have the same pattern of change and thus there is no need to show. Column b shows the value interval of input error realizations of 100 particles after reordering in the BEAR method or filtering in the IBUNE method and Column c shows the corresponding marginal mean and standard deviations at each time step. The 1ˢᵗ sequence (in cyan) shows the raw input errors of random sampling, before reordering or filtering.**

**Table 1 Descriptions of BwMod parameters**

| Model | Parameter | Description | Unit | Reference value in the synthetic case | Prior range in the case study |
|---|---|---|---|---|---|
| | $a$ | wash-off coefficient | - | 0.04 | (0, 2) |
| | $b$ | wash-off exponent | - | 1.6 | (0, 3) |
| BwMod | $\kappa$ | sediment accumulate rate | - | 0.1 | (0, 1) |
| | $Smax$ | maximum amount of sediment possible to be accumulated | kg | 7000 | (0, 15000) |

**Table 2 Summary of the calibration scenarios in case studies**

| Scenario in the synthetic case 1 | Notation | Input error model in the synthetic input generation | Prior information of input error model in calibration |
|---|---|---|---|
| 1 | *add-fixed* | $X^o = X^* + \varepsilon, \varepsilon \sim N(0.2, 0.5^2)$ | $X^o = X^* + \varepsilon, \varepsilon \sim N(0.2, 0.5^2)$ |
| 2 | *add-inferred* | | $X^o = X^* + \varepsilon, \varepsilon \sim N(\mu, \sigma^2), \mu=0.2, \sigma \in (0,1)$ |
| 3 | *mul-fixed* | $X^o = X^* \exp(\varepsilon), \varepsilon \sim N(0.2, 0.5^2)$ | $X^o = X^* \exp(\varepsilon), \varepsilon \sim N(0.2, 0.5^2)$ |
| 4 | *mul-inferred* | | $X^o = X^* \exp(\varepsilon), \varepsilon \sim N(\mu, \sigma^2), \mu=0.2, \sigma \in (0,1)$ |

| Scenario in the synthetic case 2 | Notation | Observational error model in the synthetic output generation | Prior information of input error model in calibration |
|---|---|---|---|
| 1 | *mul-fixed* | $Y^o = Y^* \exp(\varepsilon), \varepsilon \sim N(0, \sigma_Y^2)$ | $X^o = X^* \exp(\varepsilon), \varepsilon \sim N(0.2, 0.5^2)$ |
| 2 | *mul-inferred* | $\sigma_Y^2 = 0, 0.1, 0.2, 0.3, 0.4$ | $X^o = X^* \exp(\varepsilon), \varepsilon \sim N(\mu, \sigma^2), \mu=0.2, \sigma \in (0,1)$ |

| Scenario in the real case | Notation | Input data source in the real case | Prior information of input error model in calibration |
|---|---|---|---|
| 1 | *O-fixed* | Observations from the rating curve (USGS database) | $X^o = X^* \exp(\varepsilon), \varepsilon \sim N(0, \sigma^2), \sigma=0.103$ |
| 2 | *O-inferred* | | $X^o = X^* \exp(\varepsilon), \varepsilon \sim N(0, \sigma^2), \sigma \in (0,1)$ |
| 3 | *S-fixed* | Simulations from a hydrological model | $X^o = X^* \exp(\varepsilon), \varepsilon \sim N(0, \sigma^2), \sigma=0.764$ |
| 4 | *S-inferred* | | $X^o = X^* \exp(\varepsilon), \varepsilon \sim N(0, \sigma^2), \sigma \in (0,1)$ |

**Table 3 Characteristics of the study catchments and calibration data**

| USGS station number | location | State | Drainage area (km$^2$) |
|---|---|---|---|
| 04087030 | Menomonee River at Menomonee Fall | Wisconsin, USA | 89.83 |

| | land use | | | |
|---|---|---|---|---|
| Urban (percent) | Agricultural (percent) | Natural (percent) | Period of Data | Number of Data (days) |
| 35 | 38 | 27 | 2009/10/01 - 2012/09/29 | 1095 |

**Table A 1 An example illustrating the BEAR method**

| row | time step | 1 | 2 | 3 | 4 | 5 | 6 | 7 | 8 | 9 | 10 | 11 | 12 | 13 | 14 | 15 | 16 | 17 | 18 | 19 | 20 |
|---|---|---|---|---|---|---|---|---|---|---|---|---|---|---|---|---|---|---|---|---|---|
| | | | | | | | 1st iteration (the input errors are randomly sampled) | | | | | | | | | | | | | | |
| 1 | sampled input error | 0.07 | -0.12 | 0.07 | 0.16 | 0.05 | .0.07 | 0.07 | -0.03 | 0.03 | -0.08 | 0.09 | -0.11 | -0.11 | -0.08 | -0.29 | 0.14 | 0.03 | -0.08 | 0.14 | -0.17 |
| 2 | input error rank | 13 | 3 | 14 | 20 | 12 | 17 | 15 | 9 | 10 | 7 | 16 | 4 | 5 | 6 | 1 | 19 | 11 | 8 | 18 | 2 |
| 3 | model residual error | -0.29 | 0.49 | -0.58 | -0.98 | -0.78 | 0.29 | -0.66 | 0.59 | -1.31 | -0.31 | -0.87 | 0.76 | 0.46 | 0.54 | 0.25 | -0.80 | -0.07 | 0.56 | -0.23 | 0.40 |
| | MSE | | | | | | | | | | 0.40 | | | | | | | | | | |
| | | | | | | | 2nd iteration (the input errors are randomly sampled) | | | | | | | | | | | | | | |
| 4 | sampled input error | -0.01 | -0.02 | 0.03 | 0.03 | -0.09 | 0.00 | -0.02 | 0.06 | 0.11 | 0.11 | -0.09 | 0.01 | -0.12 | -0.11 | 0.00 | 0.15 | -0.08 | 0.04 | -0.02 | 0.11 |
| 5 | input error rank | 9 | 6 | 14 | 13 | 3 | 10 | 8 | 16 | 17 | 18 | 4 | 12 | 1 | 2 | 11 | 20 | 5 | 15 | 7 | 19 |
| 6 | model residual error | -0.13 | 0.23 | -0.43 | -0.41 | -0.21 | 0.70 | -0.23 | 0.09 | -1.88 | -1.52 | 0.20 | 0.17 | 0.53 | 0.60 | -0.43 | -0.72 | 0.36 | 0.12 | 0.47 | -0.82 |
| | MSE | | | | | | | | | | 0.47 | | | | | | | | | | |
| | | | | | | | 3rd iteration (the error ranks are updated via the secant method) | | | | | | | | | | | | | | |
| 7 | calculated pre-rank | 5.8 | 8.7 | 14.0 | 8.0 | -0.3 | 22.0 | 4.3 | 17.3 | -6.1 | 4.2 | 6.2 | 14.3 | 31.3 | 42.0 | 4.7 | 29.0 | 10.0 | 16.9 | 14.4 | 7.6 |
| 8 | ranked rank (post rank) | 6 | 10 | 12 | 9 | 2 | 17 | 4 | 16 | 1 | 3 | 7 | 14 | 19 | 20 | 5 | 18 | 11 | 15 | 13 | 8 |
| | | | | | | | 3rd iteration (the input errors are reordered with the updated error ranks) | | | | | | | | | | | | | | |
| 9 | reordered input error | -0.02 | 0.00 | 0.01 | -0.01 | -0.11 | 0.11 | -0.09 | 0.06 | -0.12 | -0.09 | -0.02 | 0.03 | 0.11 | 0.15 | -0.08 | 0.11 | 0.00 | 0.04 | 0.03 | -0.02 |
| 10 | model residual error | -0.23 | 0.20 | -0.34 | -0.24 | -0.12 | 0.19 | 0.14 | 0.08 | -0.40 | -0.31 | -0.22 | 0.03 | -0.17 | 0.26 | -0.09 | -0.55 | 0.11 | 0.14 | 0.27 | -0.23 |
| 11 | MSE | | | | | | | | | | 0.06 | | | | | | | | | | |

**(a) at the 1st and 2nd iteration where the input errors are randomly sampled**

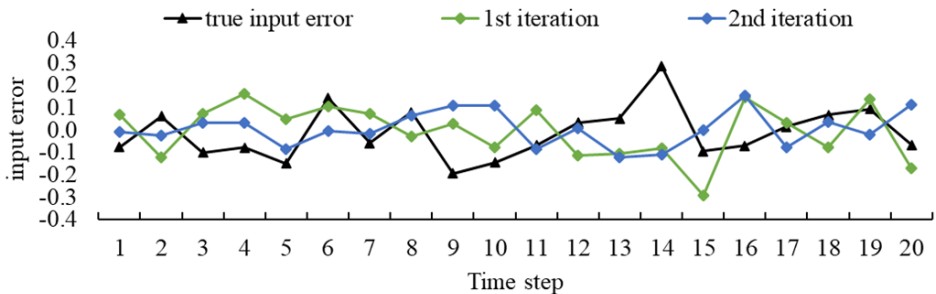

**(b) at the 3rd iteration where the input errors are reordered according to the updated error ranks**

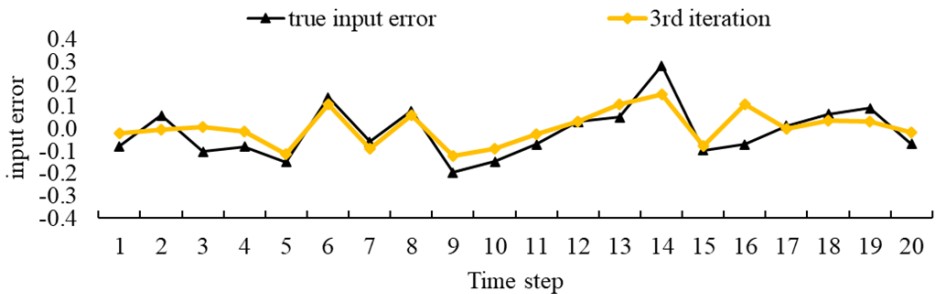

**Figure A 1 Demonstration of the input error estimated in Table A.1**

The BEAR method for identifying the input errors is implemented after generating the model parameters and contains two main parts: sampling the errors from an assumed error distribution and reordering them with the inferred ranks via the secant method. An example is illustrated in Table A 1 and the explanation about the specific steps is presented in the following contents.

(1) In the 1st iteration ($q=1$), the errors are randomly sampled from the assumed error distribution (row 1), and then are sorted to get their ranks (row 2). This error series is employed to modify the input data, which leads to a new model simulation and model residual (row 3).

(2) Repeat step (1) in the 2nd iteration ($q=2$) as two sets of samples are prerequisites for the updating via the secant method. The results are shown in row 4, 5 and 6. Figure A 1(a) demonstrates that the ranges of the error distribution are the same between the true input errors (black line) and the sampled errors (blue and green lines) as they come from the same error distribution under the condition that prior knowledge of the input error distribution is correct. However, the values at each time step cannot match due to the randomness of the sampling.

(3) At the 1st time step in the 3rd iteration ($i=1$, $q=3$ in Eq. (4)), the pre-rank $K_{1,3}$ is calculated via the secant method (illustrated as the following equation). The details are demonstrated in solid boxes in Table A.1.

$$K_{1,3} = k_{1,2} - \varepsilon_{1,2}^{p} \frac{k_{1,2} - k_{1,1}}{\varepsilon_{1,2}^{p} - \varepsilon_{1,1}^{p}} = 9 - (-0.13)\frac{9-13}{-0.13-(-0.29)} = 5.8$$

(4) Repeat step (3) for all the time steps. The calculated pre-ranks are shown in row 7.

(5) Sort all the pre-ranks to get the integral error rank (row 8).

(6) According to the updated error ranks (row 8), the sampled errors in the 2nd iteration (row 4) are reordered. The example for the 1st time step is demonstrated in dotted boxes in Table A.1. The error rank at the 1st time step is updated as 6, and the rank 6 corresponds to the error value -0.02 in the 2nd iteration. Therefore, -0.02 is the input error at the 1st time step in the 3rd iteration. Following this example, the sampled errors at all the time steps are reordered. The results are shown in row 9. Figure A 1 (b) demonstrates that after reordering the errors with the inferred ranks, the estimated errors are much closer to the true input error, and the mean square error (MSE) of the model residual reduces in Table A 1.

(7) The reordered input error will lead to new input data, a new model simulation and a new model residual. The residual result and its MSE statistic are shown in row 10 and 11 respectively.

(8) Check the convergence: If the objective function or likelihood function meets the convergence criterion, stop and the input error estimation is accepted. Otherwise, $q=q+1$, repeat step (3)~(8) until $q$ is larger than the maximum number of iteration $Q$.

**Appendix B: The time series of results in the case study**

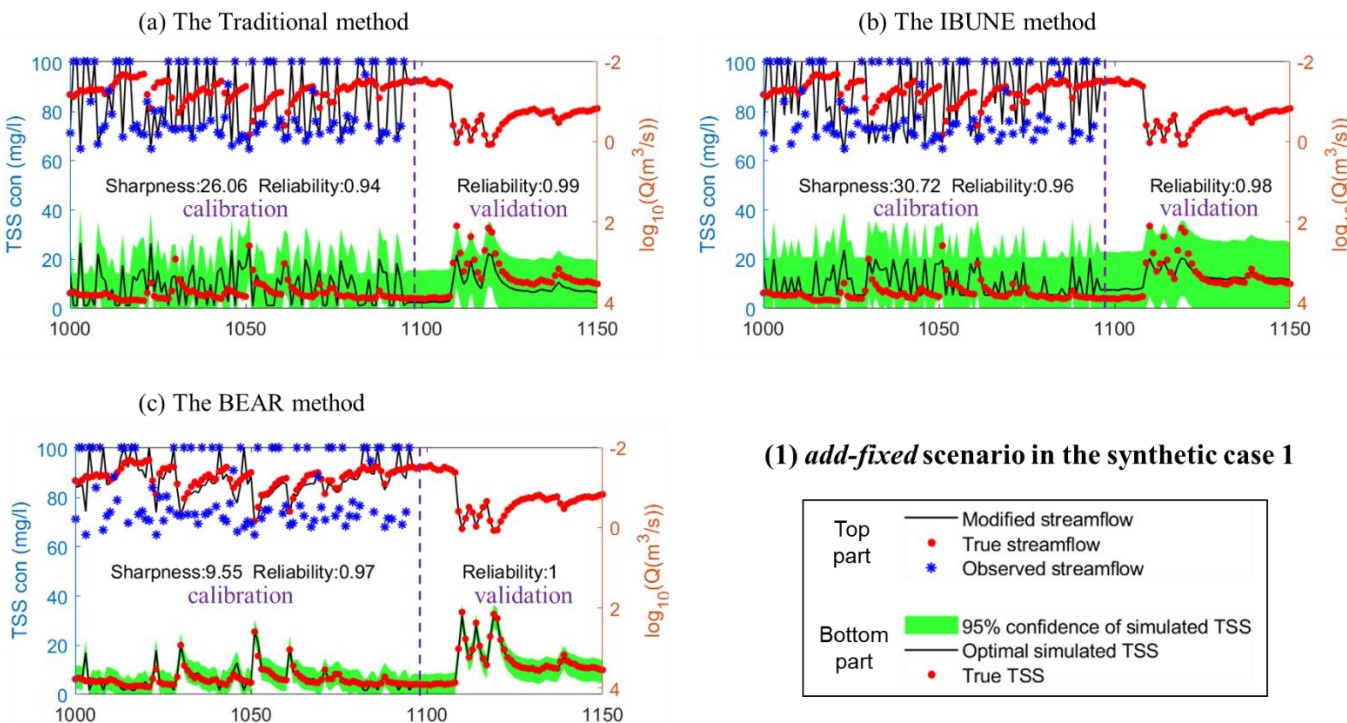

**Figure B 1(1) Comparison of time series of synthetic data and uncertainty bands estimated via three calibration methods (including the Traditional method, the IBUNE method and the BEAR method; algorithms are explained in Sect. 2.4) for a select period of *add-fixed* scenario in the synthetic case 1 (notations are given in Table 2)**

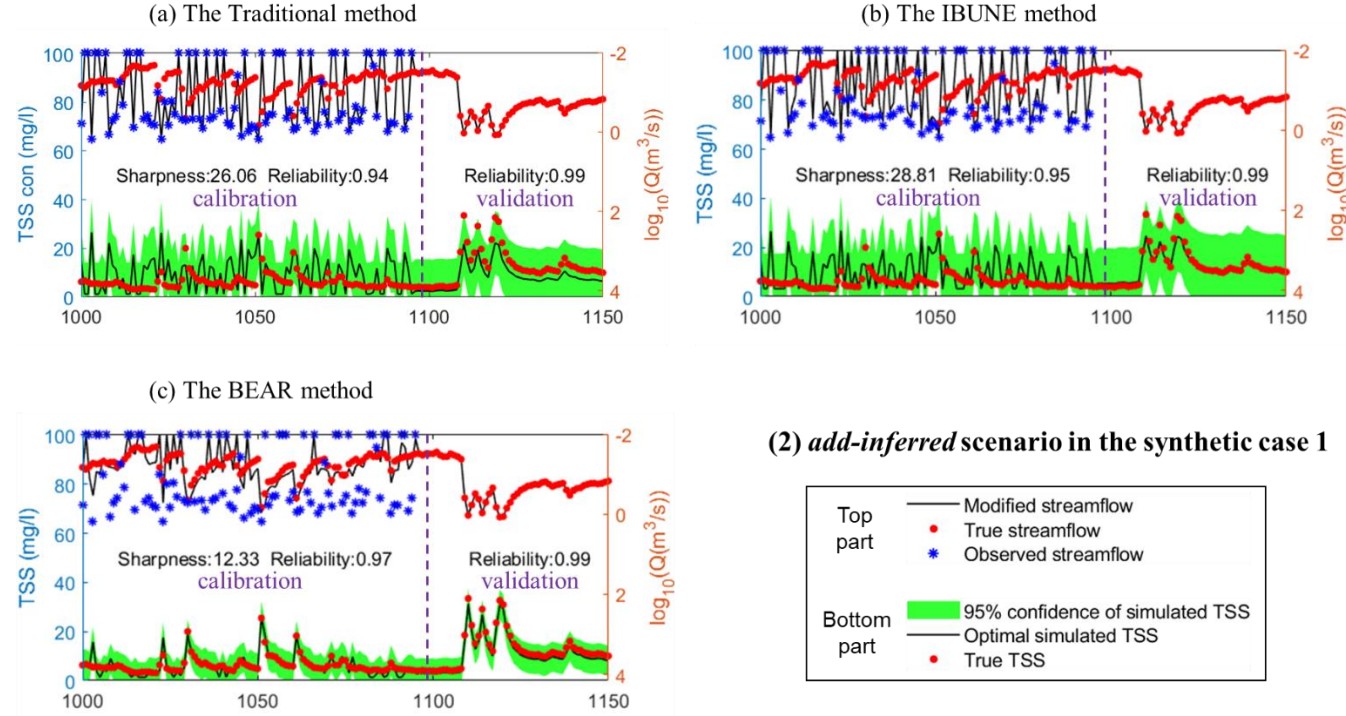

**Figure B 1(2) Comparison of time series of synthetic data and uncertainty bands estimated via three calibration methods (including the Traditional method, the IBUNE method and the BEAR method; algorithms are explained in Sect. 2.4) for a select period of *add-inferred* scenario in the synthetic case 1 (notations are given in Table 2)**

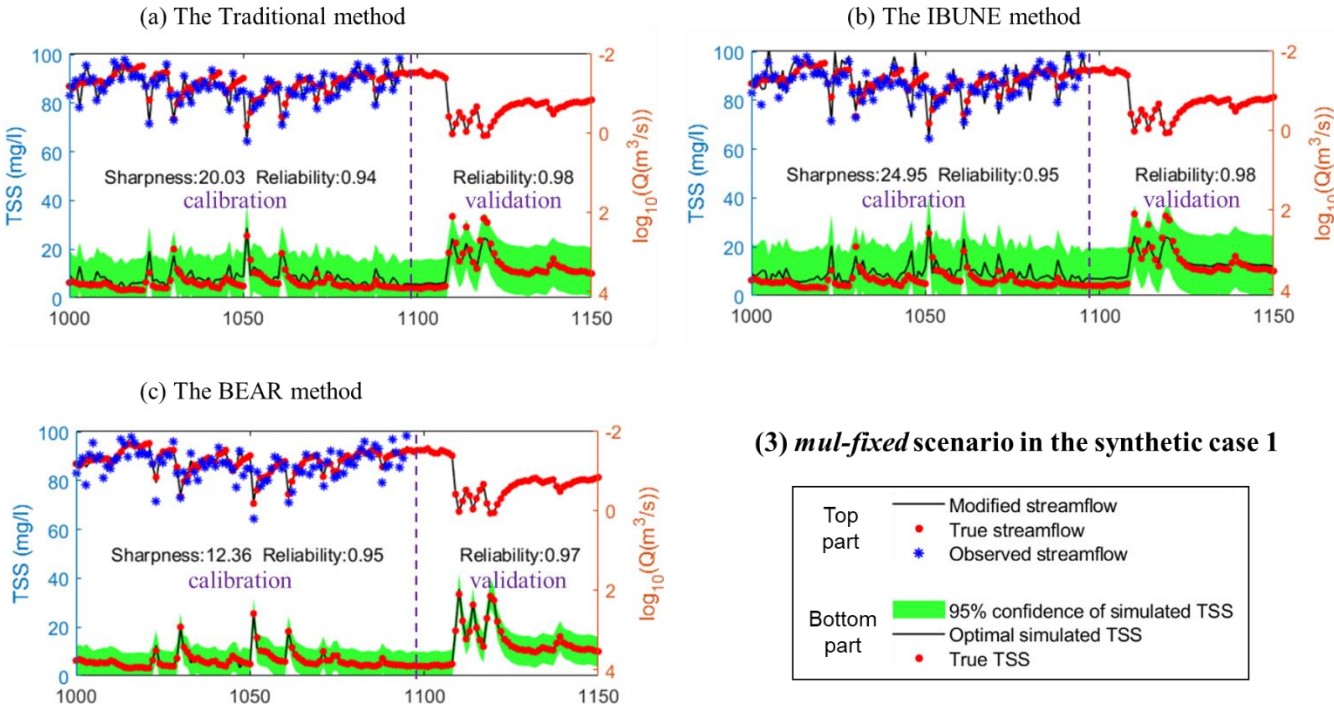

(a) The Traditional method

(b) The IBUNE method

(c) The BEAR method

(3) *mul-fixed* scenario in the synthetic case 1

**Figure B 1(3) Comparison of time series of synthetic data and uncertainty bands estimated via three calibration methods (including the Traditional method, the IBUNE method and the BEAR method; algorithms are explained in Sect. 2.4) for a select period of *mul-fixed* scenario in the synthetic case 1 (notations are given in Table 2)**

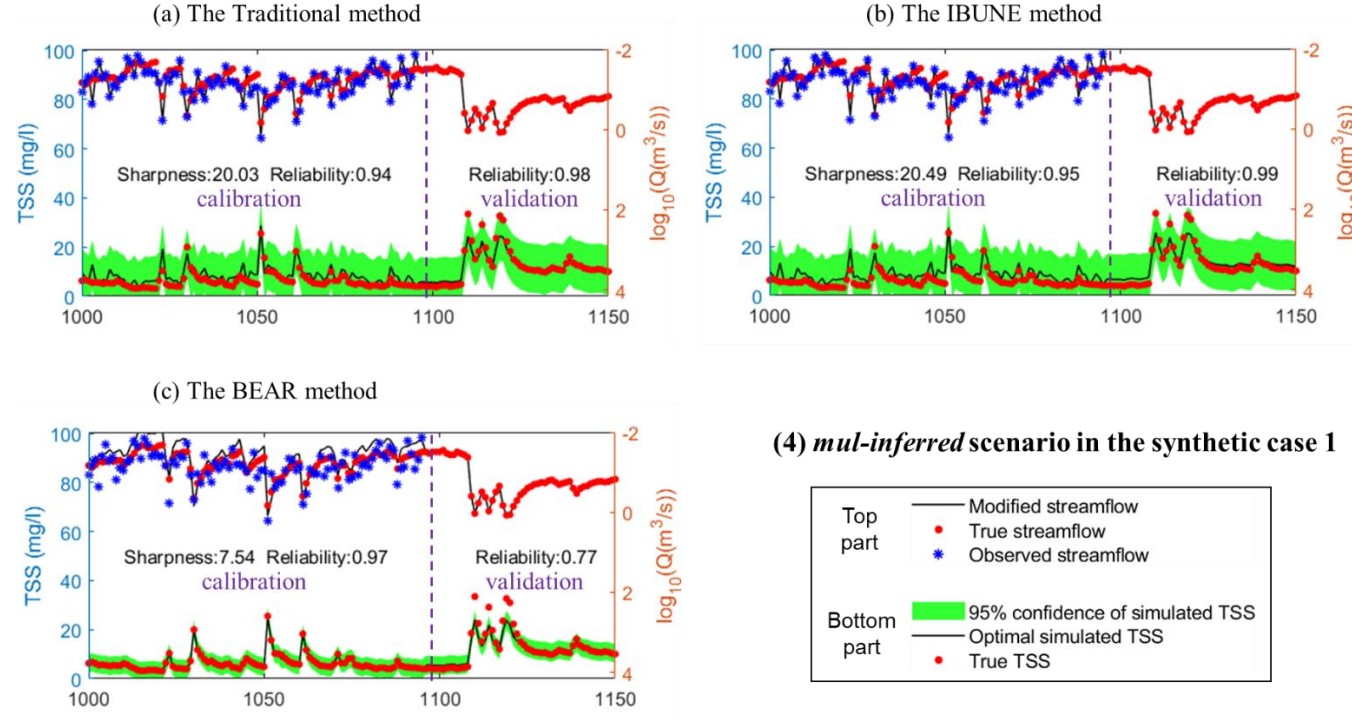

**Figure B 1(4) Comparison of time series of synthetic data and uncertainty bands estimated via three calibration methods (including the Traditional method, the IBUNE method and the BEAR method; algorithms are explained in Sect. 2.4) for a select period of *mul-inferred* scenario in the synthetic case 1 (notations are given in Table 2)**

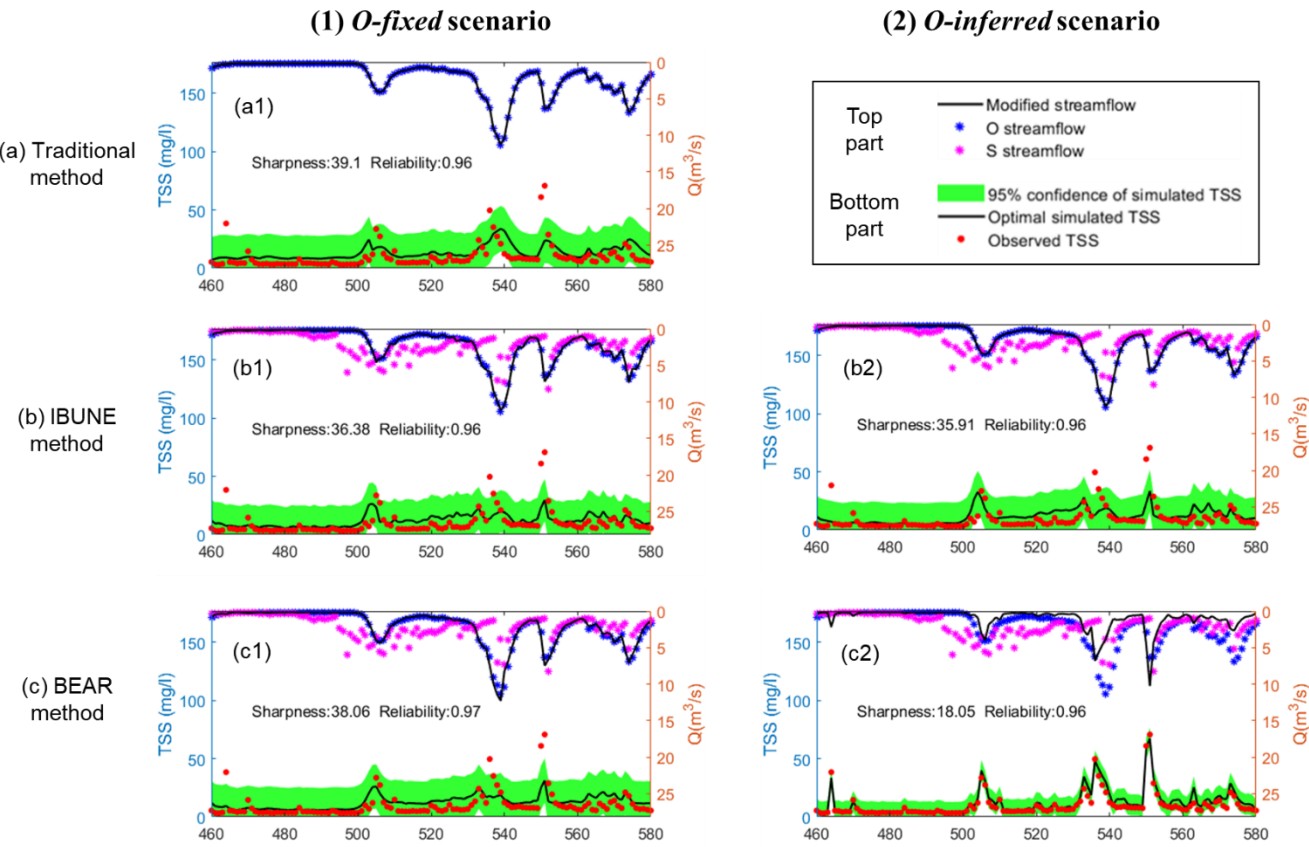

**Figure B 2(1) Comparison of time series of real data and uncertainty bands estimated via three calibration methods (including the Traditional method, the IBUNE method and the BEAR method, algorithms are explained in Sect. 2.4) for a select period of *O-fixed*,**

***O-inferred* scenarios in the real case (notations are given in Table 2)**

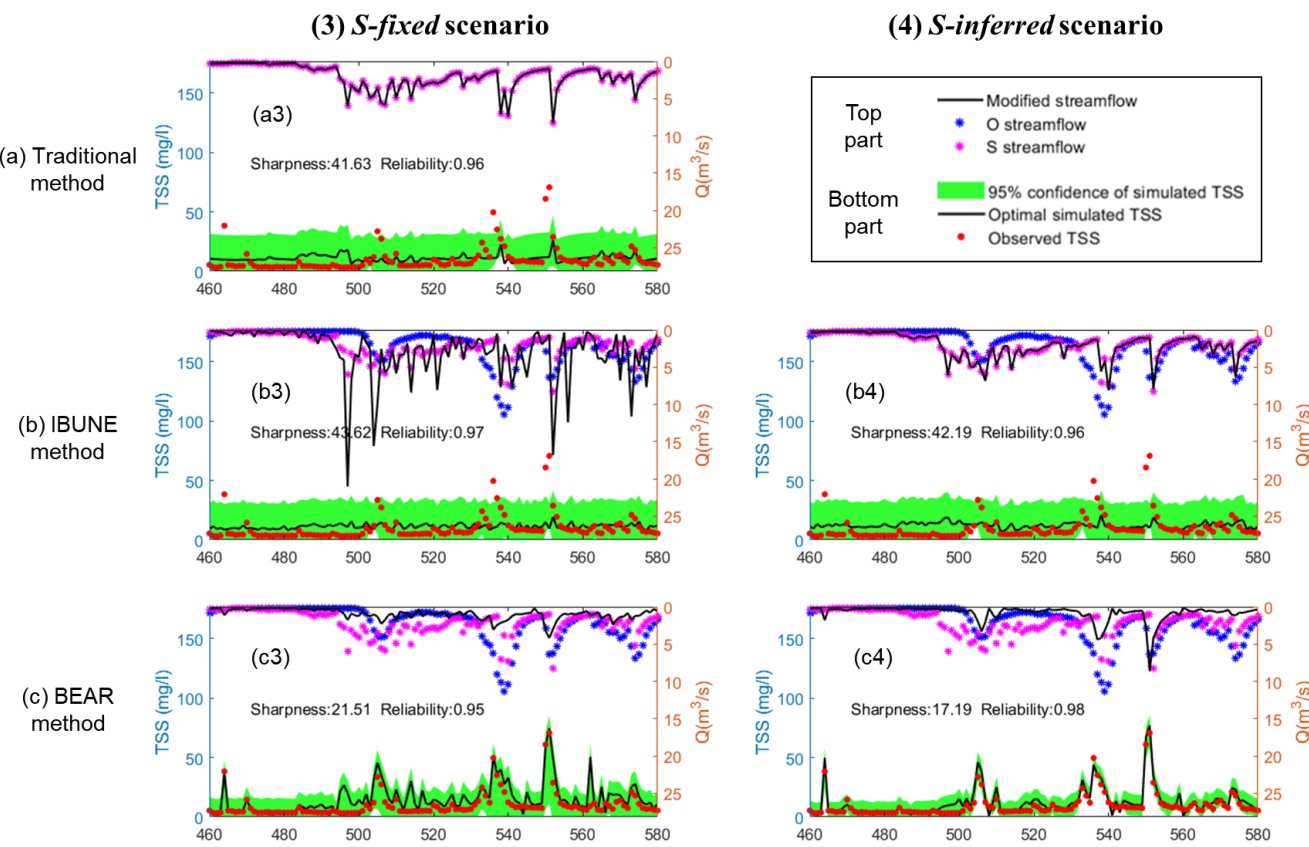

**Figure B 2(2) Comparison of time series of real data and uncertainty bands estimated via three calibration methods (including the Traditional method, the IBUNE method and the BEAR method, algorithms are explained in Sect. 2.4) for a select period of *S-fixed* and *S-inferred* scenarios in the real case (notations are given in Table 2)**

**Code/Data availability**

The daily streamflow and TSS concentration data for real case catchment (ID: USGS 04087030) can be accessed by the National Real-Time Water Quality website of USGS, the link is https://nrtwq.usgs.gov/ .

**Author contribution**

Lucy Marshall and Ashish Sharma designed the research. Xia Wu developed the research code, analyzed the results, and prepared the manuscript with contributions from all co-authors.

**Competing interests**

The authors declare that they have no conflict of interest.

**Acknowledgments**

The authors are thankful to the editor and reviewers for their constructive comments, which helped us substantially improve the quality of the manuscript. This work was supported by the Australian Research Council [FT120100269] and the Australian Research Council (ARC) Discovery Award [DP170103959] to Dr. Marshall, the National Natural Science Foundation of China [52109015] and the Jiangsu Postdoctoral Research Funding [2021K046A] to Dr. Wu, and the National Natural Science Foundation of China [51979004] and [41830752].

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
