# Peer review of "Quantifying input uncertainty in the calibration of water quality models: reordering errors via the secant method"

_Hydrology and Earth System Sciences, 2020_

## Referee Comment (RC1) · Anonymous Referee #1 · 9 Dec 2020

Review of "Quantifying input uncertainty in the calibration of water quality models: reshuffling errors via the secant method" by Xia Wu, Lucy Marshall and Ashish Sharma

**Summary** The study proposes and demonstrates an algorithm for quantifying input uncertainty called BEAR (Bayesian error analysis with reshuffling). It is claimed that the method is suitable to overcome restrictions of current state-of-the-art approaches like high dimensional computational problems or underestimation and misidentification of error sources. For this purpose, the algorithm employs the secant method to estimate a certain rank of error associated to input data from an underlying rank distribution of errors. After introducing the method, it is demonstrated on the task of total suspended

solids modelling in, first, a synthetic case study and, second, a real test case. Thereby, both, the effectiveness and the limitations are shown and discussed. Finally, transferability of the method within the field of water quality modelling and potential routes of improvement are presented.

**General comments** The issue of uncertainty quantification in modelling is for sure one of high importance. By focusing on input uncertainty this study addresses a branch that is particularly challenging in this field. Contributions in this direction deserve attention and the topic of this manuscript is suitable for the journal. However, certain issues regarding content and presentation of the material require to be addressed:

- Maybe it is just the presentation, but it was not straightforward to see how the method exactly works. Aside of a more detailed explanation, providing more illustrations to support explanations about how the method exactly works might help, e.g. displaying the secant method itself, error distribution in rank space, etc.

- By design, the BEAR method seems to shuffle and pick errors (by their ranks) such that maximum fit to the data is achieved. Is this a proper addressment of the input errors in terms of quantification of input uncertainty? For instance, in L. 232 it is discussed that "method R always has much higher correlations with the true error series" and in L. 243 it outperforms the other methods with highest NSE values. Both seem to be effects from the BEAR method searching for optimally fitting errors until exactly the error is found that minimizes the gap between model predictions and observations.

- Expectations are raised that the method overcomes issues of state-of-the-art frameworks like BATEA and IBUNE. Yet, no direct comparison is shown which makes it hard to see the benefit of the method. Both these methods are frequently mentioned and a comparison is claimed. So far, there is a comparison of cases abbreviated by "T" (traditional), "D" (distribution) and "R" (BEAR method

itself). "D" is referred to be "similar to the basic framework of the IBUNE method". However, this does not provide an actual comparison.

- The method is supposed to reduce "the potential search space for input errors" (L.360). I wonder whether this is the objective quantification of input uncertainty? Isn't it rather a comprehensive assessment of the errors and noise associated to input error and not searching in a sub-space of already collected errors and then selecting the one that fits best during predictions?

- Generally, a thorough discussion on the used error distributions is missing, e.g. why is a bias of 0.2 in the error function assigned without further discussion (l. 211)

- There is at least one article cited in the manuscript, that does not appear in the list of references (please see specific comments, l.190). Please assure correct referencing.

**Specific comments**

- L. 37-38: "...estimate the residuals between the measurements and proxy values..." -> yet, measurement error is not addressed

- L. 68: "variable" -> "scalar"– both, vectors and scalars represent variables

- Eq. 3: unnecessary, since given by equation (1)

- L. 84-91: repetitive, add details to the corresponding paragraph in the introduction

- L. 92: "innovation" -> rather "introduction" or simple "The secant method" as chapter header – the innovation was made before

- L. 98-99: Rank definition and concept -> requires further explanation

- L. 128ff: it sound like in ABC the requirements on the likelihood function are looser and therefore the method is easier to apply. However, requirements are also strict but ABC allows for Bayesian inference if the likelihood function is intractable. -> Please reformulate and clarify.

- L. 132ff: Notation "OF" not explained. Overall, the introduction of ABC and SMC is not clear. Further, the motivation why SMC is used here is not given.

- L. 146: "…when 1000 proposed parameter sets…" -> is this suggested as general approach or an arbitrary choice for this study. Please explain.

- L.171ff: Please replace abbreviations T, D and R by their names. With all abbreviations that follow it is hard to keep track.

- LL. 190+196: "Sikorska et al, 2015" → missing in references

- Eq. 9: define parameter "b"

- L. 215ff: incomplete sentence

- L. 229ff: "calibrated via method T,…" -> misleading explanation. Please provide a more specific explanation of the calibration process under error scenarios T, D and R

- L. 257: "…the impacts of model structural error and output data error cannot be ignored." vs. L.264: "…other sources of uncertainty can be ignored" -> sound like a contradiction, please elaborate

- L. 282-283: "This illustrates that the impacts of other…" -> unclear phrase, please clarify and re-formulate

- L. 291: "… could be regarded as the reference value." -> Why? Please explain.

- L. 295-296: "... have an infinite number of combinations, while the error rank has limited combinations, dependent on data length." -> What is exactly meant here?

- L. 297ff. "Compared with the IBUNE framework..." -> there is no real comparison made, please see major comments

- L. 340: "for method R, an accurate input error model can constrain the adverse impacts..." -> wasn't this the problem to begin with? Please clarify this sentence.

- L 354-355: "However, the ability of these approaches needs further discussion in systems with correlated responses." -> Please clarify – what is the exact problem and why do ARMA models fit here?

- L. 358: "developed" -> "proposed" – the methods are already known but used in a way to address input error here.

- L 362: "... addresses the high dimensionality problem..." -> not shown

**Figures**

- General: Legends in figures should be improved, e.g. in terms of colors or placing

- General: Provide higher resolution and unify the legend (see especially Fig. 4 and 6)

- Figure 3: please use colors that are better distinguishable (see cases "T" and "R")

- Figure 3(4): NSE = 1 is unrealistic. Please see major comments.

- Figure 4 (c3,c4): model predictions are clearly shifted. Please elaborate on this offset.

- Figures 4 and 6: Maybe it is better to show these figures in the appendix and only present the most important subfigures in the main text.

**Tables**

- Tables 1-1 and 1-2: The tables could be presented as additional files but are not helpful in the main article
- Table 3: the "fixed" scenarios in the real test case are not fixed but provide small hyperparameter ranges

**Technical corrections**

- L. 128: double ","
- L. 142: "sth" -> make "s" italic
- Eq. 8: unspecified symbol
- L. 314: "q increasing until the objective..." -> incomplete sentence

---

## Referee Comment (RC2) · Anonymous Referee #2 · 10 Dec 2020

This paper developed a new algorithm called BEAR for accurate quantification of input errors in water quality modeling. The precondition of the BEAR algorithm is that the input uncertainty should be dominant and that the prior information of the input error model can be estimated. Results of both synthetic data and observed data indicated the efficiency of the algorithm. Overall, the paper is well rewritten and the topic is suitable for the journal. However, the following issues should be further explained and clarified before its submission: (1) There have been many studies focusing on the uncertainty of input data errors for hydrologic modelling, and many methods including Bayesian algorithm can be used for handling the issue. However, the gap between previous studies and this study was not explained clearly in Introduction. The motivation

of this study should be clearly clarified. (2) More detailed steps about how to use the BEAR algorithm should be explained. Besides, the advantages of the BEAR algorithm compared with conventional methods should be more clearly clarified for making clear understanding from readers. (3) Actually, the availability of prior knowledge of the input data error is important for modelling, but is also a difficult issue. It may be not enough only mentioning this issue in Conclusion. At least more discussions and the potential solutions should be provided. (4) The quality of some Figures in the manuscript should be improved to make all information clear.
* * *

---

## Referee Comment (RC3) · Anonymous Referee #3 · 17 Dec 2020

Complex patterns in input uncertainty such as spatial or temporal error correlations are an important topic in environmental science. In their present study, the authors seek to explore the ubiquitous issue of complex input uncertainty structures by proposing a novel method called Bayesian error analysis with reshuffling (BEAR). The proposed method is based on sampling an estimated input error and subsequently sorting the resulting realizations in an order which reduces residual mismatch to the observations. The authors then proceed to demonstrate the performance of their algorithm for a synthetic and a real case and compare its performance to a number of alternative setups.

I find the approach a very interesting and creative idea, and always appreciate it if

someone takes the risks inherent to exploring a new methodological idea. Unfortunately, I have some reservations concerning its theoretical justifiability, which I hope the authors can address. Failing that, there might also be alternative ways to achieve similar effects which might stand on more robust theoretical foundations. Concerning these suggestions and the method itself, I have the following (major) comments:

**1. Theoretical foundations**: A key step of the approach is sorting input error realizations to reduce the residual mismatch between model predictions and system observations. I fear that this compromises the randomness of the error realizations, with potential consequences for the validity of the Bayesian inference that are difficult to predict. Rigorously deriving this algorithm from more basic theoretical foundations might better illustrate the consequences of the authors' assumptions for Bayesian inference. If the authors do not have this expertise themselves (not to worry: few people in the environmental sciences do), I recommend seeking the help of the local statistics department – they are often keen to help. It seems an unfortunate truth of Bayesian statistics that interesting ideas for algorithms which sound nice on paper often tend to violate the Bayesian framework in unforeseen ways.

**2. Improving future performance**: While I am not intimately familiar with the alternative methods (BATEA, IBUNE) referenced by the authors, improving future model performance is a common motivation for learning complex uncertainty structures in hydrological models. The approach in this manuscript, however, requires a concurrent time series of observations. As a consequence, it does not really improve the error model itself and hence offers little value for improving future predictions. On the other hand, attempting to learn time-varying bias or correlation structures in the input errors – admittedly a sometimes computationally formidable task – can increase the likelihood of future predictions substantially. Such approaches would also be significantly easier to justify. I would recommend mentioning this limitation for the BEAR algorithm in the manuscript, or – better yet – either explore or propose ways on how this limitation could be circumvented.

**3. The influence of other error types**: It seems the authors focussed most of their attention on input uncertainty. While I agree that input uncertainty can play a very important role, the influence of observation errors and model structural uncertainty plays a substantial role as well. In this study, the authors assumed both the model and the output observations were error-free – and derived their algorithm accordingly –, but in practice these assumptions are virtually never met. I would encourage the authors to explore how (if at all) their algorithm can avoid surrogacy effects in the presence of observation and model errors. (What I mean by "surrogacy effects" is that the algorithm's adjustments to the input error realizations also 'soak up' [and consequently mask] errors in the output observations and the model itself in a bid to reduce the output residuals. This is of course undesirable.) See also my penultimate minor comment below.

**4. Deterministic functions as an alternative**: If the authors find that their algorithm might be based on flawed assumptions (following a more detailed theoretical derivation or investigation of the distribution it effectively samples from), the authors might wish to explore possible alternatives. If reducing the output residuals through adjustments to the input data remains the goal, a safer route might be to couple a deterministic input pre-treatment routine with the WQM and add its parameterization to the WQM parameter vector. Functionally, this pre-treatment routine can simply be interpreted as part of the deterministic model. Choices for this pre-treatment routine could be, for example, a one-dimensional spline which re-scales input magnitudes non-linearly (see the attached Figure 1, for an example with three extra parameters). More complex function choices might allow the consideration of lag, temporal or spatial correlations, etc. This would have the additional advantage over the BEAR framework that this pre-treatment routine could also improve future predictions, assuming that it compensated true bias and is not overfitted. This comment is not a request for change, but a hopefully constructive suggestion for alternatives so that the authors might salvage some of their work in case it would turn out theoretically indefensible.

**5. BEAS instead of BEAR**: This could be filed under nit-picking, but since the algorithm's name features so prominently, I chose to raise this to a major comment instead. The use of the word 'shuffling' implies randomness in the re-ordering. If I understood the authors' algorithm correctly, though, the re-ordering itself is entirely deterministic. As such, changing the name to something along the lines of "Bayesian error analysis with sorting" (BEAS) or "Bayesian error analysis with re-ordering" (if the authors like to retain their – admittedly very nice – acronym) might better reflect its deterministic nature.

**6. Why ABC**: In the manuscript, the authors use an "Approximate Bayesian Computation via Sequential Monte Carlo" (ABC-SMC) approach. While I am not personally familiar with this approach, I struggle to see why it is necessary to resort to ABC, aside from any potential (forgive me) self-inflicted complications induced by the BEAR algorithm. The model and output variables seem pretty simple, so to my untrained eyes it is difficult to see why the formulation of an analytical likelihood should be impossible in this case. One could also cast the procedure the authors presented in Figure 1 with very few changes in terms of an MCMC routine, provided the re-ordering or error realizations ends up being statistically justifiable, of course. I would encourage the authors to provide a bit more detail on why ABC was necessary.

**7. Focus on a good fit**: A key idea which seems to permeate the present manuscript is that it is desirable to obtain error realizations, if necessary by force (i.e., re-ordering), which match the observations as closely as possible. This is of course true, but not at all cost. Even assuming a severely mischaracterized prior input error distribution (e.g., a Gaussian with a standard deviation of 1E-10 and a mean of -1E8), one could theoretically obtain an error realization which causes the model to fit the observations perfectly if we only drew sufficiently (read: infinitely) many samples. The challenge, then, is not to find such a realization within our prior distribution (it will exist in any distribution with sufficiently broad support), but to find a distribution from which there is a high probability to obtain such a sample. Crucially, such a distribution should be independent

from future observations, and I fear that this may not be the case for the approach proposed in this manuscript. In this approach, after re-ordering, the realizations are no longer i.i.d. samples from the input distribution (similarly to how one could interpret correlated Gaussian samples merely re-ordered independent Gaussian samples, but would nonetheless be wrong in claiming that an independent Gaussian distribution is identical to a correlated Gaussian distribution). If the authors decide to pursue my request for a derivation of a theoretical foundation for their approach (see comment 1), I recommend focussing on investigating from what effective distribution they are really sampling. Considering BEAR's ability to yield a reliably good fit with seemingly arbitrary prior input error realizations, I fear that the distribution you effectively sample from may be well approximated with (for example) a Gaussian with a mean inversely obtained from the observation residuals. If this turns out to be the case, the method would be more or less equivalent to just calculating the input error residuals through an inverse method of choice by minimizing the output residuals. This would not be very useful, and I would recommend exploring one of the approaches I suggested in comment 4 instead. See also my penultimate minor comment.

Specific comments:

Line 22-25: You mention the importance of complex interactions of different error sources directly in the first paragraph but proceed to largely ignore their influence in the remaining manuscript. I think this part is important and should be discussed in greater detail in the remainder of the manuscript (particularly also the methods/theory section).

Line 49-51: During this review, I have briefly glanced into the corresponding methods BATEA and IBUNE, and apparently there was quite a commentary battle between the authors over these methods (see doi:10.1029/2007WR006538 and https://doi.org/10.1029/2008WR007215), and Renard et al. 2009 noted that IBUNE may in fact not reduce dimensionality. The choice is of course ultimately up to the authors, but it might be useful to add a small comment noting that the claim of dimension-reduction by IBUNE is also challenged.

Line 69-75: I would be careful here. In reality, there are many different error sources, certainly not the least of which is model structural error. Calibrating (i.e., simply minimizing the residuals between simulated and observed output) in the presence of other error sources is prone to surrogacy effects, so you can never really be sure you recovered the 'true' parameters. Even making the (unrealistic) assumption that we could manage to completely remove all error sources in the model and its input, we could still only retrieve the 'true' model parameters if our inverse problem is unique. I would mention these restrictions here (only works if all error sources can be removed completely, unique inverse problem).

Line 70-73: In Equation (2), the variables Yo and Ys are used without any introduction. I assume both variables stand for the observed and simulated output. Please introduce these variables.

Line 76-77: A critical thing here is that $\varepsilon$ is previously introduced as an error, which implies that you consider it to be a random variable. However, subtracting a (say) Gaussian random variable from another Gaussian variable with the same properties does not reduce variance to zero but actually doubles it (if both random variables are independent). What you seem to have in mind here only works if $\varepsilon$ and $\varepsilon$p have identical properties and are perfectly correlated (note that this implies a lot more than just sharing the same statistical moments!), or if you are talking about error realizations. You should clarify this. This relates to major comment 7. You can only create this perfect correlation if you can somehow extract the error realizations of $\varepsilon$ (which only works under the assumption that you already have the input samples, that there are no other errors, and that the inverse problem is unique). Consequently, I fear that you may create/mimic this perfect correlation by implicitly solving an inverse problem, which would make the proposed method not very useful.

Line 79: In Equation (4), you also mark the parameters – on which this entire exercise should be conditional on – as changing due to your proposed approach. The equations you have shown so far imply that the procedure you describe here is applied after

parameter calibration. During a first reading of this paper, it is not immediately clear why the calibrated parameter values should change with your proposed approach. On a second reading, it becomes evident that you do not really calibrate but sample the parameter posterior, but that this sampling process is inter-woven with your BEAR routine, hence the parameter values are also affected. I recommend commenting on this already here to save your readers some confusion. Maybe it would also help not to talk about calibration at all in this context.

Line 81-83: I would be very careful with this statement. This only happens if the model parameters $\theta$p and the input error residuals $\varepsilon$xp cannot compensate each other, i.e. if there is only a single, unique combination of parameters and errors which yields zero residual. If you have a Pareto front along which different values of $\theta$p and input error residuals $\varepsilon$xp yield zero residual, you cannot be certain that you have correctly identified the 'true' parameter values and the 'true' input error, even in a scenario where no other errors/uncertainties exist. In addition to this, my reservations concerning other error types raised in major comment 3 also apply. I would recommend changing this statement accordingly and exploring its consequences for your algorithm in greater detail.

Line 104-106: I would rephrase this a bit, because following the procedure you outlined in Figure 1 (a very nice schematic, by the way), it is not only two steps: you sample the error once, then iterate over a large number of re-ordering steps until you find an order which minimizes your output residuals. This could do with some clarification.

Line 115-116: If I understood your explanations here correctly, maybe an easier way of explaining what you are doing is that you sort your updated error ranks, then assign to each of them a new integer rank based on its position in the sorted list. This might be easier than trying to explain this procedure with scaling.

Line 125-131: As mentioned in major comment 6, please devote some space to explain why the models you use in the following necessitate the use of ABC. Even after

going through the manuscript a few times, I struggle to see why standard Bayesian approaches would be impossible to use. At the risk of evoking the anger of our ABC-focussed colleagues: direct is usually better than approximate. It also does not become clear in the manuscript why an ensemble-based approach is used – couldn't the same procedure be implemented in an MCMC-style acceptance/rejection algorithm? If there are some ABC reasons for requiring an ensemble, it you might also want to explain why and how it is used.

Line 132-136/Figure 1: This explanation of the method is very short, and essentially only explains that you approach the posterior distribution iteratively through a number of intermediate steps, but not how this is achieved exactly. Figure 1 provides more information and suggests some sort of acceptance/rejection scheme depending on whether your procedure can reduce the error residuals below a certain threshold, but the nature of the posterior distributions from which new parameter values are drawn remains undefined. You also seem to update an input error parameter $\eta$x, which seemingly contradicts statements you made suggesting the input errors are sampled from a pre-estimated distribution (Line 10, Line 193-194). This step is also never mentioned in the text itself up to this point – you only mention that you estimate the input error distribution's hyperparameters much later. The text also frequently mentions 'populations', which evoke the idea of an ensemble-based method, but none of the steps mentioned in the text so far actually seem to require an ensemble. Please provide some more (written) detail about how your algorithm functions exactly.

Line 145-148 and Line 159-162: This is just a comment towards the general "Why ABC?" discussion. It seems to me that a classic MCMC procedure would avoid the need for adjusting the acceptance threshold dynamically, as proposed parameters are always compared to the previous entry in the chain.

Line 154-157: I confess that this explanation is quite impenetrable, and probably causes more confusion here than it does good. I recommend restructuring this explanation or removing it altogether. A good alternative would be to visualize this with

a small figure, possibly added to the supporting information if length limitations to not permit embedding it into the main text.

Line 192: I do not see from Equation 8 or 9 or their surrounding text how the spatial scale factors into this model. Through the Sa variable? Please clarify this.

Line 200: In Equation 8, you do not introduce Smax and $\kappa$. Please introduce these variables as well.

Line 203: In Equation 9, you do not introduce a and Qtb. Please introduce these variables as well.

Line 205: In Equation 10, you do not introduce Qt. Please introduce this variable as well.

Line 209-211: For this section, there are a few assumptions which could warrant greater discussion. If the errors are normally estimated in advance based on a rating curve, why is there a constant offset of 0.2? Couldn't this systematic bias be corrected through the rating curve itself? Alternatively, if the offset is necessary because your errors are asymmetrically fat-tailed, wouldn't a different distribution (such as a scaled beta or gamma distribution) be a better choice? It is commendable to make the synthetic test case more challenging by introducing bias as well, but how would this be recognized a priori in a real test case if it wasn't already considered in the rating curve? Some more information might clarify the authors' choice of distribution for the audience.

Line 224-226: This part here is a bit unclear. What I deduce from the context is that you looked at two scenarios – one, where you left the prior input error fixed, and one where you estimated the input error hyperparameters as well. I would not talk about 'conditions' in this context, but rather about 'scenarios'. If I understood your drift here correctly, I would also add a comment which puts more emphasis on the fact that you subvert one of the principal assumptions you made earlier in the second scenario

(namely, that the input error distribution is a prior/pre-estimated).

Line 232-234: I would recommend to critically re-examine this part in the light of major comment 7. The high correlation of scenario R with the realizations of the synthetic true error series – which are supposed to be realizations from an independent Gaussian distribution – might be reason for concern, as they suggest that you might be implicitly solving an inverse problem for the input error residuals. This would have little to do with a Bayesian framework.

Figure 4: Unfortunately, this figure is really hard to read. If possible, I would recommend splitting this up into several figures and providing the figures for individual scenarios in the supporting information. The choice of colors also makes it very difficult to see what's going on (especially the neon green and the soft peach color). I am not familiar with the HESS compiler, but I would also recommend either a significantly larger resolution and a different image format such as .tiff or .gif, as the current figure is in quite a low resolution and has serious compression artifacts. For graphs such as this one, vector-based formats such as .svg or .pdf (if saved straight from Python with pyplot.savefig) might also allow readers of the electronic version to zoom in arbitrarily close for details. This could be particularly valuable here, since most of the relevant details are quite small.

Line 296-297: I would remove this statement, as you have not experimentally backed this statement up and it is not immediately obvious. I see little reason why inverting the observation residuals to find optimal input error realizations would be a more difficult task than re-ordering a pre-existing set of realizations. Quite the opposite, in fact.

Line 301-307: This is a very important paragraph. As I suggested in comment 7, through re-ordering you are no longer sampling from the prior input error model, which makes the protection from perfect fits you mention here somewhat arbitrary. As an illustration, I would like you to consider the behaviour of this re-ordering for longer time series: For a single observation, re-ordering can yield no improvement, and the

residual fit depends exclusively on the realization you drew. For a few observations, re-ordering will induce moderate improvements, and the residual fit depends somewhat on the realization you drew. However, in the limit of infinitely many observations (assuming the statistical moments of the prior input error distribution are correctly characterized), re-ordering your error realizations should allow the residuals to be compensated completely at every single observation, irrespective of what specific realization you drew. This makes the protection against overfitting (and the expected residual error) dependent on the length of the observation time series and seems to converge towards deterministic (over)fitting. At the same time, the effective input error uncertainty decreases to zero. In a conventional Bayesian framework, even if the correlations in the input errors are perfectly identified, this would never happen.

Line 314: The dot after (Fig 1.) should probably be a comma

Summary:

In summary, I find the approach an interesting and ambitious idea, but have reservations concerning its theoretical validity, which I hope the authors can address in their revision. If my fears concerning it solving an implicit inverse problem for the input error residuals happen to be confirmed, the authors might consider the following alternative avenues:

a) the approach might be re-interpreted as a diagnostic tool for input error residuals; there is some value in identifying input error residuals and the correlations between them. In this case, however, it may be worthwhile to investigate whether the re-shuffling strategy is needed, or whether a more straightforward inverse method might be more efficient.

b) If predictive improvements are desired, following the suggestions in major comment 4 could be a viable and interesting alternative avenue

I wish the authors the best of luck with the manuscript, and hope that my comments

are useful.

[Figure]

**Fig. 1.** Example of a non-linear re-scaling with a spline defined by three control points (red dots). You could use such a spline to scale the input values non-linearly.

---

## Author Comment (AC1) · 22 Mar 2021

**Response to Reviewer #1:**

The study proposes and demonstrates an algorithm for quantifying input uncertainty called BEAR (Bayesian error analysis with reshuffling). It is claimed that the method is suitable to overcome restrictions of current state-of-the-art approaches like high dimensional computational problems or underestimation and misidentification of error sources. For this purpose, the algorithm employs the secant method to estimate a certain rank of error associated to input data from an underlying rank distribution of errors. After introducing the method, it is demonstrated on the task of total suspended solids modelling in, first, a synthetic case study and, second, a real test case. Thereby, both, the effectiveness and the limitations are shown and discussed. Finally, transferability of the method within the field of water quality modelling and potential routes of improvement are presented.

**General comments:**

The issue of uncertainty quantification in modelling is for sure one of high importance. By focusing on input uncertainty this study addresses a branch that is particularly challenging in this field. Contributions in this direction deserve attention and the topic of this manuscript is suitable for the journal. However, certain issues regarding content and presentation of the material require to be addressed:

We thank the reviewer for the overall positive assessment of the manuscript and helpful comments. We have responded to each point in turn in the following sections. The comments from the reviewer are provided in blue text and our responses are organized point-by-point in black text. The manuscript text after the proposed changes is shown in "*black italics"* and the equation and section number are shown in yellow highlight.

It should be noted that we are proposing that the method name will change from the "Bayesian error analysis with reshuffling" into "Bayesian error analysis with reordering". This is based on suggestions by one of the reviewers, as the word "shuffling" implies randomness in the reordering, while the reordering in our method is determined by the model residual error. The term "reordering" better reflects the deterministic nature of error quantified via this new method.

1) Maybe it is just the presentation, but it was not straightforward to see how the method exactly works. Aside of a more detailed explanation, providing more illustrations to support explanations about how the method exactly works might help, e.g. displaying the secant method itself, error distribution in rank space, etc.

Thanks for your suggestion. We propose to address this by modifying the methodology in the following points to make the algorithm clearer:

(1) Summarize the main steps in the BEAR method upfront:

"*The BEAR method consists of the following key steps: 1) Sample the errors from the assumed error distribution to maintain the overall statistical characteristics of input errors; 2) Update the input error ranks via the secant method; 3) Reorder these sampled errors according to the updated error ranks, leaving the error magnitudes unchanged; 4) Repeat (2) and (3) for a few iterations until a defined target is achieved.*"

(2) Integrate an example and its illustration in Appendix to explain the specific steps involved. More explanation for the rank estimation via the secant method and the reordering steps will be added (see an illustration of this in the following Appendix A).

(3) Separate the description of the BEAR method from the ABC-SMC calibration scheme. following suggestions from other reviewers, we propose this because the ABC-SMC calibration algorithm is not necessary in the BEAR method, and the core idea of the BEAR method (the reordering strategy in the rank estimation) can be easily applied in any other calibration algorithm, for example, MCMC and SMC algorithms.

2) By design, the BEAR method seems to shuffle and pick errors (by their ranks) such that maximum fit to the data is achieved. Is this a proper addressment of the input errors in terms of quantification of input uncertainty? For instance, in L.232 it is discussed that "method R always has much higher correlations with the true error series" and in L.243 it outperforms the other methods with highest NSE values. Both seem to be effects from the BEAR method searching for optimally fitting errors until exactly the error is found that minimizes the gap between model predictions and observations.

The reviewer raised an important question. The BEAR method works well under the circumstance where the input error is dominant in the total uncertainty, where minimizing the residual error has a similar effect as minimizing the input error. From this point of view, the sampling and reordering strategy in the BEAR method provides an effective way to identify the input error according to the residual error. This is what the Reviewer refers to as "searching for optimally fitting errors until exactly the error is found that minimizes the gap between model predictions and observations". Like current methods that BEAR seeks to demonstrate an improvement over, the input error compensates for other errors, a step that is constrained by accurate prior information of the input error distribution being available. However, the compensating effect in the BEAR method is more apparent because it is much more effective than other current methods in minimizing the gap. Thus, the accuracy of the input error model is particularly important in the BEAR method. The analysis and discussion in Section 4.2 will be modified to convey this as follows:

*"The IBUNE method takes advantage of stochastic error samples to modify the input observations (Ajami et al., 2007). In Fig. 4 and Fig. 6, the uncertainty bands of modified inputs (blue parts) encompass the original input data, illustrating that the intrinsic quality of the input data plays an important role in the algorithm performance. Fig. 6 demonstrates that if the input error is insignificant in the residual, like in the O-fixed and O-inferred scenarios for the real case, the resultant simulations will fit the observed output (green line) well. Otherwise, the simulations are far away from the observed outputs (black line) due to inaccurate input observations (in the S-fixed and S-inferred scenarios in the real case). As per the finding in the previous study of Renard et al. (2010), if the $\sigma$ of input errors is inferred with the model parameters, the IBUNE method will underestimate $\sigma$ (in Fig. 3(1) and Fig. 5(a2)). If $\sigma$ is fixed as per the prior information, the input modification and model simulation cannot be improved in the scenarios with large intrinsic $\sigma$ of the input errors, demonstrated by a wider band in Fig. 6(b3) than in Fig. 6(b4). From the above, the data quality is more important than the availability of prior information for the IBUNE method, especially when the intrinsic $\sigma$ of the input error is large.*

*However, the findings in the BEAR method are quite different. Although the BEAR method infers the input error also by minimizing the model residual error, it is much more effective than the IBUNE method. For the synthetic case (Fig. 3(c)) and real case (Fig. 5(c)), the model simulations via the BEAR method (red parts) are very close to the output observations (green line). In other*

*words, the estimated input error mainly depends on the output observation. Therefore, in the real case with the same output observation (Fig. 6(c)), the modified inputs are consistent among the different scenarios. If the input uncertainty is dominant over the output observational uncertainty, the BEAR method effectively employs more accurate information (output observations) to modify the less precise information (input observations).*

*To constrain the impacts of the other sources of error, accurate prior information about the input error model is important in the BEAR method. The fixed scenarios are assumed to have more accurate prior information than inferred scenarios. In the synthetic case, fixed scenarios always produce a higher NSE of the modified input (Fig. 3(5)) and a larger correlation in the estimation error (Fig. 3(3)) than inferred scenarios. In the real case in Fig. 6, the modified inputs in fixed scenarios are closer to the streamflow observation from the rating curve than the modified inputs in inferred scenarios.*

*To sum up, the role of prior information regarding the input error model is more important in the BEAR method than in the IBUNE method. A more accurate input error model can bring a more precise estimation of input errors by constraining the adverse impacts that other sources of errors may have."*

3) Expectations are raised that the method overcomes issues of state-of-the-art frameworks like BATEA and IBUNE. Yet, no direct comparison is shown which makes it hard to see the benefit of the method. Both these methods are frequently mentioned and a comparison is claimed. So far, there is a comparison of cases abbreviated by "T" (traditional), "D" (distribution) and "R" (BEAR method itself). "D" is referred to be "similar to the basic framework of the IBUNE method". However, this does not provide an actual comparison.

The reviewer is correct that we propose denoting the BATEA and IBUNE methods instead of explicitly naming them in our comparison. We will change the abbreviation to the full name of the methods as per the reviewer's suggestion, and add more explanations about this comparison, as follows:

*"The application of the BATEA framework is limited by high dimension computation (Renard et al., 2009). In quantifying the data-varying errors (rather than the event-varying errors in the study*

*of BATEA (Kavetski et al., 2006)), the computational dimension is easily excessive and the BATEA probably becomes impractical (Haario et al., 2005). Therefore, the BATEA method is not considered in the comparison. In this study, three methods are compared to evaluate the ability of the BEAR method in quantifying input errors. The first one is the "Traditional" method, regarding the observed input as error-free without identifying input errors (i.e. Eq. (2)), while the other two methods employ a latent variable to counteract the impact of input error and build the modified input (i.e. Eq.(4)). One of them is the "IBUNE" method, where potential input errors are randomly sampled from the assumed error distribution and filtered by the minimization of the objective function (Ajami et al., 2007). Although the comprehensive IBUNE framework additionally deals with the model structural uncertainty via the Bayesian Model Averaging (BMA) method, this study only compares the capacity of its input error identification part. The last one is the "BEAR" method developed in this study. This new method adds a reordering process into the "IBUNE" method to improve the accuracy of input error quantification.*"

4) The method is supposed to reduce "the potential search space for input errors" (L.360). I wonder whether this is the objective quantification of input uncertainty? Isn't it rather a comprehensive assessment of the errors and noise associated to input error and not searching in a sub-space of already collected errors and then selecting the one that fits best during predictions?

We apologize for the lack of clarification. We will add more explanations in the revision. Here "reduce the potential search space for input errors" (L.360) is because "*In a continuous sequence of data, the potential error values have an infinite number of combinations, while the error rank has limited combinations, dependent on the data length. For example, in Table A.1, the estimated error at the 1st time step could be any value. Even under the constraint of an input error ranging from the minimized to the maximized sampled errors (i.e. [-0.29,0.16] in the 1st iteration), error magnitude estimation still has infinite possibilities due to the continuous probability distribution the error represents. In contrast, the rank is discrete, having only 20 possibilities (i.e. an integer from [1,20]). From this point of view, it is far more efficient to estimate the error rank than estimate the error value.*"

To avoid the misunderstanding the current manuscript created, we will delete "reduce the potential search space for input errors" (L.360), and change this sentence as follows:

*"The estimation focuses on the error rank rather than the error magnitude, which significantly improves the effectiveness of input error quantification."*

5) Generally, a thorough discussion on the used error distributions is missing, e.g. why is a bias of 0.2 in the error function assigned without further discussion (l.211)

Thanks for your comments. We will add a clarification as follows:

*"If the input errors are estimated based on a rating curve, like the procedure in the following real case, the error distribution should be assumed as a Gaussian distribution and the mean should be 0. However, in order to test the ability of the BEAR method in wider applications, the systematic error bias equal to 0.2 has been considered in the synthetic case. An additive formulation (denoted as 'add' in ==Table 3==) is adopted to illustrate the error generation in measurements, while the multiplicative formulation (denoted as 'mul' in ==Table 3==) is specifically applied for errors induced from a log-log regression procedure, which is common in the water quality proxy processes (Rode and Suhr, 2007)."*

6) There is at least one article cited in the manuscript, that does not appear in the list of references (please see specific comments, l.190). Please assure correct referencing.

Thanks for your comments. We will correct all the missing references.

**Specific comments**

1) L. 37-38: "…estimate the residuals between the measurements and proxy values…" -> yet, measurement error is not addressed

Thanks for your comments. we will clarify this as follows:

*"In this process, the measurement errors can be ignored given the errors introduced from the surrogate process are commonly much greater than the measurement errors (McMillan et al., 2012)."*

2) L. 68: "variable" -> "scalar"– both, vectors and scalars represent variables

"variable" will be changed into "scalar".

3) Eq. 3: unnecessary, since given by equation (1)

Equation (3) will be deleted.

4) L. 84-91: repetitive, add details to the corresponding paragraph in the introduction

Thanks for your suggestion. We will move these details into the introduction, as follows:

*"The Bayesian total error analysis (BATEA) method provides a framework that has been widely used (Kavetski et al., 2006). Time-varying input errors are defined as multipliers on the input time series and inferred along with the model parameters in the Bayesian calibration scheme. It leads to a high-dimensionality formulation, which cannot be avoided (Renard et al., 2009) and restricts application to cases where event-based multipliers (the same multiplier applied to one storm event) need to be used. In the Integrated Bayesian Uncertainty Estimator (IBUNE) (Ajami et al., 2007) approach, multipliers are not jointly inferred with the model parameters, but sampled from the assumed distribution and then filtered by the constraints of simulation fitting. This approach reduces the dimensionality significantly and can be applied in the assumption of the data-based multiplier (one multiplier for one input data) (Ajami et al., 2007). However, this approach is less effective because the probability of co-occurrence of all optimal error values is very low, results in an underestimation of the multiplier variance and misidentification of the uncertainty sources (Renard et al., 2009). From the above, a new strategy should be developed to avoid high dimensional computation and meanwhile ensure the accuracy of error identification."*

5) L. 92: "innovation" -> rather "introduction" or simple "The secant method" as chapter header – the innovation was made before

We agree with the suggestion. The section titled *"innovation"* will be changed to *"introduction"*.

6)  L. 98-99: Rank definition and concept -> requires further explanation

We will add further explanations, as follows:

*"Here, the rank is defined as the order of any individual value relative to the other sampled values and determines the relative magnitude of each error in all data errors. For example, in the 1st iteration in* Figure A 1*, the error at 15th time step, -0.29, is the smallest value among all the sampled errors, therefore, its rank is 1."*

7)  L. 128ff: it sound like in ABC the requirements on the likelihood function are looser and therefore the method is easier to apply. However, requirements are also strict but ABC allows for Bayesian inference if the likelihood function is intractable. -> Please reformulate and clarify.

Thanks for your comments. Given the BEAR algorithm could be implemented via SMC, GLUE or SCE-UA, or any common model calibration approach, and this description about ABC confuses the main contribution of the paper (i.e. the core idea of BEAR method, which is to optimize input error ranks rather than input error magnitudes) we propose recasting the implementation of our optimization algorithm via SMC.

8)  L. 132ff: Notation "OF" not explained. Overall, the introduction of ABC and SMC is not clear. Further, the motivation why SMC is used here is not given.

Thanks for your comments. "OF" here means "objective function". But according to the above reply, we propose recasting the implementation via SMC, which target is the likelihood function rather than the objective function. *"The SMC sampler is more computationally efficient than previous algorithms that have applied rejection sampling and MCMC samplers (Sisson et al., 2007, Jeremiah et al., 2011)."*

9)  L. 146: "…when 1000 proposed parameter sets…" -> is this suggested as a general approach or an arbitrary choice for this study. Please explain.

According to the reply for 7), we propose recasting the implementation via SMC. If the BEAR method is implemented using a likelihood-based calibration procedure, the proposed parameter is compared to the previous entry in the chain and there is no need to set this stop criterion, just follow traditional convergence rules.

10) L.171ff: Please replace abbreviations T, D and R by their names. With all abbreviations that follow it is hard to keep track.

Thanks for your suggestion. We will change the abbreviations into the full names in the below descriptions and related figures.

"*In this study, three methods are compared to evaluate the ability of the BEAR method in quantifying input errors. The first one is the "Traditional" method, regarding the observed input as error-free without identifying input errors (i.e. Eq. (2)), while the other two methods employ a latent variable to counteract the impact of input error and build the modified input (i.e. Eq.(4)). One of them is the "IBUNE" method, where potential input errors are randomly sampled from the assumed error distribution and filtered by the minimization of the objective function (Ajami et al., 2007). Although the comprehensive IBUNE framework additionally deals with the model structural uncertainty via the Bayesian Model Averaging (BMA) method, this study only compares the capacity of its input error identification part. The last one is the "BEAR" method developed in this study. This new method adds a reordering process to the "IBUNE" method to improve the accuracy of input error quantification..*"

11) LL. 190+196: "Sikorska et al, 2015" ! missing in references

We will add all the missing references.

12) Eq. 9: define parameter "b"

The definition is shown in Table 2. A clarification will be added:

"*where the descriptions of a and b are shown in Table 2*"

13) L. 215ff: incomplete sentence

We will complete this as follows:

"*The true output $Y^*$ is the simulated TSS concentration via BwMod corresponding to the true input $X^*$ and model parameters set as the reference values in* *Table 2*."

14) L. 229ff: "calibrated via method T,…" -> misleading explanation. Please provide a more specific explanation of the calibration process under error scenarios T, D and R

Please see the reply for 10)L. 171ff

15) L. 257: ":…the impacts of model structural error and output data error cannot be ignored." vs. L.264: ":…other sources of uncertainty can be ignored" -> sound like a contradiction, please elaborate

Thanks for your comments. We will modify the description as follows:

"*In real-life applications, the impacts of model structural error and output data error exist and may impair the implementation of the BEAR method.*"

"*As the BEAR method works well under the assumption that input uncertainty is significant, other sources of uncertainties can be ignored in comparison,*"

16) L. 282-283: "This illustrates that the impacts of other…" -> unclear phrase, please clarify and re-formulate

We will add the clarifications as follows:

"*Compared with sound estimations in the synthetic case where the modeling only suffers from the input error and parameter error, this undesirable result illustrates that the impacts of other sources of errors impair the error quantification when the prior information of input error is not accurate, regardless of the methods.*"

17) L. 291: "…could be regarded as the reference value." -> Why? Please explain.

Thanks for pointing this out. The observed streamflow from the rating curve cannot be considered as the reference value, it is just closer to the reference value than the simulated streamflow via GR4J. The explanations will be corrected as follow:

"*According to the results of the traditional method in* ==*Fig. 6*==*(a), the outputs in the "O" scenarios (in (a1) and (a2)) capture the dynamics of observed TSS concentration better than the outputs in the "S" scenarios (in (a3) and (a4)). Thus, compared with the simulated streamflow via GR4J ("S" streamflow), the observed streamflow from the rating curve ("O" streamflow) should be closer to the true input data.*"

18) L. 295-296: "…have an infinite number of combinations, while the error rank has limited combinations, dependent on data length." -> What is exactly meant here?

We will add the explanation as follows:

"*In a continuous sequence of data, the potential error values have an infinite number of combinations, while the error rank has limited combinations, dependent on the data length. For example, in Table A.1, the estimated error at the 1st time step could be any value. Even under the constraint of input error ranging from the minimized to the maximized sampled errors (i.e. [-0.29,0.16] in the 1st iteration), error magnitude estimation still has infinite possibilities due to the continuous probability distribution the error represents. In contrast, the rank is discrete, having only 20 possibilities (i.e. an integer from [1,20]). From this point of view, it is far more efficient to estimate the error rank than estimate the error value.*"

19) L. 297ff. "Compared with the IBUNE framework…" -> there is no real comparison made, please see major comments

Please see the reply to the major comment 3).

20) L. 340: "for method R, an accurate input error model can constrain the adverse impacts…" -> wasn't this the problem to begin with? Please clarify this sentence.

This point will be clarified as follows:

*"To constrain the impacts of the other sources of error, accurate prior information about the input error model is important in the BEAR method. The fixed scenarios are assumed to have more accurate prior information than inferred scenarios. In the synthetic case, fixed scenarios always produce a higher NSE of the modified input (Fig. 3(5)) and a larger correlation in the estimation error (Fig. 3(3)) than inferred scenarios. In the real case in Fig. 6, the modified inputs in fixed scenarios are closer to the streamflow observation from the rating curve than the modified inputs in inferred scenarios."*

Please also see the reply to the major comment 2).

21) L 354-355: "However, the ability of these approaches needs further discussion in systems with correlated responses." -> Please clarify – what is the exact problem and why do ARMA models fit here?

Thanks for your comments. We will add clarifications as follows:

*"The part of each residual error correlated with the previous residual errors can be represented by an autoregressive moving average (ARMA) model (Kuczera, 1983) or autoregressive (AR) model (Schaefli et al., 2007, Bates and Campbell, 2001). This correlated part is removed from the residual error and the remaining part is considered to be impacted by the input error only. Thus, the correspondence between the input error rank and the residual error part is ensured and the latter process will be the same as the application of the BEAR method in BwMod. However, the specific settings of such an approach need further discussion in systems with correlated responses, for example, in the calculation of coefficients of the ARMA or AR model since the residual error changes in each iteration of calibration."*

22) L. 358: "developed" -> "proposed" – the methods are already known but used in a way to address input error here.

We will change *"developed"* to *"proposed"*.

23) L 362: "… addresses the high dimensionality problem…" -> not shown

"Address" will be changed to "avoid" and more clarification will be added as follows:

"*The introduction of the secant method links the error rank for each input data to its corresponding residual, which avoids the high dimensionality problem resulting from calibrating all the errors as a whole*."

**Figures**

1) General: Legends in figures should be improved, e.g. in terms of colors or placing

2) General: Provide higher resolution and unify the legend (see especially Fig. 4 and 6)

3) Figure 3: please use colors that are better distinguishable (see cases "T" and "R")

Thanks for your suggestion. We will improve the quality of all the figures, including improving the resolutions and modifying the colors or placing of legends.

4) Figure 3(4): NSE = 1 is unrealistic. Please see major comments.

Thanks for your pointing it out. NSE is close to 1, not equal to 1. We will modify the demonstration to avoid this misreading. This occurs as Figure 3 shows the results of the synthetic case where the modeling only suffers from the input error and parameter error. The BEAR method is effective in isolating the input error and parameter error, which has been proved by the fact that NSE is much closer to 1. However, when the BEAR method is applied in real applications where other sources of errors will interfere, as Figure 6 shows, the fit to the output TSS observations reduces.

5) Figure 4 (c3,c4): model predictions are clearly shifted. Please elaborate on this offset.

The model applied is BwMod. When the input (streamflow) is large, the output (TSS concentration) will be reduced due to the wash-off effect. It is opposite to the hydrological model, where the large input (precipitation) will lead to a large output (discharge).

6) Figures 4 and 6: Maybe it is better to show these figures in the appendix and only present the most important subfigures in the main text

OK, we will move these two figures into Appendix.

**Tables**

1) Tables 1-1 and 1-2: The tables could be presented as additional files but are not helpful in the main article

OK, we will move these two figures into Appendix, integrating the descriptions and other figures in Appendix A to provide a more clear explanation about the BEAR method.

2) Table 3: the "fixed" scenarios in the real test case are not fixed but provide small hyperparameter ranges

Thank you for pointing this out. The small ranges will be changed into the fixed value in Table 2, as follows:

Table 1 Summary of the calibration scenarios in case studies

| Scenario in the synthetic case | Notation | Input error model in the synthetic data generation | Prior information of input error model in calibration |
|---|---|---|---|
| 1 | *add-fixed* | | $X^o = X^* + \varepsilon, \varepsilon \sim N(0.2, 0.5^2)$ |
| 2 | *add-inferred* | $X^o = X^* + \varepsilon, \varepsilon \sim N(0.2, 0.5^2)$ | $X^o = X^* + \varepsilon, \varepsilon \sim N(\mu, \sigma^2), \mu \in (-0.5, 0.5), \sigma \in (0,5)$ |
| 3 | *mul-fixed* | | $X^o = X^* \exp(\varepsilon), \varepsilon \sim N(0.2, 0.5^2)$ |
| 4 | *mul-inferred* | $X^o = X^* \exp(\varepsilon), \varepsilon \sim N(0.2, 0.5^2)$ | $X^o = X^* \exp(\varepsilon), \varepsilon \sim N(\mu, \sigma^2), \mu \in (-0.5, 0.5), \sigma \in (0,5)$ |

| Scenario in the real case | Notation | Input data source in the real case | Prior information of input error model in calibration |
|---|---|---|---|
| 1 | *O-fixed* | Observations from the rating curve (USGS database) | $X^o = X^* \exp(\varepsilon), \varepsilon \sim N(0, \sigma^2), \sigma = 0.103$ |
| 2 | *O-inferred* | | $X^o = X^* \exp(\varepsilon), \varepsilon \sim N(0, \sigma^2), \sigma \in (0,1)$ |
| 3 | *S-fixed* | Simulations from a hydrological model | $X^o = X^* \exp(\varepsilon), \varepsilon \sim N(0, \sigma^2), \sigma = 0.764$ |
| 4 | *S-inferred* | | $X^o = X^* \exp(\varepsilon), \varepsilon \sim N(0, \sigma^2), \sigma \in (0,1)$ |

**Technical corrections**

1) L. 128: double ","

Thanks, the redundant "," will be deleted.

2) L. 142: "sth" -> make "s" italic

This will be changed to be italic.

3) Eq. 8: unspecified symbol

We will remove this unspecified symbol.

4) L. 314: "q increasing until the objective: : :" -> incomplete sentence

This will be corrected as follows:

"*Considering these two points, the BEAR method set q iterations in the algorithm (Fig. 1), and q increases until a defined target is achieved .*"

**Appendix A: The illustration of the BEAR method**

Table A 1 An example illustrating the BEAR method

| row | time step $i$ | 1 | 2 | 3 | 4 | 5 | 6 | 7 | 8 | 9 | 10 | 11 | 12 | 13 | 14 | 15 | 16 | 17 | 18 | 19 | 20 |
|---|---|---|---|---|---|---|---|---|---|---|---|---|---|---|---|---|---|---|---|---|---|
| | | | | | | | | 1st iteration (random sample) | | | | | | | | | | | | | |
| 1 | sampled input error | 0.07 | -0.12 | 0.07 | 0.16 | 0.05 | .0.07 | 0.07 | -0.03 | 0.03 | -0.08 | 0.09 | -0.11 | -0.11 | -0.08 | -0.29 | 0.14 | 0.03 | -0.08 | 0.14 | -0.17 |
| 2 | input error rank $k$ | 13 | 3 | 14 | 20 | 12 | 17 | 15 | 9 | 10 | 7 | 16 | 4 | 5 | 6 | 1 | 19 | 11 | 8 | 18 | 2 |
| 3 | residual error $\varepsilon$ | -0.29 | 0.49 | -0.58 | -0.98 | -0.78 | 0.29 | -0.66 | 0.59 | -1.31 | -0.31 | -0.87 | 0.76 | 0.46 | 0.54 | 0.25 | -0.80 | -0.07 | 0.56 | -0.23 | 0.40 |
| | MSE | | | | | | | | | | 0.40 | | | | | | | | | | |
| | | | | | | | | 2nd iteration (random sample) | | | | | | | | | | | | | |
| 4 | sampled input error | -0.01 | -0.02 | 0.03 | 0.03 | -0.09 | 0.00 | -0.02 | 0.06 | 0.11 | 0.11 | -0.09 | 0.01 | -0.12 | -0.11 | 0.00 | 0.15 | -0.08 | 0.04 | -0.02 | 0.11 |
| 5 | input error rank $k$ | 9 | 6 | 14 | 13 | 3 | 10 | 8 | 16 | 17 | 18 | 4 | 12 | 1 | 2 | 11 | 20 | 5 | 15 | 7 | 19 |
| 6 | residual error $\varepsilon$ | -0.13 | 0.23 | -0.43 | -0.41 | -0.21 | 0.70 | -0.23 | 0.09 | -1.88 | -1.52 | 0.20 | 0.17 | 0.53 | 0.60 | -0.43 | -0.72 | 0.36 | 0.12 | 0.47 | -0.82 |
| | MSE | | | | | | | | | | 0.47 | | | | | | | | | | |
| | | | | | | | 3rd iteration (updating the error rank via the secant method) | | | | | | | | | | | | | | |
| 7 | calculated pre-rank $K$ | 5.8 | 8.7 | 14.0 | 8.0 | -0.3 | 22.0 | 4.3 | 17.3 | -6.1 | 4.2 | 6.2 | 14.3 | 31.3 | 42.0 | 4.7 | 29.0 | 10.0 | 16.9 | 14.4 | 7.6 |
| 8 | ranked rank $k$ | 6 | 10 | 12 | 9 | 2 | 17 | 4 | 16 | 1 | 3 | 7 | 14 | 19 | 20 | 5 | 18 | 11 | 15 | 13 | 8 |
| | | | | | | 3rd iteration (reordering errors according to the updated error ranks) | | | | | | | | | | | | | | | |
| 9 | reordered input error | -0.02 | 0.00 | 0.01 | -0.01 | -0.11 | 0.11 | -0.09 | 0.06 | -0.12 | -0.09 | -0.02 | 0.03 | 0.11 | 0.15 | -0.08 | 0.11 | 0.00 | 0.04 | 0.03 | -0.02 |
| 10 | residual error $\varepsilon$ | -0.23 | 0.20 | -0.34 | -0.24 | -0.12 | 0.19 | 0.14 | 0.08 | -0.40 | -0.31 | -0.22 | 0.03 | -0.17 | 0.26 | -0.09 | -0.55 | 0.11 | 0.14 | 0.27 | -0.23 |
| 11 | MSE | | | | | | | | | | 0.06 | | | | | | | | | | |

The implementation of the BEAR method contains two main parts: sampling the errors from an assumed error distribution and reordering them with the inferred ranks via the secant method. An example is illustrated in Table A 1 and the explanation about the specific steps is presented in the following contents.

(1) In the 1st iteration ($q=1$), the errors are randomly sampled from the assumed error distribution (row 1), and then they are sorted to get their ranks (row 2). This error series is employed to modify the input data, which corresponds to a new model simulation and model residual (row 3).

(2) Repeat the step (1) in the 2nd iteration ($q=2$) as two sets of samples are prerequisites for the updating via the secant method. The results are shown in row 4, 5 and 6. Figure A 1 demonstrates that the ranges of the error distribution are the same between the true input errors (black line) and the sampled errors (blue and green lines) as they come from the same error distribution under the condition that prior knowledge of the input error distribution is correct. However, the value at each time step is not close.

(3) At the 1st time step ($i=1$) in the 3rd iteration ($q=3$), the pre-rank $K_{1,3}$ is calculated via the secant method (illustrated as the following equation). The details are demonstrated in red boxes.

$$K_{1,3} = k_{1,2} - \varepsilon_{1,2}^{p} \frac{k_{1,2} - k_{1,1}}{\varepsilon_{1,2}^{p} - \varepsilon_{1,1}^{p}} = 9 - (-0.13) \frac{9-13}{-0.13 - (-0.29)} = 5.8$$

(4) Repeat the step (3) for all the time steps. The calculated pre-ranks are shown in row 7.

(5) Sort all the pre-ranks to get the integrity error rank (row 8).

(6) According to the updated error ranks (row 8), the sampled errors in the 2nd iteration (row 4) are reordered. The example for the 1st time step is demonstrated in black boxes. The error rank at 1st time step is updated as 6, and the rank 6 corresponds to the error value -0.02 in 2nd iteration. Therefore, -0.02 is the input error at the 1st time step in the 3rd iteration. Following this example, the sampled errors at all the time steps are reordered. The results are shown in row 9. Figure A 2 demonstrates that after reordering the errors with the inferred ranks, the estimated errors are much close to the true input error.

(7) The reordered input error will lead to a new input data, a new model simulation and a new model residual. The residual error is shown in row 10.

(8) If a defined target about the residual error is achieved, the input error estimation is accepted; Otherwise, $q=q+1$, repeat step (3)~(7) until $q$ is larger than the maximum numbers of iteration $Q$.

[Figure]

Figure A 1 Demonstration of the input error estimation in Table A 1 at the 1st and 2nd iteration where the input errors are randomly sampled

[Figure]

Figure A 2 Demonstration of the input error estimation in Table A 1 at the 3rd iteration where the input errors are reordered according to the updated error ranks

---

## Author Comment (AC2) · 22 Mar 2021

**Response to Reviewer #2:**

This paper developed a new algorithm called BEAR for accurate quantification of input errors in water quality modeling. The precondition of the BEAR algorithm is that the input uncertainty should be dominant and that the prior information of the input error model can be estimated. Results of both synthetic data and observed data indicated the efficiency of the algorithm. Overall, the paper is well rewritten and the topic is suitable for the journal. However, the following issues should be further explained and clarified before its submission:

We thank the reviewer for the overall positive assessment of the manuscript and helpful comments, which have helped to improve our study. We have responded to each point in turn in the following sections. The comments from the reviewer are provided in blue text and our responses are organized point-by-point in black text. The manuscript text after changes is shown in "*black italics*" and the equation and section number are shown in yellow highlight.

It should be noted that the method name will change from the "Bayesian error analysis with reshuffling" into "Bayesian error analysis with reordering". This is based on suggestions by one of the reviewers, as the word "shuffling" implies randomness in the reordering, while the reordering in our method is determined by the model residual error. The term "reordering" better reflects the deterministic nature of error quantified via this new method. Besides, the abbreviations of methods (T, D, R) will be changed to the full names (Traditional, IBUNE, BEAR).

1) There have been many studies focusing on the uncertainty of input data errors for hydrologic modelling, and many methods including Bayesian algorithm can be used for handling the issue. However, the gap between previous studies and this study was not explained clearly in the Introduction. The motivation of this study should be clearly clarified.

Thanks for your suggestion. The research gap and motivation will be modified in the Introduction as follows:

"*Input uncertainty can lead to bias in parameter estimation in water quality modeling (Chaudhary and Hantush, 2017, Kleidorfer et al., 2009, Willems, 2008). Improved model calibration requires isolating the input uncertainty from the total uncertainty. However, the precise quantification of time-varying input errors is still challenging when other types of uncertainties are propagated*

*through to the model results. In hydrological modeling, several approaches have been developed to characterize time-varying input errors, and these may hold promise for application in WQMs. The Bayesian total error analysis (BATEA) method provides a framework that has been widely used (Kavetski et al., 2006). Time-varying input errors are defined as multipliers on the input time series and inferred along with the model parameters in the Bayesian calibration scheme. It leads to a high-dimensionality formulation, which cannot be avoided (Renard et al., 2009) and restricts application to cases where event-based multipliers (the same multiplier applied to one storm event) need to be used. In the Integrated Bayesian Uncertainty Estimator (IBUNE) (Ajami et al., 2007) approach, multipliers are not jointly inferred with the model parameters, but sampled from the assumed distribution and then filtered by the constraints of simulation fitting. This approach reduces the dimensionality significantly and can be applied in the assumption of the data-based multiplier (one multiplier for one input data) (Ajami et al., 2007). However, this approach is less effective because the probability of co-occurrence of all optimal error values is very low, resulting in an underestimation of the multiplier variance and misidentification of the uncertainty sources (Renard et al., 2009). From the above, a new strategy should be developed to avoid high dimensional computation and meanwhile ensure the accuracy of error identification.*"

2) More detailed steps about how to use the BEAR algorithm should be explained. Besides, the advantages of the BEAR algorithm compared with conventional methods should be more clearly clarified for making clear understanding from readers.

Thanks for your suggestion. The detailed steps of the BEAR method will be added in Appendix A (see the following Appendix A), and an illustration example will be moved from the methodology part to Appendix A to make the explanation more clear. In addition, the comparison with conventional methods will be clarified as follows:

"*The application of the BATEA framework is limited by high dimension computation (Renard et al., 2009). In quantifying the data-varying errors (rather than the event-varying errors in the study of BATEA (Kavetski et al., 2006)), the computational dimension is easily excessive and the BATEA probably becomes impractical (Haario et al., 2005). Therefore, the BATEA method is not considered in the comparison. In this study, three methods are compared to evaluate the ability of the BEAR method in quantifying input errors. The first one is the "Traditional" method, regarding*

*the observed input as error-free without identifying input errors (i.e. Eq. (2)), while the other two methods employ a latent variable to counteract the impact of input error and build the modified input (i.e. Eq.(4)). One of them is the "IBUNE" method, where potential input errors are randomly sampled from the assumed error distribution and filtered by the minimization of the objective function (Ajami et al., 2007). Although the comprehensive IBUNE framework additionally deals with the model structural uncertainty via the Bayesian Model Averaging (BMA) method, this study only compares the capacity of its input error identification part. The last one is the "BEAR" method developed in this study. This new method adds a reordering process into the "IBUNE" method to improve the accuracy of input error quantification.*"

3) Actually, the availability of prior knowledge of the input data error is important for modelling, but is also a difficult issue. It may be not enough only mentioning this issue in Conclusion. At least more discussions and the potential solutions should be provided.

The reviewer raised an important point. The discussion about this will be added in Section 4.2:

"*The availability of prior information of the input error relies on the studies about benchmarking the observational errors of the water quality data and hydrologic data. When the prior information is not available, the selection of the proper input error model is important. Comparing the error parameter estimations in Figure 3, the $\mu$ and $\sigma$ estimations are less biased from the reference values in add-inferred scenario than in mul-inferred scenario. It illustrates that the compensating effect between the input error and parameter error is weaker in the additive form of the input error. However, this is probably related to the specific model structure, as exponent b in BwMod has a stronger interaction with the multiplied errors than the additive errors. Thus, more comprehensive comparisons should be taken to explore the capacity of different input error models in different model applications.*"

4) The quality of some Figures in the manuscript should be improved to make all information clear.

Thanks for pointing this out. We will improve the quality of all the figures, including improving the resolutions and modifying the colors or placing of legends.

Appendix A: The illustration of the BEAR method

Table A 1 An example illustrating the BEAR method

| row | time step $i$ | 1 | 2 | 3 | 4 | 5 | 6 | 7 | 8 | 9 | 10 | 11 | 12 | 13 | 14 | 15 | 16 | 17 | 18 | 19 | 20 |
|---|---|---|---|---|---|---|---|---|---|---|---|---|---|---|---|---|---|---|---|---|---|
| | | | | | | | | 1st iteration (random sample) | | | | | | | | | | | | | |
| 1 | sampled input error | 0.07 | -0.12 | 0.07 | 0.16 | 0.05 | .0.07 | 0.07 | -0.03 | 0.03 | -0.08 | 0.09 | -0.11 | -0.11 | -0.08 | -0.29 | 0.14 | 0.03 | -0.08 | 0.14 | -0.17 |
| 2 | input error rank $k$ | 13 | 3 | 14 | 20 | 12 | 17 | 15 | 9 | 10 | 7 | 16 | 4 | 5 | 6 | 1 | 19 | 11 | 8 | 18 | 2 |
| 3 | residual error $\varepsilon$ | -0.29 | 0.49 | -0.58 | -0.98 | -0.78 | 0.29 | -0.66 | 0.59 | -1.31 | -0.31 | -0.87 | 0.76 | 0.46 | 0.54 | 0.25 | -0.80 | -0.07 | 0.56 | -0.23 | 0.40 |
| | MSE | | | | | | | | | | 0.40 | | | | | | | | | | |
| | | | | | | | | 2nd iteration (random sample) | | | | | | | | | | | | | |
| 4 | sampled input error | -0.01 | -0.02 | 0.03 | 0.03 | -0.09 | 0.00 | -0.02 | 0.06 | 0.11 | 0.11 | -0.09 | 0.01 | -0.12 | -0.11 | 0.00 | 0.15 | -0.08 | 0.04 | -0.02 | 0.11 |
| 5 | input error rank $k$ | 9 | 6 | 14 | 13 | 3 | 10 | 8 | 16 | 17 | 18 | 4 | 12 | 1 | 2 | 11 | 20 | 5 | 15 | 7 | 19 |
| 6 | residual error $\varepsilon$ | -0.13 | 0.23 | -0.43 | -0.41 | -0.21 | 0.70 | -0.23 | 0.09 | -1.88 | -1.52 | 0.20 | 0.17 | 0.53 | 0.60 | -0.43 | -0.72 | 0.36 | 0.12 | 0.47 | -0.82 |
| | MSE | | | | | | | | | | 0.47 | | | | | | | | | | |
| | | | | | | | 3rd iteration (updating the error rank via the secant method) | | | | | | | | | | | | | | |
| 7 | calculated pre-rank $K$ | 5.8 | 8.7 | 14.0 | 8.0 | -0.3 | 22.0 | 4.3 | 17.3 | -6.1 | 4.2 | 6.2 | 14.3 | 31.3 | 42.0 | 4.7 | 29.0 | 10.0 | 16.9 | 14.4 | 7.6 |
| 8 | ranked rank $k$ | 6 | 10 | 12 | 9 | 2 | 17 | 4 | 16 | 1 | 3 | 7 | 14 | 19 | 20 | 5 | 18 | 11 | 15 | 13 | 8 |
| | | | | | | 3rd iteration (reordering errors according to the updated error ranks) | | | | | | | | | | | | | | | |
| 9 | reordered input error | -0.02 | 0.00 | 0.01 | -0.01 | -0.11 | 0.11 | -0.09 | 0.06 | -0.12 | -0.09 | -0.02 | 0.03 | 0.11 | 0.15 | -0.08 | 0.11 | 0.00 | 0.04 | 0.03 | -0.02 |
| 10 | residual error $\varepsilon$ | -0.23 | 0.20 | -0.34 | -0.24 | -0.12 | 0.19 | 0.14 | 0.08 | -0.40 | -0.31 | -0.22 | 0.03 | -0.17 | 0.26 | -0.09 | -0.55 | 0.11 | 0.14 | 0.27 | -0.23 |
| 11 | MSE | | | | | | | | | | 0.06 | | | | | | | | | | |

The implementation of the BEAR method contains two main parts: sampling the errors from an assumed error distribution and reordering them with the inferred ranks via the secant method. An example is illustrated in **Error! Reference source not found.** and the explanation about the specific steps is presented in the following contents.

(1) In the 1st iteration ($q$=1), the errors are randomly sampled from the assumed error distribution (row 1), and then they are sorted to get their ranks (row 2). This error series is employed to modify the input data, which corresponds to a new model simulation and model residual (row 3).

(2) Repeat the step (1) in the 2nd iteration ($q$=2) as two sets of samples are prerequisites for the updating via the secant method. The results are shown in row 4, 5 and 6. **Error! Reference source not found.** demonstrates that the ranges of the error distribution are the same between the true input errors (black line) and the sampled errors (blue and green lines) as they come from the same error distribution under the condition that prior knowledge of the input error distribution is correct. However, the value at each time step is not close.

(3) At the 1st time step ($i$=1) in the 3rd iteration ($q$=3), the pre-rank $K_{1,3}$ is calculated via the secant method (illustrated as the following equation). The details are demonstrated in red boxes.

$$K_{1,3} = k_{1,2} - \varepsilon_{1,2}^p \frac{k_{1,2} - k_{1,1}}{\varepsilon_{1,2}^p - \varepsilon_{1,1}^p} = 9 - (-0.13)\frac{9-13}{-0.13-(-0.29)} = 5.8$$

(4) Repeat the step (3) for all the time steps. The calculated pre-ranks are shown in row 7.

(5) Sort all the pre-ranks to get the integrity error rank (row 8).

(6) According to the updated error ranks (row 8), the sampled errors in the 2nd iteration (row 4) are reordered. The example for the 1st time step is demonstrated in black boxes. The error rank at 1st time step is updated as 6, and the rank 6 corresponds to the error value -0.02 in 2nd iteration. Therefore, -0.02 is the input error at the 1st time step in the 3rd iteration. Following this example, the sampled errors at all the time steps are reordered. The results are shown in row 9. **Error! Reference source not found.** demonstrates that after reordering the errors with the inferred ranks, the estimated errors are much close to the true input error.

(7) The reordered input error will lead to a new input data, a new model simulation and a new model residual. The residual error is shown in row 10.

(8) If a defined target about the residual error is achieved, the input error estimation is accepted; Otherwise, $q=q+1$, repeat step (3)~(7) until $q$ is larger than the maximum numbers of iteration $Q$.

[Figure]

Figure A 1 Demonstration of the input error estimation in **Error! Reference source not found.** at the 1st and 2nd iteration where the input errors are randomly sampled

[Figure]

Figure A 2 Demonstration of the input error estimation in **Error! Reference source not found.** at the 3rd iteration where the input errors are reordered according to the updated error ranks

---

## Author Comment (AC3) · 22 Mar 2021

**Response to Reviewer #3:**

We have noted the reviewer's comprehensive remarks, and the following provides our response to each point. We appreciate the level of detail that the reviewer has gone into, and would like to thank them for several of their suggestions which will significantly improve how we convey the proposed approach.

However, we think many of the comments have arisen as the reviewer may not be familiar with existing approaches to estimate errors in inputs for these types of models (BATEA, IBUNE), just as the reviewer has mentioned in the comments. We hope our response and the proposed changes strike the right balance between addressing the reviewer's comments and recognising the past work in this area which we don't wish to recreate in our manuscript.

**Response to Reviewer #3:**

Complex patterns in input uncertainty such as spatial or temporal error correlations are an important topic in environmental science. In their present study, the authors seek to explore the ubiquitous issue of complex input uncertainty structures by proposing a novel method called Bayesian error analysis with reshuffling (BEAR). The proposed method is based on sampling an estimated input error and subsequently sorting the resulting realizations in an order which reduces residual mismatch to the observations. The authors then proceed to demonstrate the performance of their algorithm for a synthetic and a real case and compare its performance to a number of alternative setups.

I find the approach a very interesting and creative idea, and always appreciate it if someone takes the risks inherent to exploring a new methodological idea. Unfortunately, I have some reservations concerning its theoretical justifiability, which I hope the authors can address. Failing that, there might also be alternative ways to achieve similar effects which might stand on more robust theoretical foundations. Concerning these suggestions and the method itself, I have the following (major) comments:

**Major comments:**

1. Theoretical foundations: A key step of the approach is sorting input error realizations to reduce the residual mismatch between model predictions and system observations. I fear that this compromises the randomness of the error realizations, with potential consequences for the validity

of the Bayesian inference that are difficult to predict. Rigorously deriving this algorithm from more basic theoretical foundations might better illustrate the consequences of the authors' assumptions for Bayesian inference. If the authors do not have this expertise themselves (not to worry: few people in the environmental sciences do), I recommend seeking the help of the local statistics department – they are often keen to help. It seems an unfortunate truth of Bayesian statistics that interesting ideas for algorithms which sound nice on paper often tend to violate the Bayesian framework in unforeseen ways.

Thanks for your comments. We propose including a section that will derive this algorithm from the theoretical foundation of Bayesian inference (see Appendix B). We believe the remarks of the reviewer here are because the original manuscript implements an approximate Bayesian approach to model calibration which confuses the main contribution of the paper (which is to optimize input error ranks, rather than input error magnitudes). To address this, we propose recasting the implementation of our algorithm via SMC. The BEAR algorithm could be implemented via SMC, or GLUE, or SCE-UA, or any common model calibration approach, simply by optimising the latent-input errors on the error rank rather than error magnitude to realise the deterministic relationship between the residual error and the input error. It should be noted that we don't get the posterior distribution of the error ranks, we aim to optimise the error ranks by finding the deterministic relationship between the input error rank and the model parameters (see Appendix B).

**2. Improving future performance:** While I am not intimately familiar with the alternative methods (BATEA, IBUNE) referenced by the authors, improving future model performance is a common motivation for learning complex uncertainty structures in hydrological models. The approach in this manuscript, however, requires a concurrent time series of observations. As a consequence, it does not really improve the error model itself and hence offers little value for improving future predictions. On the other hand, attempting to learn time-varying bias or correlation structures in the input errors – admittedly a sometimes computationally formidable task – can increase the likelihood of future predictions substantially. Such approaches would also be significantly easier to justify. I would recommend mentioning this limitation for the BEAR

algorithm in the manuscript, or – better yet – either explore or propose ways on how this limitation could be circumvented.

We think the reviewer's comments here may be due to a misunderstanding of how errors manifest in these types of hydrologic-water quality models, and how these models are used for predictions/scenarios vs in calibration. Additionally, we must clarify that water quality model applications such as the ones illustrated here rely on observed inputs alone, with observed response data being used only in model calibration. A better calibration of the water quality model is what is the intended outcome from our study.

The reviewer is correct in that the approach we propose does not 'improve' the error model but it improves the water quality model specification to now have parameters closer to what would be achieved under no error conditions. This is because it is assumed that the error model is known a priori and the overall objective is not to improve on this. We believe this is not an unreasonable assumption, as the model inputs are derived using a hydrologic model or rating curve whose error can be derived independently of the water quality model implementation. These independent observations provide insight to the distribution of input errors that can then be leveraged in the model calibration.

By identifying the order of a sample of input errors from this distribution, the model calibration results in model parameters that are closest to the 'truth', as can be seen in our synthetic case studies where the true parameters are known (Figure 3). **The overall goal in identifying proper input errors is then to ultimately improve our estimate of the model parameters**, so that the model can be more effectively used for scenario analysis (where we may know the hydrologic regime of a catchment in a hypothetical future), for forecasting under the assumption of perfect inputs (where the driving hydrologic forecast is independently obtained via a numerical weather prediction and a hydrologic model) or for regionalisation of the water quality model (where the model is transferred to a catchment without data). In all of these cases, an ideal model has unbiased parameter estimates. This is our goal in identifying the optimal ranks of input errors via BEAR, not to use the model for predictions with input data suffering the same errors.

3. The influence of other error types: It seems the authors focussed most of their attention on input uncertainty. While I agree that input uncertainty can play a very important role, the influence of

observation errors and model structural uncertainty plays a substantial role as well. In this study, the authors assumed both the model and the output observations were error-free – and derived their algorithm accordingly –, but in practice these assumptions are virtually never met. I would encourage the authors to explore how (if at all) their algorithm can avoid surrogacy effects in the presence of observation and model errors. (What I mean by "surrogacy effects" is that the algorithm's adjustments to the input error realizations also 'soak up' [and consequently mask] errors in the output observations and the model itself in a bid to reduce the output residuals. This is of course undesirable.) See also my penultimate minor comment below.

The reviewer has raised here a very important point, and we completely agree with the issue of compounding sources of error and potential 'surrogacy' effects.

We have aimed to address this in our case by implementing a real data case which likely has issues of error surrogacy, to identify how our approach works in a 'real' setting. We believe this helps balance our discussion of the approach and its potential limitations. To additionally ensure we do not oversell our method given other sources of error, we also propose including a synthetic example in Appendix C that examines how the method performs under increasing error on the model calibration data. We hope this will help properly identify the usefulness of our approach but its potential limitations depending on the settings in which BEAR might be implemented.

4. Deterministic functions as an alternative: If the authors find that their algorithm might be based on flawed assumptions (following a more detailed theoretical derivation or investigation of the distribution it effectively samples from), the authors might wish to explore possible alternatives. If reducing the output residuals through adjustments to the input data remains the goal, a safer route might be to couple a deterministic input pre-treatment routine with the WQM and add its parameterization to the WQM parameter vector. Functionally, this pre-treatment routine can simply be interpreted as part of the deterministic model. Choices for this pre-treatment routine could be, for example, a one-dimensional spline which re-scales input magnitudes non-linearly (see the attached Figure 1, for an example with three extra parameters). More complex function choices might allow the consideration of lag, temporal or spatial correlations, etc. This would have the additional advantage over the BEAR framework that this pretreatment routine could also improve future predictions, assuming that it compensated true bias and is not overfitted. This

comment is not a request for change, but a hopefully constructive suggestion for alternatives so that the authors might salvage some of their work in case it would turn out theoretically indefensible.

[Figure]

**Fig. 1.** Example of a non-linear re-scaling with a spline defined by three control points (red dots). You could use such a spline to scale the input values non-linearly.

We appreciate the thought that the reviewer has put into this comment, and we have considered this concept carefully, however, respectfully we do not see it as a viable pathway for the types of errors that we see in inputs to WQMs. The approach you suggested here is a non-linear bias correction to the input data which will correct systematic biases. In our case, it is the randomness of the observational error that means individual errors can't be explicitly identified prior to modeling.

This random error does lead to biased model parameters in calibration. We present here a synthetic case to demonstrate such, where the model inputs are specified as $X^o = X^* + \boldsymbol{\varepsilon}_X, \boldsymbol{\varepsilon}_X \sim N(0, 0.2^2)$, the observed outputs are specified as $Y^o = Y^* = M(X^*, \theta^*)$, and the objective function is selected as the MSE of the residual errors in log space. The mean of Gaussian distribution of the input error is set as 0, where the observed inputs are not systematically biased from the synthetic true inputs (Figure R1). According to the approach suggested by the reviewer, the observed inputs do not need

modification, or it makes no difference before and after the suggested approach. However, in the comparison of different methods in Figure R.1, if we do not deal with the stochastic errors in input data, like the traditional method (the same as the suggested approach in this case), the model parameters will be biased from their reference values. In contrast, the BEAR method can effectively identify the stochastic errors, beyond the systematic bias of the input data, as the BEAR method reorders the error rank according to the residual error, and alleviates the impacts of both stochastic and systematic errors on the model parameters.

We additionally note that we are not proposing a method for rescaling errors, we are simply re-ordering their positions in the data to better correspond to deviations from model fits.

[Figure]

Figure R1 The posterior distributions of the model parameters estimated via the different methods and the scatter plot of the observed inputs vs true inputs in the synthetic case

5. BEAS instead of BEAR: This could be filed under nit-picking, but since the algorithm's name features so prominently, I chose to raise this to a major comment instead. The use of the word 'shuffling' implies randomness in the re-ordering. If I understood the authors' algorithm correctly, though, the re-ordering itself is entirely deterministic. As such, changing the name to something along the lines of "Bayesian error analysis with sorting" (BEAS) or "Bayesian error analysis with

re-ordering" (if the authors like to retain their – admittedly very nice – acronym) might better reflect its deterministic nature.

We very much appreciate the valuable suggestion here, as the reviewer is completely correct. Yes, "reordering" better reflects the deterministic nature of error quantified via this new method. Therefore, the method name will be changed into "Bayesian error analysis with reordering" to keep the BEAR acronym for short.

6. Why ABC: In the manuscript, the authors use an "Approximate Bayesian Computation via Sequential Monte Carlo" (ABC-SMC) approach. While I am not personally familiar with this approach, I struggle to see why it is necessary to resort to ABC, aside from any potential (forgive me) self-inflicted complications induced by the BEAR algorithm. The model and output variables seem pretty simple, so to my untrained eyes it is difficult to see why the formulation of an analytical likelihood should be impossible in this case. One could also cast the procedure the authors presented in Figure 1 with very few changes in terms of an MCMC routine, provided the re-ordering or error realizations ends up being statistically justifiable, of course. I would encourage the authors to provide a bit more detail on why ABC was necessary.

Thanks for the helpful suggestions here. The reviewer is completely correct that the ABC method is not necessary, and the core idea of the BEAR method (the reordering strategy in the rank estimation) can be easily applied in any other calibration algorithm, for example, MCMC and SMC algorithms. In order to clarify this point, we propose modifying the manuscript to make this clear in the algorithm.

7. Focus on a good fit: A key idea which seems to permeate the present manuscript is that it is desirable to obtain error realizations, if necessary by force (i.e., re-ordering), which match the observations as closely as possible. This is of course true, but not at all cost. Even assuming a severely mischaracterized prior input error distribution (e.g., a Gaussian with a standard deviation of 1E-10 and a mean of -1E8), one could theoretically obtain an error realization which causes the model to fit the observations perfectly if we only drew sufficiently (read: infinitely) many samples. The challenge, then, is not to find such a realization within our prior distribution (it will exist in

any distribution with sufficiently broad support), but to find a distribution from which there is a high probability to obtain such a sample. Crucially, such a distribution should be independent from future observations, and I fear that this may not be the case for the approach proposed in this manuscript. In this approach, after re-ordering, the realizations are no longer i.i.d. samples from the input distribution (similarly to how one could interpret correlated Gaussian samples merely re-ordered independent Gaussian samples, but would nonetheless be wrong in claiming that an independent Gaussian distribution is identical to a correlated Gaussian distribution). If the authors decide to pursue my request for a derivation of a theoretical foundation for their approach (see comment 1), I recommend focussing on investigating from what effective distribution they are really sampling. Considering BEAR's ability to yield a reliably good fit with seemingly arbitrary prior input error realizations, I fear that the distribution you effectively sample from may be well approximated with (for example) a Gaussian with a mean inversely obtained from the observation residuals. If this turns out to be the case, the method would be more or less equivalent to just calculating the input error residuals through an inverse method of choice by minimizing the output residuals. This would not be very useful, and I would recommend exploring one of the approaches I suggested in comment 4 instead. See also my penultimate minor comment.

The reviewer has raised an interesting point here, that the reordered errors may include features (such as autocorrelation or dependency) that are not present in the original error sample. We propose including statistical analysis of the reordered errors (e.g. ACF, PACF, correlation of the errors to modelled TSS magnitude) as a diagnostic for the real case on the potential interactions between observation error and model structural error.

A point of clarification: in each subsequent iteration of the BEAR algorithm, a new population of input errors are sampled from their a priori distribution. This means that the distribution of errors at each iteration is the same prior to reordering, i.e. the population of errors do not converge to a distribution that has different statistical features (mean, standard deviation, skewness). However, we do recognize that the reordering process provides insight into how the input errors may be ideally distributed across time that can inform how the surrogacy effect is manifested (see comment 3). For example, if the errors are highly autocorrelated this could highlight potential model structural errors if the WQM has not properly represented the storage characteristics of TSS in the catchment. We believe that including further diagnostics will help elucidate this effect.

Note that the approach does in fact rely on an appropriate prior distribution being identified describing the input error distribution (see Appendix C). If this error distribution is mischaracterized and the whole statistical features are different from the real input, the resultant outputs will not converge to the calibration data, and will lead to larger residual errors. Subsequent updates of the error values will aim to find a more appropriate sample of the input error to get a smaller residual error. In some cases, the parameter error will compensate this mischaracterized error distribution, leading to an overfitting problem. This compensation should occur on not only a single input error realization, but also the whole error distribution. In Figure C.1 of Appendix C, the standard deviation of the input error is estimated accurately, converging to the reference value when there is no interference of other sources of errors. However, if there is interference of other sources of errors, the impacts can not be avoided in any of the methods unless the method can deal with all sources of errors together.

**Specific comments:**

Line 22-25: You mention the importance of complex interactions of different error sources directly in the first paragraph but proceed to largely ignore their influence in the remaining manuscript. I think this part is important and should be discussed in greater detail in the remainder of the manuscript (particularly also the methods/theory section).

Thanks for your suggestion, we agree this was not being properly highlighted. We propose including an explanation in the methodology that emphasizes this:

*"It should be noted that the derivation of the BEAR method is based on the assumption that the model only suffers from the input error and parameter error, but other sources of error (i.e. model structural error and output observational error) can also impair the estimation of the model parameters and are inevitable in the WQM. The ability of the BEAR method is tested in the real case where the interference of other sources of error have been considered."*

Line 49-51: During this review, I have briefly glanced into the corresponding methods BATEA and IBUNE, and apparently there was quite a commentary battle between the authors over these methods (see doi:10.1029/2007WR006538 and https://doi.org/10.1029/2008WR007215), and Renard et al. 2009 noted that IBUNE may in fact not reduce dimensionality. The choice is of course ultimately up to the authors, but it might be useful to add a small comment noting that the claim of dimension reduction by IBUNE is also challenged.

In the comments from Renard et al. 2009, there are two different implementations of the IBUNE framework. The first is the same as the method D described in this manuscript, and Renard et al. 2009 argue the randomness in this approach will lead to an underestimation of the input error variance. The second implementation is the same as the BATEA approach, which can not reduce the computational dimension.

According to the IBUNE paper (Ajami et al., 2007), the authors argue that IBUNE can circumvent the high dimensionality issues by randomly sampling the multiplier to each time step from an assumed normal distribution. Therefore, we interpret the first implementation as the 'true' version of IBUNE, and provide discussion on this in the manuscript.

Line 69-75: I would be careful here. In reality, there are many different error sources, certainly not the least of which is model structural error. Calibrating (i.e., simply minimizing the residuals between simulated and observed output) in the presence of other error sources is prone to surrogacy effects, so you can never really be sure you recovered the 'true' parameters. Even making the (unrealistic) assumption that we could manage to completely remove all error sources in the model and its input, we could still only retrieve the 'true' model parameters if our inverse problem is unique. I would mention these restrictions here (only works if all error sources can be removed completely, unique inverse problem).

Thank you for pointing out the equifinality problem. We propose modifying the description here as follows:

"*where the optimal parameters $\theta^p$ will lead to the same simulation corresponding to the true values $\theta^*$ and the model residual $\varepsilon^p$ will decrease to zero. Due to equifinality, the parameter $\theta^p$ may not converge to the true parameter $\theta^*$ if the inverse problem is not unique, but both may result in the same simulation. In this study, these parameters are called the ideal model parameters, also denoted as $\theta^*$ due to the same impacts of true model parameters.*"

Line 70-73: In Equation (2), the variables $Y^o$ and $Y^s$ are used without any introduction. I assume both variables stand for the observed and simulated output. Please introduce these variables.

Thanks for pointing this out. We propose clarifying this as follows:

"where $Y^s$ is the output simulated from the model $M$ corresponding to the observed input $X^o$ and model parameter $\theta^c$, and the observed output $Y^o$ is assumed without observational errors in the derivation, thus can be denoted as $Y^*$."

Line 76-77: A critical thing here is that $\varepsilon$ is previously introduced as an error, which implies that you consider it to be a random variable. However, subtracting a (say) Gaussian random variable from another Gaussian variable with the same properties does not reduce variance to zero but actually doubles it (if both random variables are independent). What you seem to have in mind here only works if $\varepsilon$ and $\varepsilon^p$ have identical properties and are perfectly correlated (note that this

implies a lot more than just sharing the same statistical moments!), or if you are talking about error realizations. You should clarify this. This relates to major comment 7. You can only create this perfect correlation if you can somehow extract the error realizations of $\varepsilon$ (which only works under the assumption that you already have the input samples, that there are no other errors, and that the inverse problem is unique). Consequently, I fear that you may create/mimic this perfect correlation by implicitly solving an inverse problem, which would make the proposed method not very useful.

We agree with your argument that "Subtracting a (say) Gaussian random variable from another Gaussian variable with the same properties does not reduce variance to zero but actually doubles it (if both random variables are independent)". This is also the reason why IBUNE (method D in this manuscript) does not improve the accuracy of the error identification. The claim in this manuscript is "*If the equivalence between $\varepsilon_X$ and $\varepsilon_X^p$ can be ensured for each data point, the modified input $X^p$ then becomes the same as the true value $X^*$. The proposed calibration (Eq. (4)) will result in an ideal calibration (Eq. (1)), where the optimal parameters $\theta^p$ will lead to the same simulation corresponding to the true values $\theta^*$ and the model residual $\varepsilon^p$ will decrease to zero. Thus, the precise identification of the input error series will result in the ideal model parameters and minimized residual error, which is the aim of model calibration considering the input error quantification.*" As the reviewer notes, we are referring here to the error realizations, not the distribution of errors.

We also agree that the perfect identification of input error series and model parameters only happens when "there are no other errors, and that the inverse problem is unique". We propose adding clarification as follows: "*It should be noted that the derivation of the BEAR method is based on the assumption that the model only suffers from the input error and parameter error, but other sources of error (i.e. model structural error and output observational error) can also impair the estimation of the model parameters and are inevitable in a WQM. The ability of the BEAR method is tested in the real case where other sources of error have been considered.*"

Considering the unique inverse problem, we propose changing the description here to "*Due to equifinality, the parameter $\theta^p$ may not converge to the true parameter $\theta^*$ if the inverse problem is not unique, but both may result in the same simulation. In this study, these parameters are called the ideal model parameters, also denoted as $\theta^*$ due to the same impacts of true model parameters.*"

Line 79: In Equation (4), you also mark the parameters – on which this entire exercise should be conditional on – as changing due to your proposed approach. The equations you have shown so far imply that the procedure you describe here is applied after parameter calibration. During a first reading of this paper, it is not immediately clear why the calibrated parameter values should change with your proposed approach. On a second reading, it becomes evident that you do not really calibrate but sample the parameter posterior, but that this sampling process is inter-woven with your BEAR routine, hence the parameter values are also affected. I recommend commenting on this already here to save your readers some confusion. Maybe it would also help not to talk about calibration at all in this context.

Apologies for the lack of clarification here- in fact, the model calibration is the ultimate goal in properly identifying the model errors. The reordering step is implemented after one set of the model parameters has been sampled. Corresponding to this set of model parameters, the optimal ranks of input errors are inferred via the secant method. Although the sampled errors have been reordered, they still follow the assumed error distribution and the overall statistical characteristics remain unchanged. In Equation (4), with the constraint of the input error distribution, if the calibrated model parameters are far from the true parameters, the residual error cannot be reduced effectively. If the model parameters are close to the ideal model parameter, the input error can be identified precisely via the secant method, and the model residual error will approach zero.

We propose clarifying the model calibration goal in the paper so that the intent is clear.

Line 81-83: I would be very careful with this statement. This only happens if the model parameters $\theta p$ and the input error residuals $\varepsilon_X^p$ cannot compensate each other, i.e. if there is only a single, unique combination of parameters and errors which yields zero residual. If you have a Pareto front along which different values of $\theta p$ and input error residuals $\varepsilon_X^p$ yield zero residual, you cannot be certain that you have correctly identified the 'true' parameter values and the 'true' input error, even in a scenario where no other errors/uncertainties exist. In addition to this, my reservations concerning other error types raised in major comment 3 also apply. I would recommend changing this statement accordingly and exploring its consequences for your algorithm in greater detail.

We appreciate your comment. Yes, the input error and model parameter error might compensate each other, which leads to a Pareto front along which different values of θp and input error residuals yield zero residual. In the manuscript, we compare two types of input error models: additive formulation ( $X^o = X^* + \varepsilon, \varepsilon \sim N(0.2, 0.5^2)$ , *add* in Table 3) and multiplicative formulation ( $X^o = X^* \exp(\varepsilon), \varepsilon \sim N(0.2, 0.5^2)$ , *mul* in Table 3). According to synthetic cases, we find the multiplicative formulation is more likely to compensate the model parameter error, but the compensation effect will reduce with accurate prior information on the input error model. Probably because the impacts of error in multiplicative formation is similar as *b* in Equation 9, or the nature of multiplicative formulation is more likely to change with the model parameter than additive formulation, due to having more flexibility. In sum, the compensate effect can be reduced through the proper selection of the input error model since it affects the identification accuracy. But this problem is common in all the methods considering input error, and is not special in the BEAR method.

Line 104-106: I would rephrase this a bit, because following the procedure you outlined in Figure 1 (a very nice schematic, by the way), it is not only two steps: you sample the error once, then iterate over a large number of re-ordering steps until you find an order which minimizes your output residuals. This could do with some clarification.

Thanks for your suggestion. We propose rephrasing the sentence as:

"*This new approach, referred to as the Bayesian error analysis with reordering (BEAR) method, contains two parts: sampling the errors from the estimated error distribution to maintain the overall statistical characteristics of input errors and reordering these sampled errors via the secant method for a few iterations until the input error order can be optimized to achieve the defined target about the residual error.*"

Line 115-116: If I understood your explanations here correctly, maybe an easier way of explaining what you are doing is that you sort your updated error ranks, then assign to each of them a new integer rank based on its position in the sorted list. This might be easier than trying to explain this procedure with scaling.

Thanks for your suggestion. We propose modifying this sentence as follows:

"*Sorting $k_{i,q}$ in all the ranks $k_{i,q}(i=1,...,n)$ can address this problem by effectively assigning to each of them a new integer rank based on its position in the sorted list.*"

Line 125-131: As mentioned in major comment 6, please devote some space to explain why the models you use in the following necessitate the use of ABC. Even after going through the manuscript a few times, I struggle to see why standard Bayesian approaches would be impossible to use. At the risk of evoking the anger of our ABC-focussed colleagues: direct is usually better than approximate. It also does not become clear in the manuscript why an ensemble-based approach is used – couldn't the same procedure be implemented in an MCMC-style acceptance/rejection algorithm? If there are some ABC reasons for requiring an ensemble, it you might also want to explain why and how it is used.

We apologize for the lack of clarification. The ABC is not necessary for the BEAR method to be implemented. We admit there is no difference in the calibration results between the standard Bayesian approaches and the ABC approach for this case study. In order to clarify this point, we propose modifying the manuscript to make this clear in the algorithm, please also see the proposed Appendix B.

Line 132-136/Figure 1: This explanation of the method is very short, and essentially only explains that you approach the posterior distribution iteratively through a number of intermediate steps, but not how this is achieved exactly. Figure 1 provides more information and suggests some sort of acceptance/rejection scheme depending on whether your procedure can reduce the error residuals below a certain threshold, but the nature of the posterior distributions from which new parameter values are drawn remains undefined. You also seem to update an input error parameter ηx, which seemingly contradicts statements you made suggesting the input errors are sampled from a pre-estimated distribution (Line 10, Line 193-194). This step is also never mentioned in the text itself up to this point – you only mention that you estimate the input error distribution's hyperparameters much later. The text also frequently mentions 'populations', which evoke the idea of an ensemblebased method, but none of the steps mentioned in the text so far actually seem to require an ensemble. Please provide some more (written) detail about how your algorithm functions exactly.

Thanks for your comments, please see the following:

1) "The nature of the posterior distributions from which new parameter values are drawn remains undefined". The steps about generating a new set of the model parameters from the previous posterior parameters are demonstrated in the below figure describing the SMC algorithm. Also, we propose adding clarification in the manuscript, as follows:

"*The details of sampling a new set of model parameters from the posterior distribution of the previous population can refer to the study of (Bonassi and West, 2015).*"

1. Initialize threshold schedule $\epsilon_1 > \cdots > \epsilon_T$
2. Set $t = 1$
    For $i = 1, \ldots, N$
    – Simulate $\theta_i^{(1)} \sim p(\theta)$ and $x \sim p(x|\theta_i^{(1)})$ until $\rho(x, x_{obs}) < \epsilon_1$
    – Set $w_i = 1/N$
3. For $t = 2, \ldots, T$
    For $i = 1, \ldots, N$
    – Repeat:
        Pick $\theta_i^*$ from the $\theta_j^{(t-1)}$'s with probabilities $w_j^{(t-1)}$,
        draw $\theta_i^{(t)} \sim K_t(\theta_i^{(t)}|\theta_i^*)$ and $x \sim p(x|\theta_i^{(t)})$;
    until $\rho(x, x_{obs}) < \epsilon_t$
    – Compute new weights as

$$w_i^{(t)} \propto \frac{p(\theta_i^{(t)})}{\sum_j w_j^{(t-1)} K_t(\theta_i^{(t)}|\theta_j^{(t-1)})}$$

Normalize $w_i^{(t)}$ over $i = 1, \ldots, N$

Figure 1 ABC-SMC algorithm (Bonassi and West, 2015)

2) A description about "update an input error parameter ηx" will be added, as follows.

"*In the real-life applications, the input error model can be approximated based on the studies described in the introduction, but the error parameter might not be estimated as a deterministic value accurately. In those cases, these error parameters should be considered as the hyperparameters and calibrated with the model parameters together (denoted as $\eta_x$ in Fig.1).*"

3) Here the term "population" is used as in the SMC algorithm, and it is like an ensemble-based method. We will clarify this by a better description of how the algorithm is implemented via SMC.

Line 145-148 and Line 159-162: This is just a comment towards the general "Why ABC?" discussion. It seems to me that a classic MCMC procedure would avoid the need for adjusting the acceptance threshold dynamically, as proposed parameters are always compared to the previous entry in the chain.

Yes, ABC is not necessary for the BEAR method and any MCMC procedure will avoid the need for adjusting the acceptance threshold. We propose recasting the implementation of our optimization algorithm via SMC.

Line 154-157: I confess that this explanation is quite impenetrable, and probably causes more confusion here than it does good. I recommend restructuring this explanation or removing it altogether. A good alternative would be to visualize this with a small figure, possibly added to the supporting information if length limitations to not permit embedding it into the main text.

Thanks for your suggestion. We will integrate an example and its illustration in the Appendix to explain the specific steps involved (see an illustration of this in the following Appendix A).

Line 192: I do not see from Equation 8 or 9 or their surrounding text how the spatial scale factors into this model. Through the Sa variable? Please clarify this.

Thanks for your comments. "*The two formulations in Bwmod were developed in a small-scale experiment (Sartor and Boyd, 1972), while in applications at the catchment scale, the conceptualized parameters largely abandon their physical meanings and the formulations can be considered a "black-box" (Bonhomme and Petrucci, 2017).*" This study attempts to simulate the sediment dynamics in the catchment scale. Thus, we say "the spatial scale is set as the catchment in this study". This will be clarified as follows:

*"This study will test the BEAR algorithm in a case of simulating the daily sediment dynamics of one catchment, thus, the time scale is typically set as daily and the spatial scale is set as the catchment."*

Line 200: In Equation 8, you do not introduce Smax and κ. Please introduce these variables as well.

The introduction of Smax and κ is shown in Table 2. We propose adding the clarification as follows:

*"where the descriptions of κ and Smax are shown in Table 2"*

Line 203: In Equation 9, you do not introduce a and Qt$^b$. Please introduce these variables as well.

Qt$^b$ should be $(Q_t)^b$, we propose modifying this as follows:

$$s(S_{a,t}) = a \cdot (Q_t)^b \cdot S_{a,t} \tag{9}$$

*"where the descriptions of a and b are shown in Table 2, and $Q_t$ is the streamflow at the catchment outlet at time t."*

Line 205: In Equation 10, you do not introduce Qt. Please introduce this variable as well.

See the above reply.

Line 209-211: For this section, there are a few assumptions which could warrant greater discussion. If the errors are normally estimated in advance based on a rating curve, why is there a constant offset of 0.2? Couldn't this systematic bias be corrected through the rating curve itself? Alternatively, if the offset is necessary because your errors are asymmetrically fat-tailed, wouldn't a different distribution (such as a scaled beta or gamma distribution) be a better choice? It is commendable to make the synthetic test case more challenging by introducing bias as well, but how would this be recognized a priori in a real test case if it wasn't already considered in the rating curve? Some more information might clarify the authors' choice of distribution for the audience.

Thanks for your comments. Yes, we agree with your statement that if the errors are normally estimated in advance based on a rating curve, the mean of the normal distribution should be 0. By including the systematic bias of 0.2 in this synthetic study we aim to test the BEAR method in wider applications, even if these are not necessarily representative of true errors. Figure 3(1) and (2) indicates that in the synthetic case only suffering from the input error and parameter error, the mean and standard deviation of input error can be estimated precisely together with the model parameters. Therefore, in situations where the input errors have no systematic bias and only a standard deviation must be calibrated, the BEAR method also works. According to your comments, we propose clarifying this as follows:

"*If the input errors are estimated based on a rating curve, like the procedure in the following real case, the mean of input error should be 0. But in order to test the ability of the BEAR method in wider applications, the systematic bias 0.2 has been considered in the synthetic case.*"

Line 224-226: This part here is a bit unclear. What I deduce from the context is that you looked at two scenarios – one, where you left the prior input error fixed, and one where you estimated the input error hyperparameters as well. I would not talk about 'conditions' in this context, but rather about 'scenarios'. If I understood your drift here correctly, I would also add a comment which puts more emphasis on the fact that you subvert one of the principal assumptions you made earlier in the second scenario (namely, that the input error distribution is a prior/pre-estimated).

Thanks for your comments. We propose modifying the description as follows:

"*Given the description in the introduction, the input error can be pre-estimated in some studies. While in other cases, the prior information about the input error cannot be estimated or the accuracy is in question. Therefore, two scenarios concerning the prior information of error parameters (i.e. $\sigma$ and $\mu$) have been considered: one is fixed as the reference values (denoted as 'fixed' in Table 3), the other one is estimated as the hyperparameters with the model parameters (denoted as 'inferred' in Table 3.*"

Line 232-234: I would recommend to critically re-examine this part in the light of major comment 7. The high correlation of scenario R with the realizations of the synthetic true error series – which

are supposed to be realizations from an independent Gaussian distribution – might be reason for concern, as they suggest that you might be implicitly solving an inverse problem for the input error residuals. This would have little to do with a Bayesian framework.

Please the reply for comment 7.

Figure 4: Unfortunately, this figure is really hard to read. If possible, I would recommend splitting this up into several figures and providing the figures for individual scenarios in the supporting information. The choice of colors also makes it very difficult to see what's going on (especially the neon green and the soft peach color). I am not familiar with the HESS compiler, but I would also recommend either a significantly larger resolution and a different image format such as .tiff or .gif, as the current figure is in quite a low resolution and has serious compression artifacts. For graphs such as this one, vector-based formats such as .svg or .pdf (if saved straight from Python with pyplot.savefig) might also allow readers of the electronic version to zoom in arbitrarily close for details. This could be particularly valuable here, since most of the relevant details are quite small.

Thanks for your suggestion. We will improve the quality of the figures, including splitting this into several figures, improving the resolutions, using colors that are better distinguishable, and modifying the placing of legends.

Line 296-297: I would remove this statement, as you have not experimentally backed this statement up and it is not immediately obvious. I see little reason why inverting the observation residuals to find optimal input error realizations would be a more difficult task than re-ordering a pre-existing set of realizations. Quite the opposite, in fact.

The approach suggested by the reviewer, "inverting the observation residuals to find optimal input error realizations", is BATEA, which calibrates all the input error series together with model parameters. This leads to a high dimension problem (Renard et al., 2009). The statement here "*It is far more efficient to estimate the error rank than estimate the error value*." is based on the explanation "*In a continuous sequence of data, the potential error values have an infinite number of combinations, while the error rank has limited combinations, dependent on the data length.*".

Besides, directly optimizing the input error value according to the model residual is more likely to change the assumption about the error distribution. Therefore, in the BEAR method, the sampling and reordering strategy can not only ensure the error distribution assumption but also avoid the high dimension problem. The explanation has been already stated in the manuscript.

"*Note that modifying each input error according to the corresponding residual error only works in the rank domain. In the value domain, if there is no constraint on the estimated input errors, they will fully compensate for the residual error with the aim of minimizing the objective function and subsequently be overfitted. There are two ways to impose restrictions. One is to regard errors and model parameters as a whole in calibration, resulting in the high dimensional computation (Kavetski et al., 2006). The other is to sample error randomly from the assumed error model IBUNE (Ajami et al., 2007), whose precision cannot be guaranteed. While in the rank domain, the value range of the sampled errors can be effectively limited by the assumed error model.*"

Line 301-307: This is a very important paragraph. As I suggested in comment 7, through re-ordering you are no longer sampling from the prior input error model, which makes the protection from perfect fits you mention here somewhat arbitrary. As an illustration, I would like you to consider the behaviour of this re-ordering for longer time series: For a single observation, re-ordering can yield no improvement, and the residual fit depends exclusively on the realization you drew. For a few observations, re-ordering will induce moderate improvements, and the residual fit depends somewhat on the realization you drew. However, in the limit of infinitely many observations (assuming the statistical moments of the prior input error distribution are correctly characterized), re-ordering your error realizations should allow the residuals to be compensated completely at every single observation, irrespective of what specific realization you drew. This makes the protection against overfitting (and the expected residual error) dependent on the length of the observation time series and seems to converge towards deterministic (over) fitting. At the same time, the effective input error uncertainty decreases to zero. In a conventional Bayesian framework, even if the correlations in the input errors are perfectly identified, this would never happen.

The reviewer is correct that reordering will not be possible if a single observation is present, as the rationale behind reordering involves not changing the magnitude but the position of the error. As

the number of samples increases, the fit will improve (assuming the error distribution has been identified properly a priori). However, an infinite number of observations will not lead to a good model fit if the prior distribution is seriously mischaracterized. For example, if the input error distribution underestimates the frequency of large streamflow errors (say with a mis-specified positive skewness) the model parameters will attempt to compensate for the reduction in overall TSS concentration and the model fit will be poorer than if a correct distribution of input errors was assumed, even with infinite observations. This highlights the need for the approach to properly identify the input error distribution ahead of the model implementation, and future work will demonstrate how model selection techniques could be used when there is limited information on input errors a priori. Please also see the reply for the comment 7.

Line 314: The dot after (Fig 1.) should probably be a comma

This will be modified as flows:

"*Considering these two points, the BEAR method set q iterations in the algorithm (Fig. 1), and q increases until a defined target about the residual error is achieved.*"

**Summary:**

In summary, I find the approach an interesting and ambitious idea, but have reservations concerning its theoretical validity, which I hope the authors can address in their revision. If my fears concerning it solving an implicit inverse problem for the input error residuals happen to be confirmed, the authors might consider the following alternative avenues:

a) The approach might be re-interpreted as a diagnostic tool for input error residuals; there is some value in identifying input error residuals and the correlations between them. In this case, however, it may be worthwhile to investigate whether the re-shuffling strategy is needed, or whether a more straightforward inverse method might be more efficient.

b) If predictive improvements are desired, following the suggestions in major comment 4 could be a viable and interesting alternative avenue.

I wish the authors the best of luck with the manuscript and hope that my comments are useful.

We thank the reviewer for their thought-provoking comments. We hope our responses above have helped clarify some of the concerns the reviewer has raised. We intend to assess the remaining issues mentioned in our revision.

**Appendix A: The illustration of the BEAR method**

Table A 1 An example illustrating the BEAR method

| row | time step $i$ | 1 | 2 | 3 | 4 | 5 | 6 | 7 | 8 | 9 | 10 | 11 | 12 | 13 | 14 | 15 | 16 | 17 | 18 | 19 | 20 |
|---|---|---|---|---|---|---|---|---|---|---|---|---|---|---|---|---|---|---|---|---|---|
| | | | | | | | | | 1st iteration (random sample) | | | | | | | | | | | | |
| 1 | sampled input error | 0.07 | -0.12 | 0.07 | 0.16 | 0.05 | .0.07 | 0.07 | -0.03 | 0.03 | -0.08 | 0.09 | -0.11 | -0.11 | -0.08 | -0.29 | 0.14 | 0.03 | -0.08 | 0.14 | -0.17 |
| 2 | input error rank $k$ | 13 | 3 | 14 | 20 | 12 | 17 | 15 | 9 | 10 | 7 | 16 | 4 | 5 | 6 | 1 | 19 | 11 | 8 | 18 | 2 |
| 3 | residual error $\varepsilon$ | -0.29 | 0.49 | -0.58 | -0.98 | -0.78 | 0.29 | -0.66 | 0.59 | -1.31 | -0.31 | -0.87 | 0.76 | 0.46 | 0.54 | 0.25 | -0.80 | -0.07 | 0.56 | -0.23 | 0.40 |
| | MSE | | | | | | | | | | 0.40 | | | | | | | | | | |
| | | | | | | | | | 2nd iteration (random sample) | | | | | | | | | | | | |
| 4 | sampled input error | -0.01 | -0.02 | 0.03 | 0.03 | -0.09 | 0.00 | -0.02 | 0.06 | 0.11 | 0.11 | -0.09 | 0.01 | -0.12 | -0.11 | 0.00 | 0.15 | -0.08 | 0.04 | -0.02 | 0.11 |
| 5 | input error rank $k$ | 9 | 6 | 14 | 13 | 3 | 10 | 8 | 16 | 17 | 18 | 4 | 12 | 1 | 2 | 11 | 20 | 5 | 15 | 7 | 19 |
| 6 | residual error $\varepsilon$ | -0.13 | 0.23 | -0.43 | -0.41 | -0.21 | 0.70 | -0.23 | 0.09 | -1.88 | -1.52 | 0.20 | 0.17 | 0.53 | 0.60 | -0.43 | -0.72 | 0.36 | 0.12 | 0.47 | -0.82 |
| | MSE | | | | | | | | | | 0.47 | | | | | | | | | | |
| | | | | | | | 3rd iteration (updating the error rank via the secant method) | | | | | | | | | | | | | | |
| 7 | calculated pre-rank $K$ | 5.8 | 8.7 | 14.0 | 8.0 | -0.3 | 22.0 | 4.3 | 17.3 | -6.1 | 4.2 | 6.2 | 14.3 | 31.3 | 42.0 | 4.7 | 29.0 | 10.0 | 16.9 | 14.4 | 7.6 |
| 8 | ranked rank $k$ | 6 | 10 | 12 | 9 | 2 | 17 | 4 | 16 | 1 | 3 | 7 | 14 | 19 | 20 | 5 | 18 | 11 | 15 | 13 | 8 |
| | | | | | | | 3rd iteration (reordering errors according to the updated error ranks) | | | | | | | | | | | | | | |
| 9 | reordered input error | -0.02 | 0.00 | 0.01 | -0.01 | -0.11 | 0.11 | -0.09 | 0.06 | -0.12 | -0.09 | -0.02 | 0.03 | 0.11 | 0.15 | -0.08 | 0.11 | 0.00 | 0.04 | 0.03 | -0.02 |
| 10 | residual error $\varepsilon$ | -0.23 | 0.20 | -0.34 | -0.24 | -0.12 | 0.19 | 0.14 | 0.08 | -0.40 | -0.31 | -0.22 | 0.03 | -0.17 | 0.26 | -0.09 | -0.55 | 0.11 | 0.14 | 0.27 | -0.23 |
| 11 | MSE | | | | | | | | | | 0.06 | | | | | | | | | | |

The implementation of the BEAR method contains two main parts: sampling the errors from an assumed error distribution and reordering them with the inferred ranks via the secant method. An example is illustrated in **Error! Reference source not found.** and the explanation about the specific steps is presented in the following contents.

(1) In the 1st iteration ($q=1$), the errors are randomly sampled from the assumed error distribution (row 1), and then they are sorted to get their ranks (row 2). This error series is employed to modify the input data, which corresponds to a new model simulation and model residual (row 3).

(2) Repeat step (1) in the 2nd iteration ($q=2$) as two sets of samples are prerequisites for updating via the secant method. The results are shown in row 4, 5 and 6. **Error! Reference source not found.** demonstrates that the ranges of the error distribution are the same between the true input errors (black line) and the sampled errors (blue and green lines) as they come from the same error distribution under the condition that prior knowledge of the input error distribution is correct. However, the value at each time step is not close.

(3) At the 1st time step ($i=1$) in the 3rd iteration ($q=3$), the pre-rank $K_{1,3}$ is calculated via the secant method (illustrated as the following equation). The details are demonstrated in red boxes.

$$K_{1,3} = k_{1,2} - \varepsilon_{1,2}^p \frac{k_{1,2} - k_{1,1}}{\varepsilon_{1,2}^p - \varepsilon_{1,1}^p} = 9\text{-}(\text{-}0.13)\frac{9-13}{-0.13-(-0.29)} = 5.8$$

(4) Repeat the step (3) for all the time steps. The calculated pre-ranks are shown in row 7.

(5) Sort all the pre-ranks to get the error rank (row 8).

(6) According to the updated error ranks (row 8), the sampled errors in the 2nd iteration (row 4) are reordered. The example for the 1st time step is demonstrated in black boxes. The error rank at the 1st time step is updated as 6, and the rank 6 corresponds to the error value -0.02 in the 2nd iteration. Therefore, -0.02 is the input error at the 1st time step in the 3rd iteration. Following this example, the sampled errors at all the time steps are reordered. The results are shown in row 9. **Error! Reference source not found.** demonstrates that after reordering the errors with the inferred ranks, the estimated errors are much close to the true input error.

(7) The reordered input error will lead to a new input data, a new model simulation and a new model residual. The residual error is shown in row 10.

(8) If a defined target about the residual error is achieved, the input error estimation is accepted; Otherwise, $q=q+1$, repeat step (3)~(7) until $q$ is larger than the maximum numbers of iteration $Q$.

[Figure]

Figure A 1 Demonstration of the input error estimation in **Error! Reference source not found.** at the 1st and 2nd iteration where the input errors are randomly sampled

[Figure]

Figure A 2 Demonstration of the input error estimation in **Error! Reference source not found.** at the 3rd iteration where the input errors are reordered according to the updated error ranks

**Appendix B: Theoretical foundation**

1) Basic notation

In general, a model $M()$ simulates the output $Y^s$ given the observed input $X^o$, model parameters $\boldsymbol{\theta}$, as follows:

$$Y^s = M(X^o, \boldsymbol{\theta}) \tag{1}$$

Here and in the following, s represents the simulated value, o represents the observed value, and * represents the true value.

2) **Input errors**

Assume the input errors are represented by input multipliers sampled from an uncorrelated lognormal distribution, and the observed input $X^o$ can then be related to the true input $X^*$ by the following equation:

$$X^o = X^* \exp(\varepsilon_X), \varepsilon_X \sim N(\mu_X, \sigma_X^2) \tag{2}$$

where the $\varepsilon_X$ are assumed to follow a Gaussian distribution with mean $\mu_X$ and variance $\sigma_X^2$.

3) Output observational errors and model structural errors

In the derivation, these two parts are assumed to be error-free, therefore,

$$Y^o = Y^* \tag{3}$$

$$M() = M^*() \tag{4}$$

4) Remnant errors

Based on the previous assumptions, the observed output equals the true output, and the difference between the simulated output and the observed output, $\varepsilon$, will be equal to the difference between the simulated output and the true output, as follows:

$$Y^s = Y^o + \varepsilon = Y^* + \varepsilon, \varepsilon \sim (0, \sigma^2) \tag{5}$$

where the remnant errors $\varepsilon$ are assumed to follow a Gaussian distribution with mean 0 and variance $\sigma^2$.

5) Bayesian inference

Following the study of Renard et al. (2010), the posterior distribution of all inferred quantities is given by Bayes' theorem as follow:

$$p(\theta,\varepsilon_X,\mu_X,\sigma_X,\sigma \,|\, Y^o,X^o) \propto$$
$$p(Y^o \,|\, \theta,\varepsilon_X,X^o)p(\varepsilon_X \,|\, \mu_X,\sigma_X)p(\theta,\mu_X,\sigma_X,\sigma) \tag{6}$$

The full posterior distribution comprises the following three parts: the likelihood of the observed output $p(Y^o \,|\, \theta,\varepsilon_X,X^o)$, the hierarchical parts of the input multiplier $p(\varepsilon_X \,|\, \mu_X,\sigma_X)$ and the prior distribution of deterministic parameters and hyperparameters $p(\theta,\mu_X,\sigma_X,\sigma,)$.

Renard et al. (2009) argue that in the IBUNE method, the $\varepsilon_X$ are different in different iterations due to random sampling, therefore, cannot be updated effectively due to breaking the theoretical foundation of Bayesian inference. In the BEAR method, the secant method is applied to find a deterministic relationship between the $\varepsilon_X$ and the model parameters $\theta$ and hyperparameters of the multipliers $\mu_X,\sigma_X$. Therefore, $\varepsilon_X$ can be determined by $\theta,\mu_X,\sigma_X$, as follows:

$$\varepsilon_X = f(\theta,\mu_X,\sigma_X) \tag{7}$$

Considering $\varepsilon_X$ are sampled from $N(\mu_X,\sigma_X^2)$, $p(\varepsilon_X \,|\, \mu_X,\sigma_X)$ are the same when $\mu_X,\sigma_X$ are determined and do not need to be considered in the algorithm. Therefore, the posterior distribution of all inferred parameters in the BEAR method are as follows:

$$p(\theta,\varepsilon_X,\mu_X,\sigma_X,\sigma \,|\, Y^o,X^o) \propto$$
$$p(Y^o \,|\, \theta,\mu_X,\sigma_X,X^o)p(\theta,\mu_X,\sigma_X,\sigma) \tag{8}$$

The problem of parameter estimation and error identification can then be interpreted as the calibration of $\theta,\mu_X,\sigma_X$ when the relationship between the errors and parameters are determined. In the value estimation of the errors, only estimating the parameters and errors together can achieve this, however, in the rank estimation of the errors, the secant method can be applied to realize the rank between each error and its corresponding residual error (depending on the parameters).

**Appendix C: The results of the synthetic cases with the increasing standard deviation of output observational errors**

In order to explore the ability of the BEAR method with the interference of other sources of errors, synthetic cases have been built as the following table.

Table C. 1 Description of the synthetic cases

| subfigure | Input error model | Output error model | Prior information of input error model in calibration |
|---|---|---|---|
| (a) | $X^\circ = X^* \exp(\varepsilon_X)$, $\varepsilon_X \sim N(0.2, 0.3^2)$ | $Y^\circ = M(X^*, \theta)\exp(\varepsilon_Y)$, $\varepsilon_Y \sim N(0, \sigma_Y^2)$ $\sigma_Y = 0, 0.1, 0.2, 0.3$ | $X^\circ = X^* \exp(\varepsilon_X)$, $\varepsilon_X \sim N(0.2, 0.3^2)$ |
| (b)(c) | | | $X^\circ = X^* \exp(\varepsilon_X)$, $\varepsilon_X \sim N(0.2, \sigma_X^2), \sigma_X \in (0,1)$ |

[Figure]

Figure C.1 Mean square error of modified input vs synthetic input(a,b) and the estimated standard deviation of input error (c) for different synthetic cases (T represents the traditional method without considering the impacts of input error, D represents the IBUNE method, and R represents the BEAR method)

Figure C.1(a) demonstrates if the prior information of standard deviation of input errors is accurate, the BEAR method will always bring a better improvement of the input data, while the IBUNE method leads to a worse modified input than the input data without modification in method T. Figure C.1(b) illustrates that if the standard deviation of input errors is estimated in a wide range, whether the BEAR method can improve the input data or not depends on the significance of the

other sources of errors. The more dominant the input error is, the more effective the BEAR method is. When there are no other sources of errors($\sigma_Y=0$), the BEAR method can isolate the input error and parameter error perfectly as the standard deviation of input error converges to the reference value. However, the results of the IBUNE method are consistent with a much smaller estimation of the standard deviation of input errors in Figure C.1(c).

To sum up, the ability of the BEAR method depends on the accuracy of the prior information of the input error model and the significance of the input error in the residual error. It makes sense that other sources of error will interfere with the identification of the input error and its prior information can constrain these negative impacts. While the IBUNE method can slightly modify the input data only when the standard deviation of the estimated input error is much smaller than the true value. It is most likely to make use of the stochastic errors to improve the original input data, but not really identify the input error.

**References**

AJAMI, N. K., DUAN, Q. & SOROOSHIAN, S. 2007. An integrated hydrologic Bayesian multimodel combination framework: Confronting input, parameter, and model structural uncertainty in hydrologic prediction. *Water resources research,* 43.

BONASSI, F. V. & WEST, M. 2015. Sequential Monte Carlo with adaptive weights for approximate Bayesian computation. *Bayesian Analysis,* 10**,** 171-187.

RENARD, B., KAVETSKI, D. & KUCZERA, G. 2009. Comment on "An integrated hydrologic Bayesian multimodel combination framework: Confronting input, parameter, and model structural uncertainty in hydrologic prediction" by Newsha K. Ajami et al. *Water Resources Research,* 45.

RENARD, B., KAVETSKI, D., KUCZERA, G., THYER, M. & FRANKS, S. W. 2010. Understanding predictive uncertainty in hydrologic modeling: The challenge of identifying input and structural errors. *Water Resources Research,* 46.

---

## Author Response (AR1)

**Response to Editor:**

We thank the editor for the detailed explanations and helpful suggestions, and we have thought carefully about the limitations of the algorithm from a theoretical perspective. Our reply is listed in the following sections. The comments from the editor are provided in blue text and our responses are organized in black text. The manuscript text after the proposed changes is shown in "*black italics*". The number of the line, equation and section refers to the revised version of the manuscript without track changes, shown in yellow highlight.

In most cases, I agree with the reviewer's comments and with the answer of the authors to those comments. There are, however, certain reviewer comments with which I fully agree and which are not yet properly addressed in the answers of the authors. Unfortunately, some of these are quite major comments and I hope that the authors can address them properly. I explain these concerns in more detail in the following.

Most importantly, it is not clear what the consequences of the reordering step in the secant method are on the acceptance probability of the sampler that forms the outer loop (in this case a ABC-SMC algorithm). Changing the acceptance probability of any sampler almost always has consequences on the convergence proofs for that algorithm, i.e. on its ability to sample the intended distribution, which in this case is the posterior. Note that this concern is still there if one uses BEAR together with other samplers such as MCMC methods or SMC algorithms, as mentioned by the authors in their replies. This means that there are doubts about which distribution the BEAR algorithm actually samples. This point was raised by Reviewer 3 in comment 1. The authors try to address this concern in Appendix B of their response, but they do not manage to show that BEAR samples the posterior distribution. Since the $\varepsilon X$ are sampled a new in each iteration of the outer loop (sampler), we have different chances to reach $\tau$s for each s and therefore different chances of accepting the proposed parameter sample (i.e. different acceptance probability). Also the ranks, $k_q$, are re-initialized in each iteration. In the same answer, in Eq. 7, the authors claim that $\varepsilon X$ are a deterministic function of the parameters, but in the sentence below and in the scheme in Figure 1, they say that they are sampled from $f(\eta)$. According to Fig. 1, even $\eta$ changes during the inference.

Thanks very much for the editor's comment here. We concur the BEAR method does not update the posterior distribution of input errors, "the $\varepsilon X$ are sampled a new in each iteration of the outer loop (sampler)", and the calculation of input errors are re-initialized in each iteration. Instead, our approach leads to an optimized estimate of $\varepsilon X$ dependent on the model parameters and constrained by the prior distribution of $\varepsilon X$.

To explain: if the sample size is large (i.e. in the case of identifying the data-based input errors) and the error distribution is fixed, there is no significant difference among the error populations in subsequent samplings. Therefore, the error series εX is determined by the optimal error rank series as estimated via the secant method. When a model parameter is generated, the residual error only depends on the input error in this study. In other words, for a set of model parameters, the optimal input error series will lead to the minimized residual error. Therefore, we have revised the manuscript to state "εX are a deterministic function of the parameters" in Appendix B.

In cases where the error distribution is not fixed in the calibration ($\eta$ changes during the inference), $\eta$ ($\mu_X, \sigma_X$ in Appendix B) is regarded as a hyperparameter, calibrated along with the model parameters. Again in this case, the optimal input error will lead to the minimized residual error. Here, "εX are a deterministic function of the model parameters and input error parameter". To avoid confusion in the core idea of the rank estimation, "$\eta$ changes during the inference" has been deleted in Figure 1 and the methodology is based on the assumption that the error distribution can be pre-estimated. But to explore the ability of the BEAR method in wider applications and fully discuss its limitations, the concept of changing $\eta$ has been considered in a synthetic case and Appendix B.

To sum up, the reordering step is implemented when the model parameters and input error distribution have been updated and aims to find the optimal input error series corresponding to the minimized residual error (not the posterior distribution of the input errors, as the Editor and reviewers have noted). Thus, after the reordering step, the optimal input error εX is a deterministic function of the model parameter and input error parameter. It is re-calculated via the secant method for a new set of the model parameter and input error parameter and does not update its posterior distribution. We hope these revisions clarify the contribution of the work.

I see two possibilities: either the authors can provide a mathematical proof that their algorithm samples the posterior distribution, or they clearly state in the manuscript that the resulting distribution of the parameters has an interpretation that is not clear and possibly different from the actual posterior distribution. In that case, the approach cannot be considered a Bayesian approach (which doesn't mean that it cannot be useful).

Thanks very much for your suggestions. As we concur in the previous point, the BEAR method does not update the posterior distribution of the input errors, and due to this point, it would be inaccurate to refer to it as a formal Bayesian approach to identify the input errors. We have clarified this as follows:

"*The reordering step is implemented when the model parameter has been updated and aims to find the*

*optimal input error series corresponding to the minimized residual error. After the reordering step, the optimal input error is a deterministic function of the model parameter. Thus, unlike formal Bayesian inference, the BEAR method does not update the posterior distribution of the input errors, but identifies the input error through the deterministic relationship between the input error and model parameter.*" (line 318-322)

In the second case, I suggest to provide a more thorough synthetic case study, where the model parameters κ, Smax, a and b are known and re-inferred by BEAR, which is not shown at the moment. One thing to investigate in such a case study is also the effect of the temporal resolution (see comments in the next paragraph).

The synthetic case we provided in the manuscript is just the case where the model parameters κ, Smax, a and b are known and re-inferred by BEAR. Rather than providing the posterior distribution of each model parameter in the manuscript, we have added results demonstrating the overall effect of the model parameters in Figure 2(5). The explanation is as follows: "*In the validation period, the simulated output corresponds to the true input and estimated model parameters, and its NSE compared to the true output can assess the accuracy of the model parameter estimation.*" (line 229-231)

BEAR assumes a direct correspondence between the input and the output at each time-step. This necessitates that the number of input and output time-steps are the same and assumes that the model output at each time-step is only affected by exactly one input value: the one at the same time. However, memory effects introduced by water quality models can be substantial. For example, in the model used by the authors, the differential equation will lead to significant memory and input that happened multiple time-steps ago will affect the model output later on. This also depends on the temporal resolution of the model. The BEAR method will have significant problems with models that show a pronounced memory effect, especially when the temporal resolution of the data is high and one model output is affected by many model inputs.

We strongly agree with the editor and reviewers' opinion that the memory effect in the model response is very important in identifying the input error and we have carefully considered how to address this problem. In previous work, we have thoroughly investigated the memory effect in quantifying the input error in a hydrological model. We applied an autoregressive (AR) model to represent autocorrelation in the model residual error and remove this from the original model residual error. The results demonstrate that the ability of the BEAR-AR method in a hydrologic model is stable, it

outperforms the BEAR method (the details please see the paper (Wu et al., 2021)).

To investigate the usefulness of this approach here, we applied an autoregressive (AR) model with order 1 to represent autocorrelation in the model residual error and remove this from the original model residual error. The resultant error series is used to adjust the input error ranks. The results are denoted as "BEAR-AR" in the following Figure D 1 and demonstrate that the ability of the AR model is not stable. Compared with the results of the BEAR method (green bars), the BEAR-AR method (pink bars) in some cases improves the input error identification but in other cases, it worsens. We think this is a function of the degree of autocorrelation relative to the observational error magnitude, and the simple autoregressive function used here. While the approach does not properly solve the potential issue of autocorrelation, we believe the additional analysis demonstrates the potential avenues to address this and future work for BEAR.

[Figure]

**Figure D 1 Comparison of Nash-Sutcliffe efficiency (NSE) of the modified input v.s true input under the interference of the output observational errors with the increasing standard deviations in two calibration scenarios in synthetic case 2 (including *mul-fixed* and *mul-inferred*; notations are given in Table 2) via three calibration methods (including the IBUNE method and the BEAR method and the BEAR-AR method)**

The possible reason and related discussion have been added in the manuscript as follows:

*"If the response caused by an input is not instantaneous but exhibits persistence (i.e. occurs over several time steps), the autocorrelation in the output should be addressed to ensure the independence assumption of the rank updating is satisfied. Current ways to deal with this problem in hydrologic modelling can provide a reference to the potential modification of the BEAR method. Autocorrelation in the residual errors can be represented by an autoregressive moving average (ARMA) model (Kuczera, 1983) or autoregressive (AR) (Schaefli et al., 2007, Bates and Campbell, 2001). The correlated part of the error is removed from the residual error and the remaining part will be only impacted by the input error. Thus, the correspondence between the input error and the residual error part is ensured and the latter process will be the same as the application of the BEAR method in this study. Following this idea, the autoregressive (AR) model has been integrated with the BEAR method in the study of Wu et al. (2021) to deal with the autocorrelation of residual errors in a hydrologic model. The results prove this integration is effective to improve the input error estimation.*

*However, this treatment may not guarantee the improvement of the input error estimation in this study where the sediment concentrate is simulated at the daily time scale (Figure D 1). At this time scale, one input (streamflow) cannot impact the response (sediment concentration) for multiple time steps and the autocorrelation may not be well represented via a simple autocorrelation function. When the temporal resolution of the data is high (i.e. minute) and one model output is affected by many inputs, the memory effect may be addressed effectively via the AR model. Therefore, the specific representation of the autocorrelation in the residual error needs further discussion through comparisons in different time scales or with different characteristics in the memory effect."* (line 356-372)

**Reference:**

WU, X., MARSHALL, L. & SHARMA, A. 2021. Quantifying input error in hydrologic modeling using the Bayesian Error Analysis with Reordering (BEAR) approach. Journal of Hydrology, 126202.

**Response to Reviewer #1:**

The study proposes and demonstrates an algorithm for quantifying input uncertainty called BEAR (Bayesian error analysis with reshuffling). It is claimed that the method is suitable to overcome restrictions of current state-of-the-art approaches like high dimensional computational problems or underestimation and misidentification of error sources. For this purpose, the algorithm employs the secant method to estimate a certain rank of error associated to input data from an underlying rank distribution of errors. After introducing the method, it is demonstrated on the task of total suspended solids modelling in, first, a synthetic case study and, second, a real test case. Thereby, both, the effectiveness and the limitations are shown and discussed. Finally, transferability of the method within the field of water quality modelling and potential routes of improvement are presented.

**General comments:**

The issue of uncertainty quantification in modelling is for sure one of high importance. By focusing on input uncertainty this study addresses a branch that is particularly challenging in this field. Contributions in this direction deserve attention and the topic of this manuscript is suitable for the journal. However, certain issues regarding content and presentation of the material require to be addressed:

We thank the reviewer for the overall positive assessment of the manuscript and helpful comments, which we believe have helped improve the quality of the manuscript. We have responded to each point in turn in the following sections. The comments from the reviewer are provided in blue text and our responses are organized point-by-point in black text. The manuscript text after the proposed changes is shown in "*black italics*". The number of the line, equation and section refers to the revised version of the manuscript without track changes, shown in yellow highlight.

It should be noted that the method name has been changed from the "Bayesian error analysis with reshuffling" into "Bayesian error analysis with reordering". This is based on suggestions by one of the reviewers, as the word "shuffling" implies randomness in the reordering, while the reordering in our method is determined by the model residual error. The term "reordering" better reflects the deterministic nature of error quantified via this new method.

1) Maybe it is just the presentation, but it was not straightforward to see how the method exactly works. Aside of a more detailed explanation, providing more illustrations to support explanations about how the method exactly works might help, e.g. displaying the secant method itself, error distribution in rank space, etc.

Thanks for your suggestion. We have addressed this by modifying the methodology in the following points to make the algorithm clearer:

(1) Summarize the main steps in the BEAR method upfront:

*"Thus, the procedure of input error quantification has been developed into the following key steps: 1) Sample the errors from the assumed error distribution to maintain the overall statistical characteristics of the input errors; 2) Update the input error ranks to minimize the model residual via the secant method (Eq. (5) and (6)); 3) Reorder these sampled errors according to the updated error ranks; 4) Repeat 2) and 3) for a few iterations until a defined target is achieved. This new algorithm is referred to as the Bayesian error analysis with reordering (BEAR)."* (line 135-139)

(2) Integrate an example and its illustration in Appendix A to explain the specific steps involved. More explanation for the rank estimation via the secant method and the reordering steps has been added.

(3) Separate the description of the BEAR method from the ABC-SMC calibration scheme in Section 2.3 following suggestions from other reviewers. We changed this because the ABC-SMC calibration algorithm is not necessary in the BEAR method, and the core idea of the BEAR method (identifying the input errors by estimating their ranks ) can be easily applied in any other calibration algorithm, for example, MCMC and SMC algorithms.

(4) Change the ABC-SMC calibration scheme to the SMC scheme because it is easy to derivate the BEAR method from a Bayesian perspective in Appendix B.

2) By design, the BEAR method seems to shuffle and pick errors (by their ranks) such that maximum fit to the data is achieved. Is this a proper addressment of the input errors in terms of quantification of input uncertainty? For instance, in L.232 it is discussed that "method R always has much higher correlations with the true error series" and in L.243 it outperforms the other methods with highest NSE values. Both seem to be effects from the BEAR method searching

for optimally fitting errors until exactly the error is found that minimizes the gap between model predictions and observations.

The reviewer raised an important question. The BEAR method works well under the circumstance where the input error is dominant in the total uncertainty, where minimizing the residual error has a similar effect as minimizing the input error. From this point of view, the sampling and reordering strategy in the BEAR method provides an effective way to identify the input error according to the residual error. This is what the Reviewer refers to as "searching for optimally fitting errors until exactly the error is found that minimizes the gap between model predictions and observations". Like current methods that BEAR seeks to demonstrate an improvement over, the input error compensates for other errors, a step that is constrained by accurate prior information of the input error distribution being available (Figure 3). However, the compensating effect in the BEAR method is more apparent because it is much more effective than other current methods in minimizing the gap. Thus, the accuracy of the input error model is particularly important in the BEAR method. The analysis and discussion in Section 4.2 have been modified to convey this as follows:

*"The IBUNE method takes advantage of stochastic error samples to modify the input observations (Ajami et al., 2007). Figure C 2 demonstrates compared with O-fixed and O-inferred scenarios, S-fixed and S-inferred scenarios applies the simulated streamflow whose input error is more significant, and the resultant simulations (black line) via the IBUNE method are further away from the observed outputs (red dots). As per the finding in the previous study of Renard et al. (2010), if the $\sigma$ of input errors is inferred with the model parameters, the IBUNE method will underestimate $\sigma$ (in Fig. 2(1) and Fig. 4(1)). If $\sigma$ is fixed via prior information, the input modification and model simulation cannot be improved, especially in the scenarios with large intrinsic $\sigma$ of input errors, demonstrated in Fig. 2 and Fig. 3. From the above, the ability of the IBUNE method depends on the input data quality and the improvement of the input data and model simulation only happens when the standard deviation of the estimated input error is small. The availability of prior information is insignificant for the IBUNE method, especially when the intrinsic $\sigma$ of the input error is large.*

*However, the findings in the BEAR method are quite different. Accurate prior information about the input error model is important in the BEAR method. Figure 3 demonstrates fixed scenarios*

*calibrated via the BEAR method always produce a higher NSE of the modified input than inferred scenarios. This is likely because the prior information can constrain the input error distribution and reduce the impacts of other sources of errors. The availability of prior information of the input error relies on studies about benchmarking the observational errors of water quality and hydrologic data, and the selection of a proper input error model is important. Comparing the results in Figure 2, when the input error model is an additive formulation, the BEAR method consistently brings the best performance regardless of the prior information of the error $\sigma$. When the input error model is a multiplicative formulation, the BEAR method cannot improve the input data if the prior information of the error $\sigma$ is not accurate. This illustrates that the compensating effect between the input error and parameter error is weaker in the additive form of the input error. This is probably related to the specific model structure, as the exponent parameter b in BwMod has a stronger interaction with the multiplicative errors than the additive errors. Thus, more comprehensive comparisons should be taken to explore the capacity of different input error models in different model applications.*

*To sum up, the ability of the BEAR method depends on the accuracy of prior information of the input error parameter and the selection of the input error model. The IBUNE method can modify the input data when the standard deviation of the estimated input error is much smaller than the true value. It is most likely to make use of the stochastic errors to improve the original input data, but not effectively identify the input error.”* (line 324-349)

3) Expectations are raised that the method overcomes issues of state-of-the-art frameworks like BATEA and IBUNE. Yet, no direct comparison is shown which makes it hard to see the benefit of the method. Both these methods are frequently mentioned and a comparison is claimed. So far, there is a comparison of cases abbreviated by "T" (traditional), "D" (distribution) and "R" (BEAR method itself). "D" is referred to be "similar to the basic framework of the IBUNE method". However, this does not provide an actual comparison.

The reviewer is correct. We have changed the abbreviation to the full name of the methods as per the reviewer's suggestion, and added more explanations about this comparison, as follows:

“*The application of the BATEA framework is limited by high dimension computation (Renard et al., 2009). It probably becomes impractical in quantifying the data-varying errors (rather than the*

*event-varying errors in the study of BATEA (Kavetski et al., 2006)), where the dimension easily exceeds 1000 (Haario et al., 2005). Therefore, the BATEA method is not considered in the comparison. In this study, three methods, including the "Traditional" method, "IBUNE" method and "BEAR" method, are compared to evaluate the ability of the BEAR method in estimating the model parameters and quantifying input errors. "Traditional" method regards the observed input as error-free without identifying input errors (i.e. Eq. (2)), while the other two methods employ a latent variable to counteract the impact of input error and build the modified input (i.e. Eq.(3)). In the "IBUNE" method, potential input errors are randomly sampled from the assumed error distribution and filtered by the maximization of the likelihood function (Ajami et al., 2007). Although the comprehensive IBUNE framework additionally deals with the model structural uncertainty via the Bayesian Model Averaging (BMA) method, this study only compares the capacity of its input error identification part. The "BEAR" method adds a reordering process into the "IBUNE" method to improve the accuracy of input error quantification."* (line 162-173)

4) The method is supposed to reduce "the potential search space for input errors" (L.360). I wonder whether this is the objective quantification of input uncertainty? Isn't it rather a comprehensive assessment of the errors and noise associated to input error and not searching in a sub-space of already collected errors and then selecting the one that fits best during predictions?

We apologize for the lack of clarification. We have added more explanation in the revision. Here "reduce the potential search space for input errors" (L.360) is because "*In a continuous sequence of data, the potential error values have an infinite number of combinations, while the error rank has limited combinations, dependent on the data length. For example, in Table A1, the estimated error at the 1st time step could be any value. Even under a constrain of a range from the minimized to the maximized sampled errors (i.e. [-0.29,0.16] in the 1st iteration), its value estimation still has infinite possibilities due to its continuous nature. While the rank is discrete, having only 20 possibilities (i.e. the integrity in [1,20]). From this point of view, it is more efficient to estimate the error rank than estimate the error value, although the rank estimation will suffer from the sampling bias problem.*" (line 296-302)

To avoid the misunderstanding the current manuscript created, we have deleted "reduce the potential search space for input errors" (L.360), and changed this sentence as follows:

*"The estimation focuses on the error rank rather than the error value, which enhances the constraints of the input error model on the estimated errors and avoids the high dimensionality problem resulting from calibrating all the errors with the model parameter as a whole."*(line 378-380)

5) Generally, a thorough discussion on the used error distributions is missing, e.g. why is a bias of 0.2 in the error function assigned without further discussion (l.211)

Thanks for your comments. We have added clarification as follows:

*"If the input errors are estimated based on a rating curve, like the procedure in the following real case, the mean of the input error should be 0. But in order to test the ability of the BEAR method in wider applications, a systematic bias 0.2 has been considered in the synthetic case. An additive formulation (denoted as 'add' in Table 2) is suitable to illustrate the error generation in measurements, while the multiplicative formulation (denoted as 'mul' in Table 2) is specifically applied for errors induced from a log-log regression procedure, which is common for water quality proxy processes (Rode and Suhr, 2007)."* (line 207-213)

6) There is at least one article cited in the manuscript, that does not appear in the list of references (please see specific comments, l.190). Please assure correct referencing.

Thanks for your comments. We have corrected all the missing references.

**Specific comments**

1) L. 37-38: "…estimate the residuals between the measurements and proxy values…" -> yet, measurement error is not addressed

Thanks for your comments. we have clarified this as follows:

*"In this process, the measurement errors are ignored given the errors introduced from the surrogate process are commonly much more than the measurement errors (McMillan et al., 2012)."* (line 35-36)

2) L. 68: "variable" -> "scalar"– both, vectors and scalars represent variables

"variable" has been changed into "scalar". (line 69)

3) Eq. 3: unnecessary, since given by equation (1)

Yes, Equation (3) has been deleted.

4) L. 84-91: repetitive, add details to the corresponding paragraph in the introduction

Thanks for your suggestion. We have moved these details into the introduction, as follows:

*"The Bayesian total error analysis (BATEA) method provides a framework that has been widely used (Kavetski et al., 2006). Time-varying input errors are defined as multipliers on the input time series and inferred along with the model parameters in the Bayesian calibration scheme. It leads to a high-dimensionality problem, which cannot be avoided (Renard et al., 2009) and restricts the application of this approach to the assumption of event-based multipliers (the same multiplier applied to one storm event). In the Integrated Bayesian Uncertainty Estimator (IBUNE) (Ajami et al., 2007) approach, multipliers are not jointly inferred with the model parameters, but sampled from the assumed distribution and then filtered by the constraints of simulation fitting. This approach reduces the dimensionality significantly and can be applied in the assumption of the data-based multiplier (one multiplier for one input data) (Ajami et al., 2007). However, this approach is less effective because the probability of co-occurrence of all optimal error/parameter values is very low, and it results in an underestimation of the multiplier variance and misidentification of the uncertainty sources (Renard et al., 2009). From the above, a new strategy should be developed to avoid high dimensional computation and ensure the accuracy of error identification."*(line 43-54)

5) L. 92: "innovation" -> rather "introduction" or simple "The secant method" as chapter header
   – the innovation was made before

Thanks for your comment. The section titled *"innovation"* has been changed to *"introduction"*. (line 103)

6) L. 98-99: Rank definition and concept -> requires further explanation

We have added further explanations, as follows:

*"Here, the rank is defined as the order of any individual value relative to the other sampled values, and determines the relative magnitude of each error in all data errors. For example, in the 1st iteration in Table A 1, the error at 15th time step, -0.29, is the smallest value among all the sampled errors, therefore, its rank is 1."* (line 106-109)

7) L. 128ff: it sound like in ABC the requirements on the likelihood function are looser and therefore the method is easier to apply. However, requirements are also strict but ABC allows for Bayesian inference if the likelihood function is intractable. -> Please reformulate and clarify.

Thanks for your comments. Given the BEAR algorithm could be implemented via SMC, GLUE or SCE-UA, or any common model calibration approach, and this description about ABC confuses the main contribution of the paper (i.e. the core idea of BEAR method, which is to optimize input error ranks rather than input error magnitudes), we have recast the implementation of our optimization algorithm via SMC (the clarification is in Section 2.3).

8) L. 132ff: Notation "OF" not explained. Overall, the introduction of ABC and SMC is not clear. Further, the motivation why SMC is used here is not given.

Thanks for your comments. "OF" here means "objective function". But according to the above reply, we have recast the implementation via SMC, which target is the likelihood function rather than the objective function. Therefore, this sentence has been deleted.

9) L. 146: "…when 1000 proposed parameter sets…" -> is this suggested as a general approach or an arbitrary choice for this study. Please explain.

According to the reply for 7), we have recast the implementation via SMC. If the BEAR method is implemented using a likelihood-based calibration procedure, the proposed parameter is

compared to the previous entry in the chain and there is no need to set this stop criterion, just follow traditional convergence rules.

10) L.171ff: Please replace abbreviations T, D and R by their names. With all abbreviations that follow it is hard to keep track.

Thanks for your suggestion. We have changed the abbreviations into the full names in the below descriptions and related figures.

*"In this study, three methods, including the "Traditional" method, "IBUNE" method and "BEAR" method, are compared to evaluate the ability of the BEAR method in estimating the model parameters and quantifying input errors. "Traditional" method regards the observed input as error-free without identifying input errors (i.e. Eq. (2)), while the other two methods employ a latent variable to counteract the impact of input error and build the modified input (i.e. Eq.(3)). In the "IBUNE" method, potential input errors are randomly sampled from the assumed error distribution and filtered by the maximization of the likelihood function (Ajami et al., 2007). Although the comprehensive IBUNE framework additionally deals with the model structural uncertainty via the Bayesian Model Averaging (BMA) method, this study only compares the capacity of its input error identification part. The "BEAR" method adds a reordering process into the "IBUNE" method to improve the accuracy of input error quantification.."* (line 165-173)

11) LL. 190+196: "Sikorska et al, 2015" ! missing in references

Thanks, we have added all the missing references.

12) Eq. 9: define parameter "b"

The definition is shown in Table 1. A clarification has been added:

"*where the descriptions of a and b are shown in Table 1*" (line 196)

13) L. 215ff: incomplete sentence

This sentence has been completed as follows:

*"The true output $Y^*$ is the simulated TSS concentration via BwMod corresponding to the true input $X^*$ and model parameters set as the reference values in Table 1."*(line 203-204)

14) L. 229ff: "calibrated via method T,…" -> misleading explanation. Please provide a more specific explanation of the calibration process under error scenarios T, D and R

Please see the reply for 10)L. 171ff

15) L. 257: ":…the impacts of model structural error and output data error cannot be ignored." vs. L.264: ":…other sources of uncertainty can be ignored" -> sound like a contradiction, please elaborate

Thanks for your comments. We have modified the description as follows:

"*Case study 1 is an ideal situation that is used to test the effectiveness of the BEAR method in isolating the input error and the model parameter error. However, in real-life cases, other sources of errors (i.e. model structural error and output data error) will impact this effectiveness.*" (line 253-255)

"*According to the results of Figure 3 and the assumption of the methodology derivation, the BEAR method works better when the input uncertainty is more significant, so another input data source with more significant data uncertainty, a streamflow simulation from a hydrological model, has been considered.*" (line 271-274)

16) L. 282-283: "This illustrates that the impacts of other…" -> unclear phrase, please clarify and re-formulate

Thanks for pointing this out. This conclusion here is not clear and may lead to confusion. We have deleted this sentence and integrated the explanations into Section 4.2, as follows:

*"The IBUNE method takes advantage of stochastic error samples to modify the input observations (Ajami et al., 2007). Figure C 2 demonstrates that compared with O-fixed and O-inferred scenarios, S-fixed and S-inferred scenarios uses simulated streamflow whose input error is more significant, and the resultant simulations (black line) via the IBUNE method are further away from the observed outputs (red dots). As per the findings in the previous study of Renard et al. (2010), if the $\sigma$ of input errors is inferred with the model parameters, the IBUNE method will underestimate $\sigma$ (in Fig. 2(1) and Fig. 4(1)). If $\sigma$ is fixed as the prior information, the input modification and model simulation cannot be improved, especially in the scenarios with large intrinsic $\sigma$ of input errors (in Fig. 2 and Fig. 3). From the above, the ability of the IBUNE method depends on the input data quality and the improvement of the input data and model simulation only happens when the standard deviation of the estimated input error is small. The availability of prior information is insignificant for the IBUNE method, especially when the intrinsic $\sigma$ of the input error is large.*

*However, the findings in the BEAR method are quite different. Accurate prior information about the input error model is important in the BEAR method. Figure 3 demonstrates fixed scenarios calibrated via the BEAR method always produce a higher NSE of the modified input than inferred scenarios. This is likely because the prior information can constrain the input error distribution and reduce the impacts of other sources of errors. The availability of prior information of the input error relies on studies about benchmarking observational errors of the water quality and hydrologic data, and the selection of a proper input error model is important. Comparing the results in Figure 2, the BEAR method always brings the best performance regardless of the prior information of the error SD when the input error model is an additive formulation, while when the input error model is a multiplicative formulation, the BEAR method cannot improve the input data if the prior information of the error SD is not accurate. This illustrates that the compensating effect between the input error and parameter error is weaker in the additive form of the input error. This is probably related to the specific model structure, as the exponent parameter b in BwMod has a stronger interaction with the multiplied errors than the additive errors. Thus, more comprehensive comparisons should be undertaken to explore the capacity of different input error models in different model applications."* (line 324-345)

17) L. 291: "…could be regarded as the reference value." -> Why? Please explain.

Thanks for pointing this out. The observed streamflow from the rating curve cannot be considered as the reference value, it is just closer to the reference value than the simulated streamflow via GR4J. The explanations have been corrected as follow:

"*According to the results of the traditional method in Fig. C2, the simulations from the "O" streamflow (in (a1)) catch the dynamics of observed TSS concentration better than the simulations from the "S" streamflow (in (a3)). Thus, compared with the simulated streamflow via GR4J ("S" streamflow), the observed streamflow from the rating curve ("O" streamflow) should be closer to the true input data.*" (line 286-289)

18) L. 295-296: "…have an infinite number of combinations, while the error rank has limited combinations, dependent on data length." -> What is exactly meant here?

We have added the explanation as follows:

"*In a continuous sequence of data, the potential error values have an infinite number of combinations, while the error rank has limited combinations, dependent on the data length. For example, in Table A.1, the estimated error at the 1st time step could be any value. Even under the constraint of input error ranging from the minimized to the maximized sampled errors (i.e. [-0.29,0.16] in the 1st iteration), error magnitude estimation still has infinite possibilities due to the continuous probability distribution the error represents. In contrast, the rank is discrete, having only 20 possibilities (i.e. an integer from [1,20]). From this point of view, it is more efficient to estimate the error rank than estimate the error value.*" (line 296-302)

19) L. 297ff. "Compared with the IBUNE framework…" -> there is no real comparison made, please see major comments

Please see the reply to the general comment 3).

20) L. 340: "for method R, an accurate input error model can constrain the adverse impacts…" -> wasn't this the problem to begin with? Please clarify this sentence.

This point has been clarified as follows:

*"Accurate prior information about the input error model is important in the BEAR method. Figure 3 demonstrates fixed scenarios calibrated via the BEAR method always produce a higher NSE of the modified input than inferred scenarios. This is likely because the prior information can constrain the input error distribution and reduce the impacts of other sources of errors."* (line 334-337)

Please also see the reply to the general comment 2).

21) L 354-355: "However, the ability of these approaches needs further discussion in systems with correlated responses." -> Please clarify – what is the exact problem and why do ARMA models fit here?

Thanks for your comments. According to our latest research, the AR model is a good tool to represent the autocorrelation of the model response in the input error identification, and we have clarified this as follows:

*"If the response caused by an input is not instantaneous but exhibits persistence (i.e. occurs over several time steps), the autocorrelation in the output should be addressed to ensure the independence assumption of the rank updating is satisfied. Current ways to deal with this problem in hydrologic modelling can provide a reference in the potential modification of the BEAR method. The part of each residual error correlated with the previous residual errors can be represented by an autoregressive moving average (ARMA) model (Kuczera, 1983) or autoregressive (AR) model (Schaefli et al., 2007, Bates and Campbell, 2001). This correlated part is removed from the residual error and the remaining part will be only impacted by the input error. Thus, the correspondence between the input error and the residual error part is ensured and the latter process will be the same as the application of the BEAR method in this study. Following this idea, the autoregressive (AR) model has been integrated with the BEAR method in the study of Wu et al. (2021) to deal with the autocorrelation of residual errors in a hydrologic model. The results prove this integration is effective to improve the input error estimation.*

*However, this treatment can not guarantee the improvement of the input error estimation in this study where the sediment concentrate is simulated at the daily time scale (Figure D 1). At this time*

*scale, one input (streamflow) may not impact the response (sediment concentration) for multiple time steps and autocorrelation may not be well represented via a simple autocorrelation function. When the temporal resolution of the data is high (i.e. minute) and one model output is affected by many inputs, the memory effect may be addressed effectively via the AR model. Therefore, the specific representation of the autocorrelation in the residual error needs further discussion through the comparisons in different time scales or with different characteristics in the memory effect."* (line 356-372)

22) L. 358: "developed" -> "proposed" – the methods are already known but used in a way to address input error here.

We have changed *"developed"* to *"proposed".* (line 375)

23) L 362: "… addresses the high dimensionality problem…" -> not shown

Thanks for your comments. "Address" has been changed to "avoid" (line 379)

**Figures**

1) General: Legends in figures should be improved, e.g. in terms of colors or placing

2) General: Provide higher resolution and unify the legend (see especially Fig. 4 and 6)

3) Figure 3: please use colors that are better distinguishable (see cases "T" and "R")

Thanks for your suggestion. We have improved the quality of all the figures, including improving the resolutions and modifying the colors or placing of legends.

4) Figure 3(4): NSE = 1 is unrealistic. Please see major comments.

Thanks for your pointing this out. NSE is close to 1, not equal to 1. This occurs in the previous Figure 3 showing the results of the synthetic case where the modeling only suffers from input error and parameter error. The BEAR method is effective in isolating the input error and parameter error, which has been proved by the fact that NSE is much closer to 1. In the current manuscript,

we have implemented the BEAR method in the SMC calibration, and in new Figure 2, the NSE is still close to 1. However, when the BEAR method is applied in applications where other sources of errors will interfere, as Figure 3 shows, the fit to the output TSS observations reduces.

5) Figure 4 (c3,c4): model predictions are clearly shifted. Please elaborate on this offset.

The model applied is BwMod. When the input (streamflow) is large, the output (TSS concentration) will be reduced due to the wash-off effect. This is opposite to a hydrological model, where the large input (precipitation) will lead to a large output (discharge).

6) Figures 4 and 6: Maybe it is better to show these figures in the appendix and only present the most important subfigures in the main text

Thanks for your suggestion. We have moved these two figures into Appendix C.

**Tables**

1) Tables 1-1 and 1-2: The tables could be presented as additional files but are not helpful in the main article

Thanks for your comments. We have moved these two figures into Appendix A, integrating the descriptions and other figures to provide a more clear explanation about the BEAR method.

2) Table 3: the "fixed" scenarios in the real test case are not fixed but provide small hyperparameter ranges

Thank you for pointing this out. The small ranges have been changed into the fixed values in Table 2, and we have checked all the related results when the BEAR method is implemented in the SMC calibration scheme.

**Technical corrections**

1) L. 128: double ","

Thanks, this sentence has been deleted as the BEAR method has been recast in the SMC calibration scheme.

2) L. 142: "sth" -> make "s" italic

Thanks, this sentence has been deleted as the BEAR method has been recast in the SMC calibration scheme.

3) Eq. 8: unspecified symbol

We have removed this unspecified symbol. (line 192). Thanks.

4) L. 314: "q increasing until the objective: : :" -> incomplete sentence

Thanks, this sentence has been deleted as the BEAR method has been recast in the SMC calibration scheme.

**Response to Reviewer #2:**

This paper developed a new algorithm called BEAR for accurate quantification of input errors in water quality modeling. The precondition of the BEAR algorithm is that the input uncertainty should be dominant and that the prior information of the input error model can be estimated. Results of both synthetic data and observed data indicated the efficiency of the algorithm. Overall, the paper is well rewritten and the topic is suitable for the journal. However, the following issues should be further explained and clarified before its submission:

We thank the reviewer for the overall positive assessment of the manuscript and helpful comments. We have responded to each point in turn in the following sections. The comments from the reviewer are provided in blue text and our responses are organized point-by-point in black text. The manuscript text after the proposed changes is shown in "*black italics*". The number of the line, equation and section refers to the revised version of the manuscript without track changes, shown in yellow highlight.

It should be noted that the method name will change from the "Bayesian error analysis with reshuffling" into "Bayesian error analysis with reordering". This is based on suggestions by one of the reviewers, as the word "shuffling" implies randomness in the reordering, while the reordering in our method is determined by the model residual error. The term "reordering" better reflects the deterministic nature of error quantified via this new method. Besides, the abbreviations of methods (T, D, R) has been changed to the full names (Traditional, IBUNE, BEAR).

1) There have been many studies focusing on the uncertainty of input data errors for hydrologic modelling, and many methods including Bayesian algorithm can be used for handling the issue. However, the gap between previous studies and this study was not explained clearly in the Introduction. The motivation of this study should be clearly clarified.

Thanks for your suggestion. The research gap and motivation have been modified in the Introduction as follows:

"*Input uncertainty can lead to bias in parameter estimation in water quality modeling (Chaudhary and Hantush, 2017, Kleidorfer et al., 2009, Willems, 2008). Improved model calibration requires isolating the input uncertainty from the total uncertainty. However, the precise quantification of*

*time-varying input errors is still challenging when other types of uncertainties are propagated through to the model results. In hydrological modeling, several approaches have been developed to characterize time-varying input errors, and these may hold promise for application in WQMs. The Bayesian total error analysis (BATEA) method provides a framework that has been widely used (Kavetski et al., 2006). Time-varying input errors are defined as multipliers on the input time series and inferred along with the model parameters in a Bayesian calibration scheme. This leads to a high-dimensionality problem, which cannot be avoided (Renard et al., 2009) and restricts the application of this approach to the assumption of event-based multipliers (the same multiplier applied to one storm event). In the Integrated Bayesian Uncertainty Estimator (IBUNE) (Ajami et al., 2007) approach, multipliers are not jointly inferred with the model parameters, but sampled from the assumed distribution and then filtered by the constraints of simulation fitting. This approach reduces the dimensionality significantly and can be applied in the assumption of data-based multipliers (one multiplier for one input data) (Ajami et al., 2007). However, this approach is less effective because the probability of co-occurrence of all optimal error/parameter values is very low, resulting in an underestimation of the multiplier variance and misidentification of the uncertainty sources (Renard et al., 2009). From the above, a new strategy should be developed to avoid high dimensional computation and ensure the accuracy of error identification.*" (line 39-54)

2) More detailed steps about how to use the BEAR algorithm should be explained. Besides, the advantages of the BEAR algorithm compared with conventional methods should be more clearly clarified for making clear understanding from readers.

Thanks for your suggestion. The detailed steps of the BEAR method have been added in Appendix A and an illustration example has been moved from the methodology part to Appendix A to make the explanation clearer. In addition, the comparison with conventional methods has been clarified as follows:

"*The application of the BATEA framework is limited by high dimension computation (Renard et al., 2009). This becomes impractical in quantifying the data-varying errors (rather than the event-varying errors in the study of BATEA (Kavetski et al., 2006)), where the dimension easily exceeds 1000 (Haario et al., 2005). Therefore, the BATEA method is not considered in the comparison. In this study, three methods, including the "Traditional" method, "IBUNE" method and "BEAR" method, are compared to evaluate the ability of the BEAR method in estimating the model*

*parameters and quantifying input errors. "Traditional" method regards the observed input as error-free without identifying input errors (i.e. Eq. (2)), while the other two methods employ a latent variable to counteract the impact of input error and build the modified input (i.e. Eq.(3)). In the "IBUNE" method, potential input errors are randomly sampled from the assumed error distribution and filtered by the maximization of the likelihood function (Ajami et al., 2007). Although the comprehensive IBUNE framework additionally deals with the model structural uncertainty via the Bayesian Model Averaging (BMA) method, this study only compares the capacity of its input error identification approach. The "BEAR" method adds a reordering process into the "IBUNE" method to improve the accuracy of input error quantification."*(line 162-173)

3) Actually, the availability of prior knowledge of the input data error is important for modelling, but is also a difficult issue. It may be not enough only mentioning this issue in Conclusion. At least more discussions and the potential solutions should be provided.

The reviewer raised an important point. The discussion about this has been added in Section 4.2:

*"The availability of prior information of the input error relies on studies about benchmarking the observational errors of water quality and hydrologic data, and the selection of a proper input error model is important. Comparing the results in Figure 2, the BEAR method always brings the best performance regardless of the prior information of the error SD when the input error model is the additive formulation, while when the input error model is the multiplicative formulations, the BEAR method cannot improve the input data if the prior information of the error SD is not accurate. This illustrates that the compensating effect between the input error and parameter error is weaker in the additive form of the input error. This is probably related to the specific model structure, as the exponent parameter b in BwMod has a stronger interaction with multiplicative errors than additive errors. Thus, more comprehensive comparisons should be taken to explore the capacity of different input error models in different model applications."*(line 337-345)

4) The quality of some Figures in the manuscript should be improved to make all information clear.

Thanks for pointing this out. We have improved the quality of all the figures, including improving the resolutions and modifying the colors or placing of legends.

**Response to Reviewer #3:**

We have noted the reviewer's comprehensive remarks, and the following provides our response to each point. We appreciate the level of detail that the reviewer has gone into, and would like to thank the reviewer for the suggestions which have significantly improved how we convey the proposed approach. The comments from the reviewer are provided in blue text and our responses are organized point-by-point in black text. The manuscript text after the changes is shown in "*black italics*". The number of the line, equation and section refers to the revised version of the manuscript without track changes, shown in yellow highlight.

Complex patterns in input uncertainty such as spatial or temporal error correlations are an important topic in environmental science. In their present study, the authors seek to explore the ubiquitous issue of complex input uncertainty structures by proposing a novel method called Bayesian error analysis with reshuffling (BEAR). The proposed method is based on sampling an estimated input error and subsequently sorting the resulting realizations in an order which reduces residual mismatch to the observations. The authors then proceed to demonstrate the performance of their algorithm for a synthetic and a real case and compare its performance to a number of alternative setups.

I find the approach a very interesting and creative idea, and always appreciate it if someone takes the risks inherent to exploring a new methodological idea. Unfortunately, I have some reservations concerning its theoretical justifiability, which I hope the authors can address. Failing that, there might also be alternative ways to achieve similar effects which might stand on more robust theoretical foundations. Concerning these suggestions and the method itself, I have the following (major) comments:

**Major comments:**

1. Theoretical foundations: A key step of the approach is sorting input error realizations to reduce the residual mismatch between model predictions and system observations. I fear that this compromises the randomness of the error realizations, with potential consequences for the validity of the Bayesian inference that are difficult to predict. Rigorously deriving this algorithm from more basic theoretical foundations might better illustrate the consequences of the authors' assumptions

for Bayesian inference. If the authors do not have this expertise themselves (not to worry: few people in the environmental sciences do), I recommend seeking the help of the local statistics department – they are often keen to help. It seems an unfortunate truth of Bayesian statistics that interesting ideas for algorithms which sound nice on paper often tend to violate the Bayesian framework in unforeseen ways.

Thanks for your insightful comments. We have now added Appendix B that derives the BEAR algorithm, showing how we aim to optimise the input errors relative to the estimated model parameters. We believe the remarks of the reviewer here arise from the implementation of an Approximate Bayesian Computation (ABC) approach in model calibration, which is not a formal Bayesian method and confuses the main contribution of the paper (which is to optimize input error ranks rather than input error magnitudes). To address this, we have recast the implementation of our algorithm via the Sequential Monte Carlo (SMC) approach (see Section 2.3). The BEAR algorithm could be implemented via SMC, or GLUE, or SCE-UA, or any common model calibration approach, simply by optimising the latent-input errors on the error rank rather than error magnitude to realise the deterministic relationship between the residual error and the input error. It should be noted that we don't estimate the posterior distribution of the error ranks, we only optimise the error ranks by finding the deterministic relationship between the input error rank and model residual error (see Appendix B).

**2. Improving future performance:** While I am not intimately familiar with the alternative methods (BATEA, IBUNE) referenced by the authors, improving future model performance is a common motivation for learning complex uncertainty structures in hydrological models. The approach in this manuscript, however, requires a concurrent time series of observations. As a consequence, it does not really improve the error model itself and hence offers little value for improving future predictions. On the other hand, attempting to learn time-varying bias or correlation structures in the input errors – admittedly a sometimes computationally formidable task – can increase the likelihood of future predictions substantially. Such approaches would also be significantly easier to justify. I would recommend mentioning this limitation for the BEAR algorithm in the manuscript, or – better yet – either explore or propose ways on how this limitation could be circumvented.

The reviewer is correct in that the approach we propose does not 'improve' the error model but it improves the water quality model specification. It is assumed that the error model is known a priori and the overall objective is not to improve on this. We believe this is not an unreasonable assumption, as the model inputs are derived using a hydrologic model or rating curve whose error can be derived independently of the water quality model implementation, like the real case in this study. These independent observations provide insight into the distribution of input errors that can then be leveraged in the model calibration.This point has been clarified as follows:

"*The above approach does not improve the input error model itself but improves the WQM specification to have parameters closer to what would be achieved under no error conditions. Then the model can be more effectively used for scenario analysis (where we may know the hydrologic regime of a catchment in a hypothetical future), for forecasting under the assumption of perfect inputs (where the driving hydrologic forecast is independently obtained via a numerical weather prediction and a hydrologic model) or for regionalization of the WQM (where the model is transferred to a catchment without data). In all of these cases, an ideal model should have unbiased parameter estimates. This is our goal in identifying the optimal input errors, not to use the model for predictions with input data suffering the same errors.*" (line 96-102)

In order to prove the BEAR method will provide better predictions under 'no input error' conditions, we added validation results in the synthetic case (Figure 2(5)), where the input is set as the true value (without error) and the model parameters are set as the calibrated values via different methods. Compared with the traditional and IBUNE methods, the BEAR method brings a simulation closer to the true output with similar reliability and smaller sharpness (Appendix C), which means a better quantification of the input errors will result in less biased model parameters and produce a better prediction.

3. The influence of other error types: It seems the authors focussed most of their attention on input uncertainty. While I agree that input uncertainty can play a very important role, the influence of observation errors and model structural uncertainty plays a substantial role as well. In this study, the authors assumed both the model and the output observations were error-free – and derived their algorithm accordingly –, but in practice these assumptions are virtually never met. I would encourage the authors to explore how (if at all) their algorithm can avoid surrogacy effects in the

presence of observation and model errors. (What I mean by "surrogacy effects" is that the algorithm's adjustments to the input error realizations also 'soak up' [and consequently mask] errors in the output observations and the model itself in a bid to reduce the output residuals. This is of course undesirable.) See also my penultimate minor comment below.

The reviewer has raised here a very important point, and we completely agree with the issue of compounding sources of error and potential 'surrogacy' effects.

We have aimed to address this by implementing a real data case which likely has issues of error surrogacy, to identify how our approach works in a 'real' setting. We believe this helps balance our discussion of the approach and its potential limitations.

To additionally ensure we do not oversell our method given other sources of error, we have added a synthetic example in Section 3.3 that examines how the BEAR method performs under increasing error on the model calibration data. We hope this will help properly identify the usefulness of our approach but its potential limitations depending on the settings in which BEAR might be implemented.

4. Deterministic functions as an alternative: If the authors find that their algorithm might be based on flawed assumptions (following a more detailed theoretical derivation or investigation of the distribution it effectively samples from), the authors might wish to explore possible alternatives. If reducing the output residuals through adjustments to the input data remains the goal, a safer route might be to couple a deterministic input pre-treatment routine with the WQM and add its parameterization to the WQM parameter vector. Functionally, this pre-treatment routine can simply be interpreted as part of the deterministic model. Choices for this pre-treatment routine could be, for example, a one-dimensional spline which re-scales input magnitudes non-linearly (see the attached Figure 1, for an example with three extra parameters). More complex function choices might allow the consideration of lag, temporal or spatial correlations, etc. This would have the additional advantage over the BEAR framework that this pretreatment routine could also improve future predictions, assuming that it compensated true bias and is not overfitted. This comment is not a request for change, but a hopefully constructive suggestion for alternatives so that the authors might salvage some of their work in case it would turn out theoretically indefensible.

[Figure]

**Fig. 1.** Example of a non-linear re-scaling with a spline defined by three control points (red dots). You could use such a spline to scale the input values non-linearly.

We appreciate the thought that the reviewer has put into this comment, and we have considered this concept carefully, however, respectfully we do not see it as a viable pathway for the types of errors that we see in inputs to WQMs. The approach you suggested here is a non-linear bias correction to the input data which will correct systematic biases. However, the randomness of the observational error can't be explicitly identified and it does lead to biased model parameters in calibration. We present here a synthetic case to demonstrate such, where the model inputs are specified as $X^o = X^* + \varepsilon_X, \varepsilon_X \sim N(0, 0.2^2)$ , the observed outputs are specified as $Y^o = Y^* = M(X^*, \theta^*)$ , and the likelihood function in the SMC approach is calculated in log-transformed space. The mean of Gaussian distribution of the input error is set as 0, where the observed inputs are not systematically biased from the synthetic true inputs (Figure R1). Implied in the approach suggested by the reviewer, the observed inputs do not need modification. However, in the comparison of different methods in Figure R1, if we do not deal with the stochastic errors in input data, like the traditional method (the same as the suggested approach by the reviewer), the model parameters will be biased from their reference values. In contrast, the BEAR method can effectively identify the stochastic errors, beyond the systematic unbias of input data, as the BEAR method can push each input error close to the true value by reordering its rank according to the

residual error, and alleviate the impacts of both stochastic and systematic errors in input data on the model parameters.

Additionally note that we do not propose a method for rescaling errors, we are simply re-ordering their positions in the data to better correspond to deviations from model fits.

[Figure]

Figure R1 The posterior distributions of the model parameters estimated via the different methods and the scatter plot of the observed inputs vs true inputs in the synthetic case

5. BEAS instead of BEAR: This could be filed under nit-picking, but since the algorithm's name features so prominently, I chose to raise this to a major comment instead. The use of the word 'shuffling' implies randomness in the re-ordering. If I understood the authors' algorithm correctly, though, the re-ordering itself is entirely deterministic. As such, changing the name to something along the lines of "Bayesian error analysis with sorting" (BEAS) or "Bayesian error analysis with re-ordering" (if the authors like to retain their – admittedly very nice – acronym) might better reflect its deterministic nature.

We very much appreciate the valuable suggestion here, as the reviewer is completely correct. Yes, "reordering" better reflects the deterministic nature of error quantified via this new method.

Therefore, the method name will be changed into "Bayesian error analysis with reordering" to keep the BEAR acronym for short.

6. Why ABC: In the manuscript, the authors use an "Approximate Bayesian Computation via Sequential Monte Carlo" (ABC-SMC) approach. While I am not personally familiar with this approach, I struggle to see why it is necessary to resort to ABC, aside from any potential (forgive me) self-inflicted complications induced by the BEAR algorithm. The model and output variables seem pretty simple, so to my untrained eyes it is difficult to see why the formulation of an analytical likelihood should be impossible in this case. One could also cast the procedure the authors presented in Figure 1 with very few changes in terms of an MCMC routine, provided the re-ordering or error realizations ends up being statistically justifiable, of course. I would encourage the authors to provide a bit more detail on why ABC was necessary.

Thanks for the helpful suggestions here. The reviewer is completely correct that the ABC method is not necessary, and the core idea of the BEAR method (the reordering strategy according to the inferred ranks) can be easily applied in any other calibration algorithm, for example, MCMC or SMC algorithms. In order to clarify this point, we have recast the implementation of our algorithm via the Sequential Monte Carlo (SMC) approach (see Section 2.3), which is clearer to derive the BEAR algorithm from a Bayesian perspective under the assumption of optimised input errors (see Appendix B). This point has been clarified as follows:

"*The core strategy of the BEAR method is to identify the input errors by estimating their ranks, which can be easily integrated into formal Bayesian inference schemes (for example, Markov chain Monte Carlo (MCMC, (Marshall et al., 2004)) and Sequential Monte Carlo (SMC, (Jeremiah et al., 2011, Del Moral et al., 2006))) and other calibration schemes (for example, the generalized likelihood uncertainty estimation (GLUE, (Beven and Binley, 1992))). Based on a traditional calibration approach, the BEAR method works by replacing the observed input with the modified input that is obtained through the estimated input error rank via the secant method.*" (line 142-147)

7. Focus on a good fit: A key idea which seems to permeate the present manuscript is that it is desirable to obtain error realizations, if necessary by force (i.e., re-ordering), which match the observations as closely as possible. This is of course true, but not at all cost. Even assuming a severely mischaracterized prior input error distribution (e.g., a Gaussian with a standard deviation of 1E-10 and a mean of -1E8), one could theoretically obtain an error realization which causes the model to fit the observations perfectly if we only drew sufficiently (read: infinitely) many samples. The challenge, then, is not to find such a realization within our prior distribution (it will exist in any distribution with sufficiently broad support), but to find a distribution from which there is a high probability to obtain such a sample. Crucially, such a distribution should be independent from future observations, and I fear that this may not be the case for the approach proposed in this manuscript. In this approach, after re-ordering, the realizations are no longer i.i.d. samples from the input distribution (similarly to how one could interpret correlated Gaussian samples merely re-ordered independent Gaussian samples, but would nonetheless be wrong in claiming that an independent Gaussian distribution is identical to a correlated Gaussian distribution). If the authors decide to pursue my request for a derivation of a theoretical foundation for their approach (see comment 1), I recommend focussing on investigating from what effective distribution they are really sampling. Considering BEAR's ability to yield a reliably good fit with seemingly arbitrary prior input error realizations, I fear that the distribution you effectively sample from may be well approximated with (for example) a Gaussian with a mean inversely obtained from the observation residuals. If this turns out to be the case, the method would be more or less equivalent to just calculating the input error residuals through an inverse method of choice by minimizing the output residuals. This would not be very useful, and I would recommend exploring one of the approaches I suggested in comment 4 instead. See also my penultimate minor comment.

The reviewer has raised an interesting point here, that the reordered errors may include features (such as autocorrelation or dependency) that are not present in the original error sample. We have calculated the PACF of the reordered errors as a diagnostic for the case study on the potential interactions between observation error and other sources of error. The residuals show the reordering step does transform the autocorrelation of the residual error into input error. We have carefully considered the way to address this problem and applied an autoregressive (AR) model with order 1 to represent the autocorrelated part in the model residual error and remove this part from the original model residual error, the remaining part is used to adjust the input error ranks.

This approach was useful in a study of ours that considered the "memory effect" of hydrologic models and the persistence of input errors in the model outputs (Wu et al., 2021). The results are denoted as "BEAR-AR" in the following Figure D 1 and demonstrate that the ability of the AR model is not consistent in this study where the sediment concentrate is simulated at the daily time scale. Compared with the results of the BEAR method (green bars), the BEAR-AR method (pink bars) in some cases improves the input error identification but in other cases, it worsens.

[Figure]

**Figure D 1 Comparison of Nash-Sutcliffe efficiency (NSE) of the modified input v.s true input under the interference of the output observational errors with the increasing standard deviations in two calibration scenarios in synthetic case 2 (including *mul-fixed* and *mul-inferred*; notations are given in Table 2) via three calibration methods (including the IBUNE method and the BEAR method and the BEAR-AR method, the BEAR-AR method is the BEAR method after applying the autoregressive (AR) model to deal with the residual error)**

In using the AR model for quantifying the input error in a hydrological model, the results demonstrate that the BEAR-AR method in a hydrologic model is stable, and outperforms the

BEAR method (the details please see the paper (Wu et al., 2021)). The possible reason for these differences and related discussion have been added in the manuscript as follows:

*"If the response caused by an input is not instantaneous but exhibits persistence (i.e. occurs over several time steps), the autocorrelation in the output should be addressed to ensure the independence assumption of the rank updating is satisfied. Current ways to deal with this problem in hydrologic modelling can provide a reference to the potential modification of the BEAR method. Autocorrelation in the residual errors can be represented by an autoregressive moving average (ARMA) model (Kuczera, 1983) or autoregressive (AR) (Schaefli et al., 2007, Bates and Campbell, 2001). The correlated part of the error is removed from the residual error and the remaining part will be only impacted by the input error. Thus, the correspondence between the input error and the residual error part is ensured and the latter process will be the same as the application of the BEAR method in this study. Following this idea, the autoregressive (AR) model has been integrated with the BEAR method in the study of Wu et al. (2021) to deal with the autocorrelation of residual errors in a hydrologic model. The results prove this integration is effective to improve the input error estimation.*

*However, this treatment may not guarantee the improvement of the input error estimation in this study where the sediment concentrate is simulated at the daily time scale (Figure D 1). At this time scale, one input (streamflow) may not impact the response (sediment concentration) for multiple time steps and autocorrelation may not be well represented via a simple autocorrelation function. When the temporal resolution of the data is high (i.e. minute) and one model output is affected by many inputs, the memory effect may be addressed effectively via the AR model. Therefore, the specific representation of the autocorrelation in the residual error needs further discussion through comparisons in different time scales or with different characteristics in the memory effect."* (line 356-372)

A point of clarification: in each subsequent iteration of the BEAR algorithm, a new population of input errors are sampled from their a priori distribution. This means that the distribution of errors at each iteration is the same prior to reordering, i.e. the population of errors do not converge to a distribution that has different statistical features (mean, standard deviation, skewness). However, we do recognize that the reordering process provides insight into how the input errors may be

ideally distributed across time that can inform how the surrogacy effect is manifested (see comment 3). For example, if the errors are highly autocorrelated, this could highlight potential model structural errors, resulting from that the WQM has not properly represented the storage characteristics of TSS in the catchment. We believe that including further diagnostics will help elucidate this effect.

Note that the approach does in fact rely on an appropriate prior distribution being identified describing the input error distribution. If this error distribution is mischaracterized and the statistical features are different from the real input, the resultant outputs will not converge to the calibration data, and will lead to larger residual errors. A new updating will be taken to find a more appropriate assumption of the input error to get a smaller residual error. In some cases, the parameter error compensates this mischaracterized error distribution of input error and leads to an overfitting problem. This impact depends on the selection of the input error model, but can not be avoided in all the methods unless a method can properly deal with all sources of errors together. More discussions have been added in Section 4.2.

**Specific comments:**

Line 22-25: You mention the importance of complex interactions of different error sources directly in the first paragraph but proceed to largely ignore their influence in the remaining manuscript. I think this part is important and should be discussed in greater detail in the remainder of the manuscript (particularly also the methods/theory section).

Thanks for your suggestion, we agree this was not being properly highlighted. We have added an explanation in the methodology that emphasizes this:

*"It should be noted that the derivation of the BEAR method is based on the assumption that the model only suffers from input error and parameter error, but other sources of error (i.e. model structural error and output observational error) can also impair the estimation of the model parameters and are inevitable in the WQM. Considering this realistic situation, the ability of the BEAR method will be tested in a case study where the interference of other sources of error has been considered."* (line 77-80)

Line 49-51: During this review, I have briefly glanced into the corresponding methods BATEA and IBUNE, and apparently there was quite a commentary battle between the authors over these methods (see doi:10.1029/2007WR006538 and https://doi.org/10.1029/2008WR007215), and Renard et al. 2009 noted that IBUNE may in fact not reduce dimensionality. The choice is of course ultimately up to the authors, but it might be useful to add a small comment noting that the claim of dimension reduction by IBUNE is also challenged.

Thanks for your comment. In the comments from Renard et al. 2009, there are two different interpretations of the IBUNE framework. Interpretation A is the same as method D (the IBUNE method in the current manuscript) and Renard et al. 2009 argue the randomness in this approach will lead to an underestimation of the input error variance. Interpretation B is the same as the BATEA approach, which can not reduce the computational dimension.

According to the IBUNE paper (Ajami et al., 2007), the authors state that IBUNE can circumvent the high dimensionality issues by randomly sampling the multiplier to each time step from an assumed normal distribution. Therefore, we consider Interpretation A as the 'true' version of IBUNE and provide discussion on this in the manuscript.

Line 69-75: I would be careful here. In reality, there are many different error sources, certainly not the least of which is model structural error. Calibrating (i.e., simply minimizing the residuals between simulated and observed output) in the presence of other error sources is prone to surrogacy effects, so you can never really be sure you recovered the 'true' parameters. Even making the (unrealistic) assumption that we could manage to completely remove all error sources in the model and its input, we could still only retrieve the 'true' model parameters if our inverse problem is unique. I would mention these restrictions here (only works if all error sources can be removed completely, unique inverse problem).

Thank you for pointing out the equifinality problem. We have modified the description here as follows:

*"The proposed calibration (Eq. (3)) will turn into an ideal calibration where the optimal parameters $\theta^p$ will lead to the same simulation corresponding to the true values $\theta^*$ and the model residual $\varepsilon^p$ will decrease to zero. If the inverse problem (from the zero residual to find the optimal parameter) is not unique, the calibrated parameter $\theta^p$ may not converge to the true parameter $\theta^*$, but lead to the same simulation as the true parameter. In this study, these parameters are also denoted as $\theta^*$ and called ideal model parameters."* (line 86-90)

Line 70-73: In Equation (2), the variables $Y^o$ and $Y^s$ are used without any introduction. I assume both variables stand for the observed and simulated output. Please introduce these variables.

Thanks for pointing this out. We have clarified this as follows:

*"where $Y^s$ is the output simulated from the model M corresponding to the observed input $X^o$ and model parameter $\theta^c$, and the observed output $Y^o$ is assumed without observational errors in the derivation, thus can be denoted as $Y^*$."* (line 75-76)

Line 76-77: A critical thing here is that $\varepsilon$ is previously introduced as an error, which implies that you consider it to be a random variable. However, subtracting a (say) Gaussian random variable from another Gaussian variable with the same properties does not reduce variance to zero but actually doubles it (if both random variables are independent). What you seem to have in mind

here only works if $\varepsilon$ and $\varepsilon^p$ have identical properties and are perfectly correlated (note that this implies a lot more than just sharing the same statistical moments!), or if you are talking about error realizations. You should clarify this. This relates to major comment 7. You can only create this perfect correlation if you can somehow extract the error realizations of $\varepsilon$ (which only works under the assumption that you already have the input samples, that there are no other errors, and that the inverse problem is unique). Consequently, I fear that you may create/mimic this perfect correlation by implicitly solving an inverse problem, which would make the proposed method not very useful.

We agree with your argument that "Subtracting a (say) Gaussian random variable from another Gaussian variable with the same properties does not reduce variance to zero but actually doubles it (if both random variables are independent)". This is also the reason why the IBUNE method cannot provide accurate error identification by random sampling. The claim in this manuscript is "*If the equivalence between $\varepsilon_X$ and $\varepsilon_X^p$ can be ensured for each data point, the modified input $X^p$ then becomes the same as the true value $X^*$. The proposed calibration (Eq. (3)) will turn into an ideal calibration where the optimal parameters $\theta^p$ will lead to the same simulation corresponding to the true values $\theta^*$ and the model residual $\varepsilon^p$ will decrease to zero*" (line 85-87). As the reviewer notes, we refer here to the error realizations, not the distribution of errors.

We also agree that the perfect identification of input error series and model parameters only happens when "there are no other errors, and that the inverse problem is unique". Considering the impacts of other error sources, we have added the clarification as follows: "*It should be noted that the derivation of the BEAR method is based on the assumption that the model only suffers from input error and parameter error, but other sources of error (i.e. model structural error and output observational error) can also impair the estimation of the model parameters and are inevitable in the WQM. Considering this realistic situation, the ability of the BEAR method will be tested in a case study where the interference of other sources of error has been considered.*" (line 77-80). Considering the unique inverse problem, we have changed the description here to "*If the inverse problem (from the zero residual to find the optimal parameter) is not unique, the calibrated parameter $\theta^p$ may not converge to the true parameter $\theta^*$, but lead to the same simulation as the true parameter.*" (line 87-89)

Line 79: In Equation (4), you also mark the parameters – on which this entire exercise should be conditional on – as changing due to your proposed approach. The equations you have shown so far imply that the procedure you describe here is applied after parameter calibration. During a first reading of this paper, it is not immediately clear why the calibrated parameter values should change with your proposed approach. On a second reading, it becomes evident that you do not really calibrate but sample the parameter posterior, but that this sampling process is inter-woven with your BEAR routine, hence the parameter values are also affected. I recommend commenting on this already here to save your readers some confusion. Maybe it would also help not to talk about calibration at all in this context.

Apologies for the lack of clarification here. The reordering step for the input errors is implemented after one set of the model parameters has been updated. Corresponding to this set of model parameters, the optimal ranks of input errors are inferred via the secant method (the detailed algorithm has been clarified in ==Appendix A==). Although the sampled errors have been reordered, they still follow the assumed error distribution and the overall statistical characteristics remain unchanged. With the constraint of the input error distribution, in ==Equation (3)==, if the calibrated model parameters are far from the ideal model parameters, the reordered errors will not equal the true input errors and the residual error cannot be reduced effectively. If the model parameters are close to the true model parameters, the relative rank of the input errors can be identified precisely via the secant method, and the model residual error will approach zero. Minimizing the model residual error (or maximizing the likelihood function in Bayesian inference) here is the same as the goal of the model calibration. We have clarified the model calibration goal in the paper so that the intent is clear. The improved model calibration is the ultimate goal in properly identifying the input errors.

Line 81-83: I would be very careful with this statement. This only happens if the model parameters $\theta p$ and the input error residuals $\varepsilon_X^p$ cannot compensate each other, i.e. if there is only a single, unique combination of parameters and errors which yields zero residual. If you have a Pareto front along which different values of $\theta p$ and input error residuals $\varepsilon_X^p$ yield zero residual, you cannot be certain that you have correctly identified the 'true' parameter values and the 'true' input error, even in a scenario where no other errors/uncertainties exist. In addition to this, my reservations

concerning other error types raised in major comment 3 also apply. I would recommend changing this statement accordingly and exploring its consequences for your algorithm in greater detail.

We appreciate your comment. Yes, the input error and model parameter error might compensate each other, which leads to a Pareto front along which different values of θp and input error residuals yield zero residual. We have added the clarification as follows: "*Besides, if the identified input error and the model parameter can compensate each other, multiple combinations of model parameter and input error may yield zero residual and their estimates will be biased from the ideal values. A possible way to weaken this compensation effect will be explored in Sect. 4.2.*" (line 90-92).

In the manuscript, we compare two types of input error models: an additive formulation ( $X^o = X^* + \varepsilon, \varepsilon \sim N(0.2, 0.5^2)$ , *add* in Table 2) and a multiplicative formulation ( $X^o = X^* \exp(\varepsilon), \varepsilon \sim N(0.2, 0.5^2)$ , *mul* in Table 2). According to our synthetic cases, we find the multiplicative formulation is more likely to compensate the model parameter error, but the compensation effect will reduce with accurate prior information on the input error model. This is probably because the impacts of errors in the multiplicative formation are similar to the parameter *b* in Equation 8, or the nature of the multiplicative formulation is more likely to change with the model parameter than the additive formulation, due to having more flexibility. In sum, the compensation effect can be reduced through proper selection of the input error model as this affects the identification accuracy. But this problem is common in all the methods considering input error, and is not special to the BEAR method. More discussion has been added in Section 4.2.

Line 104-106: I would rephrase this a bit, because following the procedure you outlined in Figure 1 (a very nice schematic, by the way), it is not only two steps: you sample the error once, then iterate over a large number of re-ordering steps until you find an order which minimizes your output residuals. This could do with some clarification.

Thanks for your suggestion. We have rephrased the sentence as:

"*Thus, the procedure of input error quantification has been developed via the following key steps: 1) Sample the errors from the assumed error distribution to maintain the overall statistical characteristics of the input errors; 2) Update the input error ranks to minimize the model residual*

*via the secant method (Eq. (5) and (6)); 3) Reorder these sampled errors according to the updated error ranks; 4) Repeat 2) and 3) for a few iterations until a defined target is achieved. This new algorithm is referred to as Bayesian error analysis with reordering (BEAR).''* (line 135-139).

Line 115-116: If I understood your explanations here correctly, maybe an easier way of explaining what you are doing is that you sort your updated error ranks, then assign to each of them a new integer rank based on its position in the sorted list. This might be easier than trying to explain this procedure with scaling.

Thanks for your suggestion. We have modified this sentence as follows:

"*Sorting $k_{i,q}$ in all the ranks $k_{i,q} (i=1,...,n)$ can address this problem by effectively assigning to each of them a new integer rank based on its position in the sorted list.*" (line 125-126).

Line 125-131: As mentioned in major comment 6, please devote some space to explain why the models you use in the following necessitate the use of ABC. Even after going through the manuscript a few times, I struggle to see why standard Bayesian approaches would be impossible to use. At the risk of evoking the anger of our ABC-focussed colleagues: direct is usually better than approximate. It also does not become clear in the manuscript why an ensemble-based approach is used – couldn't the same procedure be implemented in an MCMC-style acceptance/rejection algorithm? If there are some ABC reasons for requiring an ensemble, it you might also want to explain why and how it is used.

We apologize for the lack of clarification. The ABC is not necessary for the BEAR method to be implemented. We admit there is no difference in the calibration results between the standard Bayesian approaches and the ABC approach for this case study. We have clarified this as follows:

"*The core strategy of the BEAR method is to identify the input errors by estimating their ranks, which can be easily integrated into formal Bayesian inference schemes (for example, Markov chain Monte Carlo (MCMC, (Marshall et al., 2004)) and Sequential Monte Carlo (SMC, (Jeremiah et al., 2011, Del Moral et al., 2006))) and other calibration schemes (for example, the generalized likelihood uncertainty estimation (GLUE, (Beven and Binley, 1992))). Based on the traditional calibration approach, the BEAR method works by replacing the observed input with a*

*modified input that is obtained through the estimated input error rank via the secant method.*" (line 142-147).

To avoid confusion, we have recast the implementation of our algorithm via the Sequential Monte Carlo (SMC) approach (Section 2.3) and redraw the flowchart (Figure 1). Besides, we have derived the foundation of the BEAR method in Appendix B.

Line 132-136/Figure 1: This explanation of the method is very short, and essentially only explains that you approach the posterior distribution iteratively through a number of intermediate steps, but not how this is achieved exactly. Figure 1 provides more information and suggests some sort of acceptance/rejection scheme depending on whether your procedure can reduce the error residuals below a certain threshold, but the nature of the posterior distributions from which new parameter values are drawn remains undefined. You also seem to update an input error parameter $\eta x$, which seemingly contradicts statements you made suggesting the input errors are sampled from a pre-estimated distribution (Line 10, Line 193-194). This step is also never mentioned in the text itself up to this point – you only mention that you estimate the input error distribution's hyperparameters much later. The text also frequently mentions 'populations', which evoke the idea of an ensemble-based method, but none of the steps mentioned in the text so far actually seem to require an ensemble. Please provide some more (written) detail about how your algorithm functions exactly.

Thanks for your comments.

1)To clarify the algorithm, we have recast the implementation via the Sequential Monte Carlo (SMC) approach (Section 2.3) and redraw the flowchart (Figure 1). Also, we have changed the description in the manuscript, as follows:

*"The details of the SMC algorithm can be found in the study of Jeremiah et al. (2011). Figure 1 demonstrates the integration of the BEAR method into the SMC sampler. In the SMC scheme, s refers to the number of sequential updating populations of parameter vectors (particles). The maximum number of the population S is set as 200 in this study. In each sequential population, N particles of model parameters are calibrated. N is set as 100 in this study. For each particle of the model parameters, the corresponding input error ranks are updated over q iterations, where q increases until the acceptance probability is larger than a number randomly sampled from 0 to 1.*

*It should be noted that if the model parameters are far away from the true values, especially in the initial population, iterative updating of the error ranks will have little effect in reducing the model residual. Therefore, the maximum number of iterations should be set, referred to as Q. Q is set as 20 in this study. If q exceeds Q, the algorithm returns to the mutation step in Fig. 1."* (line 151-160)

2) To avoid confusion, We have removed "update an input error parameter ηx" from the methodology part and Figure 1, but kept this consideration in the case study and added the clarification as follows:

"*Given the description in the introduction, the input error model can be pre-estimated independent of calibration by analysing the input data in some studies. While in other cases, the input error model cannot be estimated or the accuracy is in question. Therefore, two scenarios about the prior information of $\sigma$ have been considered: one is fixed as the reference values (denoted as 'fixed' in Table 2), the other one is estimated as the hyperparameters with the model parameters (denoted as 'inferred' in Table 2)."* (line 214-218)

3) The term "population" is used as in the SMC algorithm, and it is like an ensemble-based method. We have clarified this by a better description of how the algorithm is implemented via SMC, as follows: *"A population means a group of parameter vectors (particles) that is updated in each iteration"* (line 153)

Line 145-148 and Line 159-162: This is just a comment towards the general "Why ABC?" discussion. It seems to me that a classic MCMC procedure would avoid the need for adjusting the acceptance threshold dynamically, as proposed parameters are always compared to the previous entry in the chain.

Yes, ABC is not necessary for the BEAR method and any MCMC procedure will avoid the need for adjusting the acceptance threshold. We have recast the implementation of our optimization algorithm via SMC. See the reply for Line 125-131.

Line 154-157: I confess that this explanation is quite impenetrable, and probably causes more confusion here than it does good. I recommend restructuring this explanation or removing it

altogether. A good alternative would be to visualize this with a small figure, possibly added to the supporting information if length limitations to not permit embedding it into the main text.

Thanks for your suggestion. We have integrated an example and its illustration in Appendix A to explain the specific steps involved, and a detailed explanation of this has been given in step (6).

Line 192: I do not see from Equation 8 or 9 or their surrounding text how the spatial scale factors into this model. Through the Sa variable? Please clarify this.

Thanks for your comments. "*The two formulations in Bwmod were developed in a small-scale experiment (Sartor and Boyd, 1972), while in applications at the catchment scale, the conceptualized parameters largely abandon their physical meanings and the formulations can be considered a "black-box" (Bonhomme and Petrucci, 2017).*" This study attempts to simulate the sediment dynamics in the catchment scale. Thus, we say "the spatial scale is set as the catchment in this study". This has been clarified as follows:

"*This study will test the BEAR algorithm in a case of simulating the daily sediment dynamics of one catchment, thus, the time scale is typically set as daily and the spatial scale is set as the catchment.*" (line 183-184)

Line 200: In Equation 8, you do not introduce Smax and κ. Please introduce these variables as well.

Thanks for your comments. The introduction of Smax and κ is shown in Table 1. We have added clarification as follows:

"*where the descriptions of κ and Smax are shown in Table 1.*" (line 192)

Line 203: In Equation 9, you do not introduce a and Qt^b. Please introduce these variables as well.

Thanks for your comments. $Qt^b$ should be $(Q_t)^b$, we have modified Equation (8) and the description as follows:

$$s(S_{a,t}) = a \cdot (Q_t)^b \cdot S_{a,t}$$

"*where the descriptions of a and b are shown in Table 1, and $Q_t$ is the streamflow at the catchment outlet at time t.*" (line 196)

Line 205: In Equation 10, you do not introduce Qt. Please introduce this variable as well.

See the above reply.

Line 209-211: For this section, there are a few assumptions which could warrant greater discussion. If the errors are normally estimated in advance based on a rating curve, why is there a constant offset of 0.2? Couldn't this systematic bias be corrected through the rating curve itself? Alternatively, if the offset is necessary because your errors are asymmetrically fat-tailed, wouldn't a different distribution (such as a scaled beta or gamma distribution) be a better choice? It is commendable to make the synthetic test case more challenging by introducing bias as well, but how would this be recognized a priori in a real test case if it wasn't already considered in the rating curve? Some more information might clarify the authors' choice of distribution for the audience.

Thanks for your comments. Yes, we agree with your statement that if the errors are normally estimated in advance based on a rating curve, the mean of the normal distribution should be 0. By including the systematic bias of 0.2 in this synthetic study, we aimed to test the BEAR method in wider applications, even if these are not necessarily representative of true errors. Figure 2 indicates that in the synthetic case where the input errors have a systematic bias, the BEAR method can bring more accurate estimations of the input errors and the model parameters. Therefore, in situations where the input errors have no systematic bias, the BEAR method also works. According to your comments, we added the clarification as follows:

"*If the input errors are estimated based on a rating curve, like the procedure in the following real case, the mean of input error should be 0. But in order to test the ability of the BEAR method in wider applications, a systematic bias 0.2 has been considered in the synthetic case even though this is unlikely to manifest in real situations.*" (line 207-210)

 This part here is a bit unclear. What I deduce from the context is that you looked at two scenarios – one, where you left the prior input error fixed, and one where you estimated the input error hyperparameters as well. I would not talk about 'conditions' in this context, but rather about 'scenarios'. If I understood your drift here correctly, I would also add a comment which puts more emphasis on the fact that you subvert one of the principal assumptions you made earlier in the second scenario (namely, that the input error distribution is a prior/pre-estimated).

Thanks for your comments. To clarify this, we have changed "conditions" to "scenarios" here and have modified the description as follows:

"*Given the description in the introduction, the input error can be pre-estimated independent of calibration by analysing the input data in some studies. While in other cases, the input error model cannot be estimated or its accuracy is in question. Therefore, two scenarios about the prior information of $\sigma$ have been considered: one is fixed as the reference values (denoted as 'fixed' in Table 2), the other one is estimated as the hyperparameters with the model parameters (denoted as 'inferred' in Table 2).*" (line 214-220)

Line 232-234: I would recommend to critically re-examine this part in the light of major comment 7. The high correlation of scenario R with the realizations of the synthetic true error series – which are supposed to be realizations from an independent Gaussian distribution – might be reason for concern, as they suggest that you might be implicitly solving an inverse problem for the input error residuals. This would have little to do with a Bayesian framework.

Thanks for your comments. It should be noted that "correlation" here is not the correlation among the estimated input errors, but the correlation between the estimated input error and the true input error, which can be used to evaluate the ability of the method to identify the dynamics of the input error. Regarding the fact that reordering will break the iid assumption of the input errors, please see the reply of comment 7.

Figure 4: Unfortunately, this figure is really hard to read. If possible, I would recommend splitting this up into several figures and providing the figures for individual scenarios in the supporting

information. The choice of colors also makes it very difficult to see what's going on (especially the neon green and the soft peach color). I am not familiar with the HESS compiler, but I would also recommend either a significantly larger resolution and a different image format such as .tiff or .gif, as the current figure is in quite a low resolution and has serious compression artifacts. For graphs such as this one, vector-based formats such as .svg or .pdf (if saved straight from Python with pyplot.savefig) might also allow readers of the electronic version to zoom in arbitrarily close for details. This could be particularly valuable here, since most of the relevant details are quite small.

Thanks for your suggestion. We have improved the quality of the figures, including splitting this into several figures, improving the resolutions, using colours that are better distinguishable, and modifying the placing of legends. Please see Appendix C in the manuscript.

Line 296-297: I would remove this statement, as you have not experimentally backed this statement up and it is not immediately obvious. I see little reason why inverting the observation residuals to find optimal input error realizations would be a more difficult task than re-ordering a pre-existing set of realizations. Quite the opposite, in fact.

Thanks for your insightful comments. The approach suggested by the reviewer, "inverting the observation residuals to find optimal input error realizations", is BATEA, which calibrates all the input error series together with model parameters. This leads to a well established high dimension issue (Renard et al., 2009). The statement here "*It is far more efficient to estimate the error rank than estimate the error value.*" is based on the explanation "*In a continuous sequence of data, the potential error values have an infinite number of combinations, while the error rank has limited combinations, dependent on the data length.*". To clarify this, we have added some explanations as follows:

"*For example, in Table A1, the estimated error at the 1st time step could be any value. Even under a constraint of the range from the minimized to the maximized sampled errors (i.e. [-0.29,0.16] in the 1st iteration), its value estimation still has infinite possibilities due to the continuous nature of the error. In contrast, the rank is discrete, having only 20 possibilities (i.e. the integrity in [1,20]). From this point of view, it is more efficient to estimate the error rank than estimate the error value.*" (line 298-302)

Considering the advantages of reordering a pre-existing set of realizations over directly optimizing the input error value, please see the explanation as follows:

"*Besides, to avoid the high-dimension calculation, modifying each input error according to the corresponding residual error only works in the rank domain. In the value domain, if there is no constraint on the estimated input errors, they will fully compensate for the residual error to maximize the likelihood function and subsequently be overfitted. There are two ways to impose restrictions. One is to regard errors and model parameters as a whole in calibration, like the BATEA framework (Kavetski et al., 2006), resulting in a high dimensional computation. The other is to sample error randomly from the assumed error model, like the IBUNE framework (Ajami et al., 2007), whose precision cannot be guaranteed due to the error randomness. However, in the BEAR method the inference focuses on the error rank where the value range of the sampled errors can be effectively limited by the assumed error model. Additionally, adjusting the order of the sampled errors according to the inferred error rank can reduce the randomness in the IBUNE framework (Ajami et al., 2007), which significantly improves the accuracy of the error estimation (as demonstrated by much higher NSEs than the IBUNE method in Fig. 2). The reordering step is implemented when the model parameter has been updated and aims to find the optimal input error series corresponding to the minimized residual error. After the reordering step, the optimal input error is a deterministic function of the model parameter. Thus, unlike formal Bayesian inference, the BEAR method does not update the posterior distribution of the input errors, but identifies the input error through the deterministic relationship between the input error and model parameter*"
(line 309-322)

Line 301-307: This is a very important paragraph. As I suggested in comment 7, through re-ordering you are no longer sampling from the prior input error model, which makes the protection from perfect fits you mention here somewhat arbitrary. As an illustration, I would like you to consider the behaviour of this re-ordering for longer time series: For a single observation, re-ordering can yield no improvement, and the residual fit depends exclusively on the realization you drew. For a few observations, re-ordering will induce moderate improvements, and the residual fit depends somewhat on the realization you drew. However, in the limit of infinitely many observations (assuming the statistical moments of the prior input error distribution are correctly

characterized), re-ordering your error realizations should allow the residuals to be compensated completely at every single observation, irrespective of what specific realization you drew. This makes the protection against overfitting (and the expected residual error) dependent on the length of the observation time series and seems to converge towards deterministic (over) fitting. At the same time, the effective input error uncertainty decreases to zero. In a conventional Bayesian framework, even if the correlations in the input errors are perfectly identified, this would never happen.

The reviewer is correct that reordering will not be possible if a single observation is present, as the rationale behind reordering involves changing the position of the error but not the magnitude. As the number of samples increases, the fit will improve (assuming the error distribution has been identified properly a priori). However, an infinite number of observations will not lead to a good model fit if the prior distribution is seriously mischaracterized. For example, if the input error distribution underestimates the frequency of large streamflow errors (say with a mis-specified positive skewness) the model parameters will attempt to compensate for the reduction in overall TSS concentration and the model fit will be poorer than if a correct distribution of input errors was assumed, even with infinite observations. This highlights the need for the approach to properly identify the input error distribution ahead of time, and future work will demonstrate how model selection techniques could be used when there is limited information on input errors a priori. Please also see the reply for the comment 7.

Line 314: The dot after (Fig 1.) should probably be a comma

Thanks, this sentence has been deleted as the BEAR method has been recast in the SMC calibration scheme.

**Summary:**

In summary, I find the approach an interesting and ambitious idea, but have reservations concerning its theoretical validity, which I hope the authors can address in their revision. If my fears concerning it solving an implicit inverse problem for the input error residuals happen to be confirmed, the authors might consider the following alternative avenues:

a) The approach might be re-interpreted as a diagnostic tool for input error residuals; there is some value in identifying input error residuals and the correlations between them. In this case, however, it may be worthwhile to investigate whether the re-shuffling strategy is needed, or whether a more straightforward inverse method might be more efficient.

b) If predictive improvements are desired, following the suggestions in major comment 4 could be a viable and interesting alternative avenue.

I wish the authors the best of luck with the manuscript and hope that my comments are useful.

We thank the reviewer for their thought-provoking comments. We hope our responses above have helped clarify the concerns the reviewer has raised.

[revised manuscript text omitted]

---

## Referee Report (RR1)

I thank the authors for their detailed revision of the manuscript. Overall, their revision helped clarifying a lot of uncertain aspects of the algorithm, and their responses to my concerns were mostly satisfactory, but work still remains to be done. Unfortunately, I fear that my main concerns about the theoretical foundations and consequences of the error re-ordering remain inadequately addressed. The derivation provided in Appendix B does not sufficiently clarify these points eithers: the core question (*how exactly the re-ordering affects the base error distribution's statistical moments or hyperparameters, and what the consequences are in Bayesian terms*) remains unaddressed.

Since I take it that the authors would like to stick with the error re-ordering approach, I would argue that this leaves you with two possible pathways:

1) Provide a thorough theoretical derivation and in-depth investigation of what effect the error re-ordering really has, and what this means in Bayesian terms. After experimenting a bit with error re-ordering myself (see the Python code snippet below), I believe that a good start point might be couplings or measure transport (Pierre E. Jacob has a nice online lecture series on that called *Couplings and Monte Carlo*), as you seem to convert one distribution (the raw error distribution) into one with different statistical moments, one somehow moulded to the ideal error realizations.

2) Alternatively, you could simply drop the "Bayesian" attribute from your study or replace it with "Pseudo-Bayesian". This might require that you adjust the acronym. Even without full theoretical justification, your algorithm can still provide a useful heuristic, and maybe that is enough. Even in this case, however, I believe the manuscript would benefit from an isolated analysis of what mechanically happens to the input error distribution when subject to reordering. I have provided a few thoughts on this below.

As a consequence, I would recommend another round of major revisions. If you follow option (2), I recommend specifically to add a new section into the theory/methodology chapter which explores and illustrates the effects of re-ordering errors on the raw error distributions in detail (this is important for the reader's understanding of the approach– it should be part of the main manuscript, not the Appendix). To make space for this section, you could absorb some of the practical comments in the discussion (specifically, sections 4.1 and 4.3). There are also a lot of tangential comments addressing reviewer concerns throughout the manuscript which could be removed if their key points are addressed in this new section. This might also support the narrative thread of the manuscript by helping you to avoid the need to go on explanatory tangents.

As I don't want to leave you hang out to dry on such a large and amorphous task, I would specifically suggest exploring a simple example case in this proposed section. Specifically:

• Ignore the model (for the purpose of error reordering this is unnecessary); instead, skip straight to positing some hidden "ideal" sequence of error realizations which perfectly compensate the true residual error (similar to your figure A1); derive the corresponding ranks;
• Use a simple, structured residual error sequence to make it easier to read the induced effect. The sequence doesn't have to be realistic, merely insightful;
• Use a residual error distribution perfectly adjusted to the structured error you defined. In a synthetic test case, it's easy to derive an empirical cdf.
• Then explore the consequences of re-ordering for time series of different length or input error distributions of different quality. Discuss the statistical moments after reordering.

I have provided an example Python script for this below and appended some of its result figures for different time series lengths of 10, 100, and 1000 in Figures 1, 2, and 3. In this example, I arbitrarily assumed the true/ideal error realizations to follow a sine curve. Note that something more realistic (like random samples from a Gaussian distribution) would have also worked, but the simplicity of a sine curve makes it significantly easier to read the effect of the re-ordering.

Some thoughts on the results:

As I suspected in major comment #7 for the first round of revisions, the longer the time series, the more likely the method is to achieve a "perfect fit", so the effect of error re-ordering depends on the length of the time series. You discuss this briefly in the manuscript, but I think that this is among the most important mechanisms of BEAR, so it is worth demonstrating in isolation. For short time series (Figure 1), error re-ordering can already induce some degree of improvements by causing the marginal sample mean to follow the "ideal" error realizations; at the same time, the marginal error standard deviation decreases. This effect is exacerbated for longer time series (T=100, Figure 2, and T=1000, Figure 3). The consequence seems to be that the unordered, raw error distribution is "molded" to the ideal error realizations. I suspect (and you seem to share these suspicions in your responses) that in the limit of an infinitely long time series, error realizations would be compensated perfectly. This is important to discuss for prospective users of your manuscript, as it affects the algorithm's behaviour in somewhat unexpected ways.

[Figure]

*Figure 1. Effect of error re-ordering for a perfect error distribution, an ensemble size of N=1000 and a time series length of T=10. The upper subplot shows the "ideal" realizations to compensate some residual error. The left centre plot shows N unordered realizations of an error distribution with the correct statistical moments of the ideal realizations (obtained by forming a cdf for a full sine wave). The right centre plot shows the marginal mean and standard deviation at each time step. The left bottom plot shows the N error realizations in the subplot above after ordering, and the right bottom plot shows the corresponding marginal mean and standard deviation.*

[Figure]

*Figure 2. Effect of error re-ordering for a perfect error distribution, an ensemble size of N=1000 and a time series length of T=100. The upper subplot shows the "ideal" realizations to compensate some residual error. The left centre plot shows N unordered realizations of an error distribution with the correct statistical moments of the ideal realizations (obtained by forming a cdf for a full sine wave). The right centre plot shows the marginal mean and standard deviation at each time step. The left bottom plot shows the N error realizations in the subplot above after ordering, and the right bottom plot shows the corresponding marginal mean and standard deviation.*

[Figure]

*Figure 3. Effect of error re-ordering for a perfect error distribution, an ensemble size of N=1000 and a time series length of T=1000. The upper subplot shows the "ideal" realizations to compensate some residual error. The left centre plot shows N unordered realizations of an error distribution with the correct statistical moments of the ideal realizations (obtained by forming a cdf for a full sine wave). The right centre plot shows the marginal mean and standard deviation at each time step. The left bottom plot shows the N error realizations in the subplot above after ordering, and the right bottom plot shows the corresponding marginal mean and standard deviation.*

Other interesting things to visualize might be what happens if the error distribution is not perfect (for this, just replace "vals = dist(np.random.uniform(size=(1000,resolution)))" in the code with some other distribution). You already show this indirectly in your models, but demonstrating this effect in isolation rather than through the lens of performance metrics might be a lot clearer. Feel free to take inspiration from my example code or use it directly. I have attached it at the end of this manuscript.

Of course, this code snippet just demonstrates what happens when we are re-ordering error realizations, not how you arrive at the error ranks and their interaction with the parameter inference in the first place (which are potentially additional topics to discuss). I hope that even if you decide to

follow option (1), this snippet might give you some ideas on where to start with the Bayesian justification. Good luck!

**Specific comments** (any line numbers I list correspond to the non-track-changes manuscript):

Line 47: the same multiplier applied to one storm event

Maybe "the same multiplier applied to all time steps of one storm event" might be better, if I understand this part correctly; using "same" for a singular object ("storm event") sounds a bit strange.

Line 184: the time scale is typically set as daily and the spatial scale is set as the catchment

This is not particularly clear. I assume you simulate in daily timesteps, and aggregate the catchment's (presumably) surface area into a single spatial unit? If so, it might be better to replace this with "thus, we use daily time steps and consider the catchment a single, homogeneous spatial unit" or something along these lines.

Lines 302: From this point of view, it is more efficient to estimate the error rank than estimate the error value,

This sentence ends on a comma, not a period.

Line 309-311: Besides, to avoid the high-dimension calculation, modifying each input error according to its corresponding residual error only works in the rank domain. In the value domain, if there is no constraint on the estimated input errors, they will fully compensate for the residual error to maximize the likelihood function and subsequently be overfitted

This requires more discussion in the revised manuscript, as it seemingly contradicts what you write in the paragraph immediately prior: In the previous paragraph, you recommend sampling error realizations repeatedly and selecting the optimal realization to overcome "sampling bias" and improve the fit to the actual observations. However, in this paragraph you praise this very same sampling bias for preventing overfit. These are contradictory messages: provided you get the ranks right, if you were to resample an infinite number of times, you would eventually get an error realization which compensates the true error perfectly (even if your input error distribution is a really poor approximation to the "true" error distribution), thus negating your protection against overfit. The fact that this protection against overfit depends on the length of the time series might not be that much of an issue if you interpret your approach merely as a heuristic, but even in this case you need some practical guidelines on when to re-sample for short time series. The proposed dedicated section might help clearing some of this confusion up.

Line 320-322: Thus, unlike formal Bayesian inference, the BEAR method does not update the posterior distribution of the input errors, but identifies the input error through the deterministic relationship between the input error and model parameter.

As far as I can see, this is the first time you mention that BEAR is not a formal Bayesian inference method, so I suspect you would go for option (2). In any case, mentioning this in the discussion for the first time is a bit late: Something this important should be stated earlier, as early as the introduction or abstract. If you add the section exploring the consequences of error reordering, this would also be a good place to elaborate on this.

Aside from this manuscript structuring argument, the statement in these lines is also unfortunately wrong. Refer to the attached figures for demonstration. I also elaborate more on this two comments below.

Line 383: "However, the work in this study still identifies a few areas needing to be explored."

Nit-pick: This sentence is a bit unwieldy, in my opinion. How about "However, this study identifies a few areas which still need to be explored:"?

Response file / Response to major comment #7: "*A point of clarification: in each subsequent iteration of the BEAR algorithm, a new population of input errors are sampled from their a priori distribution. This means that the distribution of errors at each iteration is the same prior to reordering, i.e. the population of errors do not converge to a distribution that has different statistical features (mean, standard deviation, skewness).*"

You are of course correct, but I fear your comment misses the point somewhat: While it is undoubtedly true that the raw input error distribution never changes, BEAR *does* effectively update the input error distribution. In the figures from my code snippet, this can be seen in the bottom right subplots when compared to the centre right subplots. It becomes evident that the both the effective mean and standard deviation (and likely higher statistical moments as well) change dramatically after re-ordering. In essence, the fact that you are re-ordering transforms your raw error distribution and causes you to sample some different latent distribution instead. What this distribution really is remains the key question of your entire approach. If you can figure that out, you'll be one large step closer to justifying this approach theoretically. =)

In a sense, what you are doing seems distantly related to ideas in measure transport, see for example Marzouk et al. (2016) for an overview. In measure transport, the ultimate goal is to indirectly sample from an (almost) arbitrary target distribution. This is achieved by sampling a simple reference distribution instead (for example a multivariate standard Gaussian), then converting these reference samples through a deterministic function into samples from the target distribution. Of course, finding the correct transformation function is the key objective of this entire endeavour, and consequently its main challenge. In your study, you approach this from the opposite direction: you have some transformation, now you should find out what distribution you are sampling.

In summary, I would say the parallels to your approach are as follows: even though the reference distribution (corresponding to your raw input error distribution) never changes, the *pushforward distribution* (corresponding to the latent distribution your re-ordered error realizations are effectively sampled from) changes with the transformation function (in your case, the re-ordering according to different error ranks).

Marzouk, Y., Moselhy, T., Parno, M., & Spantini, A. (2016). An introduction to sampling via measure transport. *arXiv preprint arXiv:1602.05023*; https://arxiv.org/abs/1602.05023.

**Example code**.

```python
import numpy as np
import matplotlib.pyplot as plt
from scipy.interpolate import interp1d
from matplotlib.gridspec import GridSpec
import copy

**Set a random seed**
np.random.seed(0)

**Create some dummy error values to generate an artificial ranking**
resolution  = 10 # Adjust this for different time series lengths

**Create a sequence of "ideal" error realizations following a sine curve**
x = np.linspace(0,2*np.pi,resolution)
y = np.sin(x)

**Get the sorted indices of these error realizations**
ranks   = np.argsort(y)

**Create a sorter list which can be used to reshape a sorted list according to**
**the ranks we just defined**
sorter  = np.argsort(ranks)

**Get the statistical moments of this residual error distribution**
mean    = np.mean(y)
std     = np.std(y)

**Create a figure, plot the ranks for each error realization**
plt.figure(figsize=(16,6))
```

```python
gs = GridSpec(nrows=3,ncols=2)

plt.subplot(gs[0,:])

plt.scatter(x,y)

plt.xlabel('data series $t$')

plt.ylabel('ideal residual \n error')

for idx,i in enumerate(ranks):

    plt.text(x[i],y[i],str(idx))

**Create a cdf for a sine curve ---------------------------------------------**

**Sample the sin cure with high resolution**
dummy_sine_samples = np.sin(np.linspace(0,2*np.pi,10000))

**Sort these samples**
dummy_sine_samples = np.sort(dummy_sine_samples)

**Search for potential duplicates**
sineval,count = np.unique(dummy_sine_samples,return_counts=True)

**Create a cumulative sum for the counts**
cumsum = np.cumsum(count,dtype=float)

**Standardize the cumulative sum**
cumsum -= cumsum[0]

cumsum /= cumsum[-1]

**Create an interpolation function which takes a quantile as input and returns**
**the corresponding sine value**
dist   = interp1d(

    cumsum,

    sineval )
```

```python
**Statistical moments of the raw input error distribution --------------------**

**Draw a thousand realizations of our custom error distribution**
vals    = dist(np.random.uniform(size=(1000,resolution)))

**Plot the error realizations**
plt.subplot(gs[1,0])
for n in range(1000):
    plt.plot(x,vals[n,:],color='grey',alpha=0.01)
plt.xlabel('data points $t$')
plt.ylabel('raw input error \n realizations')

**Plot the sine curve for reference**
plt.plot(x,y,'r',label = 'ideal error realizations')
plt.legend()

**Calculate the mean and covariance of the samples**
mean_1  = np.mean(vals,axis=0)
cov_1   = np.cov(vals.T)

**Plot the mean**
plt.subplot(gs[1,1])
plt.plot(x,mean_1,'r')
plt.ylabel('raw input error \n marginal mean',fontdict={'color':'r'})
plt.ylim([-1.1,1.1])
ax2 = plt.gca().twinx()
ax2.plot(x,np.sqrt(np.diag(cov_1)),'b')
ax2.set_ylabel('raw input error \n marginal std',fontdict={'color':'b'})
ax2.set_ylim([0,0.75])
```

```python
**Statistical moments of the sorted input error distribution ------------------**

**Now sort the error realizations**
vals_sorted = copy.copy(vals)
vals_sorted = np.sort(vals_sorted,axis=1)

**And redistribute them according to the ranks we determined**
for n in range(vals.shape[0]):
    vals_sorted[n,:]   = vals_sorted[n,sorter]

**Plot the re-ordered error realizations**
plt.subplot(gs[2,0])
for n in range(1000):
    plt.plot(x,vals_sorted[n,:],color='grey',alpha=0.01)
plt.xlabel('data series $t$')
plt.ylabel('sorted input error \n realizations')

**Plot the sine curve for reference**
plt.plot(x,y,'r',label = 'ideal error realizations')
plt.legend()

**Calculate the statistical moments**
mean_2  = np.mean(vals_sorted,axis=0)
cov_2   = np.cov(vals_sorted.T)

**Plot the mean**
plt.subplot(gs[2,1])
plt.plot(x,mean_2,'r')
plt.ylabel('sorted input error \n marginal mean',fontdict={'color':'r'})
plt.ylim([-1.1,1.1])
plt.xlabel('data series $t$')
```

```python
ax2 = plt.gca().twinx()

ax2.plot(x,np.sqrt(np.diag(cov_2)),'b')

ax2.set_ylabel('sorted input error \n marginal std',fontdict={'color':'b'})

ax2.set_ylim([0,0.75])

**Save the figure**

plt.savefig('consequences_or_error_reordering_resolution_'+str(resolution)+'.png',bbox_inches='tight')
```

---

## Referee Report (RR2)

**Review Round 3 of „Quantifying input uncertainty in the calibration of water quality models: reshuffling errors via the secant method" by Xia Wu, Lucy Marshall and Ashish Sharma, December 2021**

Thanks to the authors for their effort in improving the manuscript. Overall, the changes made have increased clarity and do help to better understand the working principles behind BEAR and its purpose. Yet, some of the revised sections point into a direction that I struggle to comply with:

"Therefore, a modification should be made in the IBUNE approach to improve the accuracy of input error identification." (l. 54-55) I disagree with the revised version of this sentence and therefore suggest to delete it. It indicates that there was a flaw in the IBUNE method that has to be fixed. However, as Bayesian approach, IBUNE simply samples errors without processing them unlike BEAR, which is also stated in l. 344ff.: "In the IBUNE framework (Ajami et al., 2007), the errors are also sampled from the error distribution, but not reordered. Thus, the error precision at each time step cannot be guaranteed. In the BEAR method, adjusting the sampled errors according to the inferred error rank reduces the randomness of the error allocation in the IBUNE framework…" This randomness in sampling is part of the fully Bayesian approach and not something that per se has to be fixed. That said, I find the second sentence about a "guaranteed error precision" unclear and obsolete, and suggest deleting it.

Overall, this pertains to the general question of whether BEAR is a full Bayesian approach that I referred to as "arbitrary error treatment" in earlier rounds of reviewing and that Reviewer 2 highlighted in great detail pointing also at the theoretical problems underlying the approach.

This is clarified in l. 154 ff. to some degree, i.e. "the BEAR method does not provide formal Bayesian inference". However, this statement is still somewhat hidden in the article and therefore I would like to second the suggestion of reviewer 2 in generally renaming the method a "Pseudo-Bayesian error analysis with reordering".

Renaming BEAR to pBEAR (for example) might not appear too appealing to the authors but it would be a more honest name of the method and therefore increase scientific soundness. Hence, I suggest publishing the manuscript with this minor name modification all over the manuscript. I stick to my former evaluation that this manuscript is a valuable contribution to the general discussion of input error treatment. The methodology of BEAR is an idea worth publishing and with the label "pseudo-Bayesian" this will also be a clear contribution to a broad audience.

---

## Author Response (AR2)

Comments from editor

Thank you for your considerable effort in improving this manuscript. I acknowledge that the quality has improved and major points have been addressed. One of the reviewers suggests only minor revisions at this point, while the other still sees major points that have to be clarified. I suggest that you address their comments, including the effect of reordering the realizations of the error distribution. I believe that this additional illustration would be very useful for the reader and help to understand the mechanics of the presented approach. After having considered these changes, I believe that this manuscript will eventually be a valuable contribution to stimulate new techniques of efficient parameter estimation under uncertainty.

We thank the editor and reviewers for the overall positive assessment of the manuscript.

According to the comments of the 1st reviewer, we have revised all the descriptions related to BATEA and IBUNE and highlighted the contribution of this study on the introduction of rank estimation and the secant method in the input error identification.

Based on the comments of the 2nd reviewer, we have moved the Appendix B into the main text as Section "2.3 Bayesian inference of input uncertainty and the BEAR method" to explain the theoretical basis of the BEAR method in the Bayesian framework and added Section "4.2 The effect of reordering on the error realization" to clarify the mechanics of reordering step in the input error identification.

We appreciate these useful comments, which we believe have helped improve the quality of the manuscript and inspired more understandings of the BEAR method from different angles. We have responded to each point in turn in the following sections. The comments from the reviewer are provided in blue text and our responses are organized point-by-point in black text. The manuscript text after the proposed changes is shown in "*black italics*". The number of the line, equation and section refers to the revised version of the manuscript without track changes, shown in yellow highlight.

Anonymous Referee #1

I think the manuscript clearly improved over the last iterations. Yet, I still struggle with the discrepancy between expectations that are raised in the title, abstract and introduction, and what is presented in the article, i.e. that BEAR is as comprehensive as BATEA and IBUNE and therefore an alternative to these two. This impression is fostered by phrases like L.53-54: "From the above, a new strategy should be developed to avoid high dimensional computation and ensure the accuracy of error identification." or in chapter "2.4 Comparison with other methods", where, first, BATEA is discarded with the plain statement that it might run into dimensionality problems and then BEAR is presented as a modification to IBUNE (Lines 172-173 "The "BEAR" method adds a reordering process into the "IBUNE" method to improve the accuracy of input error quantification") .

To overcome this problem, the authors should make last modifications:

1) From the beginning on present the BEAR method as add-on to SMC or potential extension to existing methods like IBUNE, but not as a stand-alone, equal alternative to BATEA or IBUNE

2) Only keep BATEA in the introduction as benchmark reference and for discussion, but not in the comparison – if BEAR shall be presented as an alternative, the authors should have run BATEA as reference. This includes deleting repetitions like lines 311ff in the Discussion chapter 4: "There are two ways to impose restrictions. One is to regard errors and model parameters as a whole in calibration, like the BATEA framework (Kavetski et al., 2006), resulting in a high dimensional computation." Otherwise, there is the impression as if there was an actual comparison made.

Thanks for your additional review and comments. We agree with this summary that the BEAR method is a modification of the input uncertainty quantification of the IBUNE framework, but not as comprehensive as the full implementation of BATEA and IBUNE. To avoid overselling the approach, we have revised all descriptions related to BATEA and IBUNE, as follows:

1) The descriptions in line 53-56 have been modified as follows:

*"Therefore, a modification should be made in the IBUNE approach to improve the accuracy of input error identification.*

*To complete this goal, this study attempts to add a reordering strategy into the IBUNE framework and names this developed algorithm as Bayesian Error Analysis with Reordering (BEAR)."* (line 54-57)

2) The descriptions in Section 2.2 "Considering the limitations of BATEA and IBUNE framework discussed in the introduction, an improved strategy should be explored to avoid the high dimension challenge and meanwhile promote the error estimation accuracy." have been modified as follows:

*"Unlike directly estimating the input error value via existing methods, this study attempts to transform the input error quantification into the rank domain."* (line 108-109)

3) The descriptions about BATEA have been deleted in "2.5 Comparison with other methods" to focus on the comparison between IBUNE and BEAR.

*"In this study, three methods, including the "Traditional" method, "IBUNE" method and "BEAR" method, are compared to evaluate the ability of the BEAR method in estimating the model parameters and quantifying input errors. The "Traditional" method regards the observed input as error-free without identifying input errors (i.e. Eq. (2)), while the other two methods employ a latent variable to counteract the impact of input error and derive a modified input (i.e. Eq.(3)). In the "IBUNE" method, potential input errors are randomly sampled from the assumed error distribution and filtered by the maximization of the likelihood function (Ajami et al., 2007). Although the comprehensive IBUNE framework additionally deals with model structural uncertainty via Bayesian Model Averaging (BMA), this study only compares the capacity of its input error identification. The "BEAR" method adds a reordering process into the "IBUNE" method to improve the accuracy of input error quantification."* (line 195-203)

4) The discussion in Chapter 4 (repetition in lines 311ff) has been modified as follows:

*"In addition, rank estimation can make better use of the knowledge of the input error distribution. In a direct value estimation, it is difficult to keep the overall error distribution the same when the errors are updated in the calibration. The estimated errors are more likely to compensate for other sources of errors to maximize the*

*likelihood function and subsequently be overfitted. By contrast, in rank estimation, the errors at all the time steps are sampled from the pre-estimated error distribution first and then reordered. Whatever the error rank estimates are, they always follow the pre-estimated error distribution, and the compensation effect will be limited. In the IBUNE framework (Ajami et al., 2007), the errors are also sampled from the error distribution, but not reordered. Thus, the error precision at each time step cannot be guaranteed. In the BEAR method, adjusting the sampled errors according to the inferred error rank reduces the randomness of the error allocation in the IBUNE framework (Ajami et al., 2007), which significantly improves the accuracy of the error estimation (as demonstrated by much higher correlations than the IBUNE method in Fig. 2(2)).*

*Unlike formal Bayesian inference, the rank estimation does not update the posterior distribution of the input errors, but optimises their time-varying values through the relationship between the input error rank and corresponding model residual error. The rank estimation is implemented after the model parameters have been updated and the model residual error depends on the input error estimation. Thus, the reordering strategy identifies the optimal input error rank conditional to the model parameters, effectively considering the interaction between the input error and the parameter error. This is akin to calibrating the input errors along with the model parameters in the BATEA framework (Kavetski et al., 2006)."* (line 340-355)

5)   The conclusion in Chapter 5 has been modified as follows:

*"The novelties of this algorithm are: (1) The estimation focuses on the error rank rather than the error value, using the constraints of the known overall input error distribution and then improving the precision of the input error estimation by optimising the error allocation in a time series. (2) The introduction of the secant method addresses the nonlinearity in the WQM transformation and updates the error rank of each input data according to its corresponding model residual."* (line 410-414)

Personally, I find the treatment of the errors still rather arbitrary – by design, the method will minimize residuals between model and observations. I doubt that this is the intention of Bayesian methods. Therefore, it is also no surprise that NSE values from BEAR are often higher (see e.g. Fig. 1 and 2) or the variance of residuals is lower (see Fig. 4).

Thanks for raising this concern. Based on your comments, we included a section that clarifies the BEAR modification compared to classical Bayesian inference, section "==2.3 Bayesian inference of input uncertainty and the BEAR method==" to explain it. Regarding the issue of minimizing residual between model and observations, the effectiveness of our approach does depend on the assumption that the input error is dominant in the residual error. The explanation is as follows:

"*The secant method in the BEAR algorithm is applied to find the optimal ranks of input errors to minimise the model residual errors towards zero, as characterised by the minimized Residual Sum of Squares (RSS). Minimizing the RSS imposes the same effect as maximizing the likelihood function. The effectiveness of this step in quantifying the input errors is based on the assumption that the input error is dominant in the residual error and then minimizing RSS is the same as allocating the total error into the input errors. Otherwise, other dominant sources of errors will affect the estimation of the optimal input errors leading to poor input error identification.*" (==line 156-161==)

We also note that "*Unlike formal Bayesian inference, the rank estimation does not update the posterior distribution of the input errors, but optimises their time-varying values through the relationship between the input error rank and corresponding model residual error. The rank estimation is implemented after the model parameters have been updated and the model residual error depends on the input error estimation. Thus, the reordering strategy identifies the optimal input error rank conditional to the model parameters, effectively considering the interaction between the input error and the parameter error. This is akin to calibrating the input errors along with the model parameters in the BATEA framework (Kavetski et al., 2006)*" (==line 350-355==)

That said, I do see two points why it still might be worth publishing: 1st) The manuscript addresses the problem "input error" that has not been addressed as much over the last years. Yet, it is a very important one, e.g. especially regarding novel data-driven machine-learning models through which input errors might be propagated without regulation since unlike mechanistic models, they do not contain physical relations that might buffer some part of the input error. 2nd) The authors propose the use of the secant method to address the problem and even if I personally do not find the presented procedure to be the "problem-solver" modelers might look out for, readers

can make their mind of whether this method still bears more potential (working with ranks of errors, etc.) or not.

So considering the points above, I would consider the publication of this manuscript as a contribution to the discussion about issues and ideas in addressing input errors, but not as a proven framework to tackle them that outperforms existing methods.

Thanks for your suggestion. We agree with that the contribution of this study should focus on the introduction of rank estimation and the secant method in the input error identification. Therefore, we have revised the descriptions to highlight these points and deleted the statements on tackling the limitations of BATEA and IBUNE (see the above responses).

I thank the authors for their detailed revision of the manuscript. Overall, their revision helped clarifying a lot of uncertain aspects of the algorithm, and their responses to my concerns were mostly satisfactory, but work still remains to be done. Unfortunately, I fear that my main concerns about the theoretical foundations and consequences of the error re-ordering remain inadequately addressed. The derivation provided in Appendix B does not sufficiently clarify these points eithers: the core question (*how exactly the re-ordering affects the base error distribution's statistical moments or hyperparameters, and what the consequences are in Bayesian terms*) remains unaddressed. Since I take it that the authors would like to stick with the error re-ordering approach, I would argue that this leaves you with two possible pathways:

1) Provide a thorough theoretical derivation and in-depth investigation of what effect the error re-ordering really has, and what this means in Bayesian terms. After experimenting a bit with error re-ordering myself (see the Python code snippet below), I believe that a good start point might be couplings or measure transport (Pierre E. Jacob has a nice online lecture series on that called *Couplings and Monte Carlo*), as you seem to convert one distribution (the raw error distribution) into one with different statistical moments, one somehow moulded to the ideal error realizations.

Thank you for this constructive comment. We appreciated the lectures shared by the reviewer from Pierre E. Jacob and carefully considered the method on *Couplings and Monte Carlo*. However, we believe the goal of coupling is intrinsically different from our method and is not feasible in the rank estimation. *Coupling* aims to gain a posterior distribution with different statistical moments, while the BEAR method does not change the error distribution (by sampling the errors from the same pre-estimated distribution), but aims to adjust the positions of sampled errors according to the inferred ranks via the secant method. In other words, re-ordering won't change the overall statistical moments on the error population (i.e. mean and standard deviation. The details have been discussed in the additional section "==4.2 The effect of reordering on the error realization==".

2) Alternatively, you could simply drop the "Bayesian" attribute from your study or replace it with "Pseudo-Bayesian". This might require that you adjust the

acronym. Even without full theoretical justification, your algorithm can still provide a useful heuristic, and maybe that is enough. Even in this case, however, I believe the manuscript would benefit from an isolated analysis of what mechanically happens to the input error distribution when subject to reordering. I have provided a few thoughts on this below. As a consequence, I would recommend another round of major revisions. If you follow option (2), I recommend specifically to add a new section into the theory/methodology chapter which explores and illustrates the effects of re-ordering errors on the raw error distributions in detail (this is important for the reader's understanding of the approach– it should be part of the main manuscript, not the Appendix). To make space for this section, you could absorb some of the practical comments in the discussion (specifically, sections 4.1 and 4.3). There are also a lot of tangential comments addressing reviewer concerns throughout the manuscript which could be removed if their key points are addressed in this new section. This might also support the narrative thread of the manuscript by helping you to avoid the need to go on explanatory tangents. As I don't want to leave you hang out to dry on such a large and amorphous task, I would specifically suggest exploring a simple example case in this proposed section. Specifically:

• Ignore the model (for the purpose of error reordering this is unnecessary); instead, skip straight to positing some hidden "ideal" sequence of error realizations which perfectly compensate the true residual error (similar to your figure A1); derive the corresponding ranks;

• Use a simple, structured residual error sequence to make it easier to read the induced effect. The sequence doesn't have to be realistic, merely insightful;

• Use a residual error distribution perfectly adjusted to the structured error you defined. In a synthetic test case, it's easy to derive an empirical cdf.

• Then explore the consequences of re-ordering for time series of different length or input error distributions of different quality. Discuss the statistical moments after reordering.

We admit the sampling and reordering strategy in the BEAR method itself is not a formal Bayesian approach, but rather an additional step in the existing Bayesian

inference methods to find the deterministic relationship between the model residual error and the observational error. We have updated the text to make this point explicit, with an additional section "2.3 Bayesian inference of input uncertainty and the BEAR method" that clearly articulates the relationship between our approach and the classical Bayesian approach (see the following reply). Additionally, according to your suggestion, we have deleted the section "4.3 The extension to other modeling scenarios" and moved the summary of this content into Section 5 "Conclusion and recommendation".

I have provided an example Python script for this below and appended some of its result figures for different time series lengths of 10, 100, and 1000 in Figures 1, 2, and 3. In this example, I arbitrarily assumed the true/ideal error realizations to follow a sine curve. Note that something more realistic (like random samples from a Gaussian distribution) would have also worked, but the simplicity of a sine curve makes it significantly easier to read the effect of the re-ordering.

Some thoughts on the results:

As I suspected in major comment #7 for the first round of revisions, the longer the time series, the more likely the method is to achieve a "perfect fit", so the effect of error re-ordering depends on the length of the time series. You discuss this briefly in the manuscript, but I think that this is among the most important mechanisms of BEAR, so it is worth demonstrating in isolation. For short time series (Figure 1), error re-ordering can already induce some degree of improvements by causing the marginal sample mean to follow the "ideal" error realizations; at the same time, the marginal error standard deviation decreases. This effect is exacerbated for longer time series (T=100, Figure 2, and T=1000, Figure 3). The consequence seems to be that the unordered, raw error distribution is "molded" to the ideal error realizations. I suspect (and you seem to share these suspicions in your responses) that in the limit of an infinitely long time series, error realizations would be compensated perfectly. This is important to discuss for prospective users of your manuscript, as it affects the algorithm's behaviour in somewhat unexpected ways.

[Figure]

*Figure 1. Effect of error re-ordering for a perfect error distribution, an ensemble size of N=1000 and a time series length of T=10. The upper subplot shows the "ideal" realizations to compensate some residual error. The left centre plot shows N unordered realizations of an error distribution with the correct statistical moments of the ideal realizations (obtained by forming a cdf for a full sine wave). The right centre plot shows the marginal mean and standard deviation at each time step. The left bottom plot shows the N error realizations in the subplot above after ordering, and the right bottom plot shows the corresponding marginal mean and standard deviation.*

[Figure]

*Figure 2. Effect of error re-ordering for a perfect error distribution, an ensemble size of N=1000 and a time series length of T=100. The upper subplot shows the "ideal" realizations to compensate some residual error. The left centre plot shows N unordered realizations of an error distribution with the correct statistical moments of the ideal realizations (obtained by forming a cdf for a full sine wave). The right centre plot shows the marginal mean and standard deviation at each time step. The left bottom plot shows the N error realizations in the subplot above after ordering, and the right bottom plot shows the corresponding marginal mean and standard deviation.*

[Figure]

*Figure 3. Effect of error re-ordering for a perfect error distribution, an ensemble size of N=1000 and a time series length of T=1000. The upper subplot shows the "ideal" realizations to compensate some residual error. The left centre plot shows N unordered realizations of an error distribution with the correct statistical moments of the ideal realizations (obtained by forming a cdf for a full sine wave). The right centre plot shows the marginal mean and standard deviation at each time step. The left bottom plot shows the N error realizations in the subplot above after ordering, and the right bottom plot shows the corresponding marginal mean and standard deviation.*

Other interesting things to visualize might be what happens if the error distribution is not perfect (for this, just replace "vals = dist(np.random.uniform(size=(1000, resolution)))" in the code with some other distribution). You already show this indirectly in your models, but demonstrating this effect in isolation rather than through the lens of performance metrics might be a lot clearer. Feel free to take inspiration from my example code or use it directly. I have attached it at the end of this manuscript.

Of course, this code snippet just demonstrates what happens when we are re-ordering error realizations, not how you arrive at the error ranks and their interaction with the parameter inference in the first place (which are potentially additional topics to discuss). hope that even if you decide to follow option (1), this snippet might give you some ideas on where to start with the Bayesian justification. Good luck!

We appreciate your open, detailed analysis of the impact of the time series length, which helped inspire us to analyse the effect of reordering. We added one section "4.2 The effect of reordering on the error realization" and summarized all the results in the following Figure 5 to demonstrate the mechanics of error reordering.

First, we want to point out that we agree that the length of the time series will affect the

efficacy of the method. As Figure 5 shows, larger data lengths bring more accurate estimation of model parameter and input errors. The reason for this has been discussed in Section 4.2. Usually, we are operating on the assumption that we have a representative length of sampled values, certainly enough that the model parameters are estimated properly, and that this is not unusual for the applications we are looking at (where we would need at least a year of data to estimate the parameters).

Considering the changes of the marginal mean of the standard deviation, we discuss this in Section 4.2 as follows:

"*Figure 5 demonstrates the mechanics of input error reordering in the BEAR method and input error filtering in the IBUNE method to understand their effects on the input error realizations and model parameter estimation. The 1st sequence represents the situation where the raw input errors are randomly sampled from the pre-estimated error distribution, therefore, their marginal means and standard deviations are the same as the parameters of overall error distribution (demonstrated as the cyan lines in column (c)). In the later sequence. these errors are optimised via different methods. In the IBUNE method, these sampled input error series are selected by the maximized likelihood function and the interval of input errors become a little converged (in (b1)) and their marginal standard deviations reduce slightly (in (c1)). However, in the BEAR method, these input errors are rcordered according to the inferred ranks via the secant method, and the reordered errors gradually converge to the true values (represented by the blue interval are near the red line in (b2)). Therefore, their marginal means are similar to the true values and their marginal standard deviations reduce to zero (in (c2)). In the BEAR method, the promotion of the input error identification in the sequential updating will improve the model parmaeter estimation, represented by the posterior distribution of model parameter b converging to the true value in (a2). While in the IBUNE method, the identification of input errors is not precise and the bias of the model parameter still exists in (a1).*

*The data length can affect the efficacy of the BEAR method but impose little effect on the IBUNE method. The IBUNE method takes advantage of the stochastic errors and keeps the marginal error distribution almost constant. The input error realization at each time step seems independent, only filtered by the overall likelihood function. Therefore, the number of sampled errors does not matter in the IBUNE method.*

*However, in the BEAR method, the input errors at all the time steps are not sampled independently, they are from one sample set. Therefore, before or after reordering, all errors will keep the same statistical features of the input error distribution, and only their marginal distribution changes due to the convergence to the unknown true values. Figure 5 (b2) demonstrates that when the data length (the same as the error number) is small, the input error estimation might be biased from the true values. This likely arises from the above-mentioned sampling bias or the impacts of the model parameter error because the sampling bias reduces with the larger number of error samples and the impacts of parameter error are more likely to be offset when the data length is long.”*
(line 357-379)

[Figure]

**Figure 5** **Comparision of the results for scenario 1 in the synthetic case 1, the ensemble particle size of *N*=100 at different sequences of calibration (represented in different colours), via different methods (row 1: IBUNE method; row 2: BEAR method) and under different data lengths in calibration (the upper group: data length is 50; the lower group: data length is 1000, selects 50 data the same as the upper group to show). Column a shows the probability density of model parameter *b* at different sequences of calibration. The other model parameters have the same pattern of change and thus there is no need to show. Column b shows the value interval of input error realizations of 100 particles after reordering in the BEAR method or filtering in the IBUNE method and Column c shows the corresponding marginal mean and standard deviations at each time step. The 1st sequence (in cyan) shows the raw input errors of random sampling, before reordering or filtering.**

**Specific comments** (any line numbers I list correspond to the non-track-changes manuscript):

Line 47: the same multiplier applied to one storm event

Maybe "the same multiplier applied to all time steps of one storm event" might be better, if I understand this part correctly; using "same" for a singular object ("storm event") sounds a bit strange.

Thanks for your suggestion. This has been changed as recommended. (line 48)

Line 184: the time scale is typically set as daily and the spatial scale is set as the catchment

This is not particularly clear. I assume you simulate in daily time steps, and aggregate the catchment's (presumably) surface area into a single spatial unit? If so, it might be better to replace this with "thus, we use daily time steps and consider the catchment a single, homogeneous spatial unit" or something along these lines.

Thanks for your suggestion. This has been clarified as recommended. (line 214)

Lines 302: From this point of view, it is more efficient to estimate the error rank than estimate the error value, This sentence ends on a comma, not a period.

This has been corrected. (line 332) Thanks.

Line 309-311: Besides, to avoid the high-dimension calculation, modifying each input error according to its corresponding residual error only works in the rank domain. In the value domain, if there is no constraint on the estimated input errors, they will fully compensate for the residual error to maximize the likelihood function and subsequently be overfitted.

This requires more discussion in the revised manuscript, as it seemingly contradicts what you write in the paragraph immediately prior: In the previous paragraph, you recommend sampling error realizations repeatedly and selecting the optimal realization to overcome "sampling bias" and improve the fit to the actual observations. However,

in this paragraph you praise this very same sampling bias for preventing overfit. These are contradictory messages: provided you get the ranks right, if you were to resample an infinite number of times, you would eventually get an error realization which compensates the true error perfectly (even if your input error distribution is a really poor approximation to the "true" error distribution), thus negating your protection against overfit. The fact that this protection against overfit depends on the length of the time series might not be that much of an issue if you interpret your approach merely as a heuristic, but even in this case you need some practical guidelines on when to re-sample for short time series. The proposed dedicated section might help clearing some of this confusion up.

Thanks for your concerns. We believe the manuscript needs more clarification on the difference between "sampling bias" and "compensation effect".

[Figure]

*Figure R1 The cumulative probability of sampled errors and true errors (sample number =10 )*

In Figure R1, three groups of errors are sampled from the same normal distribution $(N(0,0.5^2))$, but for the same order (with the same cumulative probability distribution), the sampled errors from different groups (in different colours) are not the same. The value difference at the same order is referred to as "sampling bias". In order words, even if all the error ranks are estimated right, there is still a difference between the reordered error series and the true values, which comes from the sampling step, not the reordering step. The sampling bias is more significant when the error number is smaller

or the variance is larger. Repeatedly sampling and selecting the optimal set can reduce the potential impact of this.

It should be noted that sampling bias cannot prevent overfitting. Here, these sentences aim to explain that overfitting (referring to the compensation effect) is more likely to appear in value estimation due to the lack of constraint on the error distribution. In value estimation, it is difficult to keep the overall distribution of the sampled errors the same when the errors are updated in calibration, then the compensation for the residual error is more likely to appear. By contrast, in rank estimation, the errors at all time steps are sampled from the pre-estimated error distribution. Whatever the error rank estimates are, they always follow the error distribution, and the compensation effect will be reduced.

From the above, this has been clarified as follows:

"*For the same error distribution and the same cumulative probability distribution (corresponding to the same error rank), the errors sampled at different times could be largely different, especially for a small sample size (depending on the data length) or a large $\sigma$ of the assumed error distribution. This problem can be addressed by selecting the optimal solution from multiple samples according to the maximum likelihood function.*" (line 333-337)

"*In addition, rank estimation can make better use of the knowledge of the input error distribution. In a direct value estimation, it is difficult to keep the overall error distribution the same when the errors are updated in the calibration. The estimated errors are more likely to compensate for other sources of errors to maximize the likelihood function and subsequently be overfitted. By contrast, in rank estimation, the errors at all the time steps are sampled from the pre-estimated error distribution first and then reordered. Whatever the error rank estimates are, they always follow the pre-estimated error distribution, and the compensation effect will be limited.*" (line 340-345)

Line 320-322: Thus, unlike formal Bayesian inference, the BEAR method does not update the posterior distribution of the input errors, but identifies the input error through the deterministic relationship between the input error and model parameter.

As far as I can see, this is the first time you mention that BEAR is not a formal Bayesian

inference method, so I suspect you would go for option (2). In any case, mentioning this in the discussion for the first time is a bit late: Something this important should be stated earlier, as early as the introduction or abstract. If you add the section exploring the consequences of error reordering, this would also be a good place to elaborate on this.

Aside from this manuscript structuring argument, the statement in these lines is also unfortunately wrong. Refer to the attached figures for demonstration. I also elaborate more on this two comments below.

Thanks for this insightful comment.

We admit this statement is a bit sudden. We have clarified this as follows to show its connection with the preceding discussion in Section 4.1.

"*Unlike formal Bayesian inference, the rank estimation does not update the posterior distribution of the input errors, but optimises their time-varying values through the relationship between the input error rank and corresponding model residual error. The rank estimation is implemented after the model parameters have been updated and the model residual error depends on the input error estimation. Thus, the reordering strategy identifies the optimal input error rank conditional to the model parameters, effectively considering the interaction between the input error and the parameter error. This is akin to calibrating the input errors along with the model parameters in the BATEA framework (Kavetski et al., 2006).*" (line 350-355)

In addition, we have added the section "4.2 The effect of reordering on the error realization" to better explain this statement.

Line 383: "However, the work in this study still identifies a few areas needing to be explored."

Nit-pick: This sentence is a bit unwieldy, in my opinion. How about "However, this study identifies a few areas which still need to be explored:"?

Thanks for your suggestion. This has been changed as recommended. (line 415)

Response file / Response to major comment #7: "*A point of clarification: in each subsequent iteration of the BEAR algorithm, a new population of input errors are sampled from their a priori distribution. This means that the distribution of errors at each iteration is the same prior to reordering, i.e. the population of errors do not converge to a distribution that has different statistical features (mean, standard deviation, skewness).*"

You are of course correct, but I fear your comment misses the point somewhat: While it is undoubtedly true that the raw input error distribution never changes, BEAR *does* effectively update the input error distribution. In the figures from my code snippet, this can be seen in the bottom right subplots when compared to the centre right subplots. It becomes evident that the both the effective mean and standard deviation (and likely higher statistical moments as well) change dramatically after reordering. In essence, the fact that you are re-ordering transforms your raw error distribution and causes you to sample some different latent distribution instead. What this distribution really is remains the key question of your entire approach. If you can figure that out, you'll be one large step closer to justifying this approach theoretically. =)

Thanks for your comments. We should clarify the difference between the overall error distribution of all the time steps (sampled in a single iteration of the algorithm) vs the error distribution at each time step.

For the overall error distribution of all the time steps, this does not changes before or after reordering and in subsequent iterations of the algorithm. In the BEAR method, the errors are firstly sampled from the pre-estimated error distribution (error number = number of time steps) and randomly distributed on the different time steps. Then all the random samples are reordered according to the inferred error ranks, but the overall distribution stays the same.

For the error distribution at each time step, the aim of error identification in this kind of study is to make it converge to the true value. Just like the demonstration in your figures, the ideal result is that its mean is the same as the true value (the residual error in your cases), and its standard deviation is as small as possible.

Therefore, reordering does not cause us to sample some different latent distribution instead. The errors are always sampled from the pre-estimated overall error distribution. The converged error distribution at each time step after reordering is what we're trying

to achieve.

In a sense, what you are doing seems distantly related to ideas in measure transport, see for example Marzouk et al. (2016) for an overview. In measure transport, the ultimate goal is to indirectly sample from an (almost) arbitrary target distribution. This is achieved by sampling a simple reference distribution instead (for example a multivariate standard Gaussian), then converting these reference samples through a deterministic function into samples from the target distribution. Of course, finding the correct transformation function is the key objective of this entire endeavour, and consequently its main challenge. In your study, you approach this from the opposite direction: you have some transformation, now you should find out what distribution you are sampling.

Marzouk, Y., Moselhy, T., Parno, M., & Spantini, A. (2016). An introduction to sampling via measure transport. *arXiv preprint arXiv:1602.05023*; https://arxiv.org/abs/1602.05023.

In summary, I would say the parallels to your approach are as follows: even though the reference distribution (corresponding to your raw input error distribution) never changes, the *pushforward distribution* (corresponding to the latent distribution your re-ordered error realizations are effectively sampled from) changes with the transformation function (in your case, the re-ordering according to different error ranks).

Yes, we totally agree with your summary that "the reference distribution (corresponding to your raw input error distribution) never changes, the pushforward distribution (corresponding to the latent distribution your re-ordered error realizations are effectively sampled from) changes." It should be noted that the reference distribution is for the overall distribution of all the errors, while the pushforward distribution is for the error at each time step. Therefore, in the above response, we differentiate the overall error distribution and the error distribution at each time step, and based on your analysis, we have clarified this in Section 4.2 from the changes of the marginal mean and std.

After learning the paper describing measure transport, we have not found an effective way to apply the proposed method for input error estimation or combine it with the secant method, which we believe needs further investigation of this method. Applying

the measure transport approach in this framework is an exploration of a totally new method and we believe the reordering error approach we propose (according to inferred ranks) seems an easier way to identify the transport of the marginal error distribution. However, we agree that the measure transport approach suggests an interesting future approach to integrate or compare with the method we have proposed, potentially building a more solid theoretical foundation in formal Bayesian inference.

---

## Author Response (AR3)

Thanks for the editor and reviewer's comments. We have responded to each point in turn in the following sections. The comments are provided in blue text and our responses are organized point-by-point in black text. The manuscript text after the proposed changes is shown in "*black italics*". The number of the line, equation and section refers to the revised version of the manuscript without track changes, shown in yellow highlight.

**Editors' comments**:

I agree with the reviewer comments regarding the specifically mentioned sentences that are confusing and should be deleted.

We have deleted the mentioned sentences.

Regarding the other comment of referring to the BEAR method as pseudo-Bayesian, with the additional clarifications that the authors included in past review rounds, as far as I understand, it would depend on the technical details of how the BEAR algorithm is combined with a certain inference approach whether the final outcome is a formal or an informal Bayesian approach. Therefore, I think the need to strictly refer to the BEAR algorithm as a pseudo-Bayesian approach can be relativized. I therefore suggest the authors keep the current name of the algorithm.

Thanks for your suggestion. We agree with your opinion that whether it is a formal or an informal Bayesian approach depends on the inference approach the reordering strategy is built in. Considering the comments by the editor and the reviewers, we have kept 'BEAR' as the method name.

Additional comments:

I suggest the authors rearrange the structure of the manuscript such that the results are presented in a separate Section 4. In Section 3, the description of BwMod, as well as the setup of the case studies can be kept, but all the results should be clearly separated in a following section.

Thanks for your suggestion. We have rearranged the structure of the Case Study section as recommended. The titles of the rearranged sections are as follows:

3 Case studies

    3.1 Water quality model: the build-up/wash-off model (BwMod)

    3.2 Setup of synthetic case study

    3.3 Setup of real case study

4 Results

    4.1 Case study 1: Synthetic data suffering from input errors

    4.2 Case study 2: Synthetic data suffering from input errors and output observation errors

4.3 Case study 3: Real data

We have also changed the related descriptions. The main changes are as follows:

"*Section 3 describes the setup of these case studies and Sect. 4 demonstrates their results. Section 5 evaluates the BEAR method and its implementation. Finally, Section 6 outlines the main conclusions and recommendations for this work*." (line 65-67 in Introduction)

"*To sum up, two synthetic case studies have been analysed: Case study 1 generates synthetic data only suffering from input errors to evaluate the effectiveness of the BEAR method in isolating the input error and the model parameter error; Case study 2 additionally considers output observation errors via synthetic data generation to evaluate the impacts of other sources of error on the BEAR method.*" (line 260-263 in Case studies)

"*Nash-Sutcliffe efficiency(NSE) is selected to measure the difference between the modified input in Case study 2 and the true input.*" (line 318-319 in Results)

"*Figure 4 compares the SDs of estimated input errors, the variances of model residual errors, and reliability and sharpness of model simulations among the four calibration scenarios and three calibration methods in the real case study.*" (line 326-327 in Results)

The first paragraph of the conclusion section can be extended, and the second one shortened. The second paragraph is interesting and necessary, but some of the sentences are better placed in the discussion section (and maybe slightly extended).

Thanks for your suggestion. the conclusion section has been modified as follows:

"*Observation uncertainty in input data is inherently independent of the model process and the input error model can be estimated prior to the model calibration and simulation by analysis of the data itself. Taking advantage of the prior information of an input error model, a new method, Bayesian error analysis with reordering (BEAR), is proposed to approach the time-varying input errors in WQM inference. It contains two main processes: sampling the errors from the assumed input error distribution and reordering them with the inferred ranks via the secant method. This approach is demonstrated in the case of TSS simulation via a conceptual water quality model, BwMod. Through the investigation of synthetic data and real data, the main findings are as follows:*

*(1) The estimation of the BEAR method focuses on the error rank rather than the error value in the existing methods, which can take advantage of the constraints of the known overall error distribution and then improve the precision of the input error estimation by optimising the error allocation in a time series.*

*(2) The introduction of the secant method addresses the nonlinearity in the WQM transformation and can effectively update the error rank of each input data according to minimizing its corresponding model residual.*

*(3) The ability of the BEAR method in decomposing the input error from model residual error is limited by the accuracy and selection of the input error model and is impacted by model structural uncertainty and output observation uncertainty.*

*Therefore, the study identifies several areas which need further analysis. Firstly, the availability of*

*prior knowledge of the input error model is important. When this information is not reliable or even cannot be estimated, a significant issue is the selection of a suitable error distribution. Thus, a general measure should be found to judge whether an error model is appropriate, especially in real cases where the "true" information is limited. Secondly, this study focuses on identifying the input errors in model calibration and the derivation of the BEAR method is based on the assumption that the input error is dominant in the residual error. If the reordering strategy is developed within a more comprehensive framework to quantify multiple sources of error, this assumption will be relaxed, the interactions amongst these error sources might be well-identified and the quantification of individual errors might be improved. This study provides a starting point for developing the rank estimation via the secant method to identify input error. Further study is necessary to modify the algorithm and improve confidence in extended case studies or model scenarios."* (line 418-441)

Specific comments:

Line 35: Replace "For the surrogate error, its probability distribution" by "The probability distribution of the surrogate error"

This has been replaced. (line 39)

Line 37: Replace "more" by "larger"

This has been replaced. (line 41)

Line 52: Replace "input data" by "input data point", if this is what is meant here

Yes, this has been replaced. (line 56)

Line 60: Replace "two synthetic cases and a real case" by "two synthetic and one real case study". Check also other instances where this change is appropriate.

Yes, this has been replaced. (line 63). In addition, the contents have been replaced by "one synthetic and one real case study" (line 87).

Line 85: Replace "appealing" by "common"

This has been replaced. (line 89)

Line 147: Replace "can be" by "is often"

This has been replaced. (line 151)

Reviewer's comments:

Thanks to the authors for their effort in improving the manuscript. Overall, the changes made have increased clarity and do help to better understand the working principles behind BEAR and its purpose. Yet, some of the revised sections point into a direction that I struggle to comply with: "Therefore, a modification should be made in the IBUNE approach to improve the accuracy of input error identification." (l. 54-55) I disagree with the revised version of this sentence and therefore suggest to delete it. It indicates that there was a flaw in the IBUNE method that has to be fixed. However, as Bayesian approach, IBUNE simply samples errors without processing them unlike BEAR, which is also stated in l. 344ff.: "In the IBUNE framework (Ajami et al., 2007), the errors are also sampled from the error distribution, but not reordered. Thus, the error precision at each time step cannot be guaranteed. In the BEAR method, adjusting the sampled errors according to the inferred error rank reduces the randomness of the error allocation in the IBUNE framework…" This randomness in sampling is part of the fully Bayesian approach and not something that per se has to be fixed. That said, I find the second sentence about a "guaranteed error precision" unclear and obsolete, and suggest deleting it.

Thanks for your comments. We have deleted "Therefore, a modification should be made in the IBUNE approach to improve the accuracy of input error identification." ==in line 54-55== and "Thus, the error precision at each time step cannot be guaranteed." ==in line 344==.

Overall, this pertains to the general question of whether BEAR is a full Bayesian approach that I referred to as "arbitrary error treatment" in earlier rounds of reviewing and that Reviewer 2 highlighted in great detail pointing also at the theoretical problems underlying the approach. This is clarified in l. 154 ff. to some degree, i.e. "the BEAR method does not provide formal Bayesian inference". However, this statement is still somewhat hidden in the article and therefore I would like to second the suggestion of reviewer 2 in generally renaming the method a "Pseudo-Bayesian error analysis with reordering". Renaming BEAR to pBEAR (for example) might not appear too appealing to the authors but it would be a more honest name of the method and therefore increase scientific soundness. Hence, I suggest publishing the manuscript with this minor name modification all over the manuscript. I stick to my former evaluation that this manuscript is a valuable contribution to the general discussion of input error treatment. The methodology of BEAR is an idea worth publishing and with the label "pseudo-Bayesian" this will also be a clear contribution to a broad audience.

General: rename the method form "Bayesian" to "pseudo-Bayesian"

We appreciate the reviewer's concerns about the BEAR name. Balancing the authors request with the editor's comment, we have retained BEAR as the method name, as we believe the formal/informal Bayesian nature of the algorithm will depend on the inference approach the reordering strategy is built in.

---

## Author Response (AR4)

In this turn, there is no change required by the editor and reviewers.